# Prokineticin-2 prevents neuronal cell deaths in a model of traumatic brain injury

Zhongyuan Bao[1], Yinlong Liu[2], Binglin Chen[1], Zong Miao[1], Yiming Tu[1], Chong Li[1], Honglu Chao[1], Yangfan Ye[1], Xiupeng Xu[1], Guangchi Sun[1], Pengzhan Zhao[1], Ning Liu[1], Yan Liu [3], Xiaoming Wang [4], Sin Man Lam[5], Valerian E. Kagan [6,7], Hülya Bayır [6,8,9] & Jing Ji [1✉]

Prokineticin-2 (Prok2) is an important secreted protein likely involved in the pathogenesis of several acute and chronic neurological diseases through currently unidentified regulatory mechanisms. The initial mechanical injury of neurons by traumatic brain injury triggers multiple secondary responses including various cell death programs. One of these is ferroptosis, which is associated with dysregulation of iron and thiols and culminates in fatal lipid peroxidation. Here, we explore the regulatory role of Prok2 in neuronal ferroptosis in vitro and in vivo. We show that Prok2 prevents neuronal cell death by suppressing the biosynthesis of lipid peroxidation substrates, arachidonic acid-phospholipids, via accelerated F-box only protein 10 (Fbxo10)-driven ubiquitination, degradation of long-chain-fatty-acid-CoA ligase 4 (Acsl4), and inhibition of lipid peroxidation. Mice injected with adeno-associated virus-Prok2 before controlled cortical impact injury show reduced neuronal degeneration and improved motor and cognitive functions, which could be inhibited by Fbxo10 knockdown. Our study shows that Prok2 mediates neuronal cell deaths in traumatic brain injury via ferroptosis.

[1] Department of Neurosurgery, the First Affiliated Hospital of Nanjing Medical University, Nanjing, China. [2] Department of Neurosurgery, the Affiliated Suzhou Hospital of Nanjing Medical University, Suzhou Municipal Hospital, Suzhou, China. [3] Institute for Stem Cell and Neural Regeneration, School of Pharmacy, Nanjing Medical University, Nanjing, China. [4] Department of Immunology, Nanjing Medical University, Nanjing, China. [5] LipidALL Technologies Company Limited, Changzhou, China. [6] Center for Free Radical and Antioxidant Heath, Department of Environmental and Occupational Health, University of Pittsburgh, Pittsburgh, PA, USA. [7] Laboratory of Navigational Redox Lipidomics, IM Sechenov Moscow State Medical University, Moscow, Russian Federation. [8] Safar Center for Resuscitation Research, Department of Critical Care Medicine, University of Pittsburgh, Pittsburgh, PA, USA. [9] Children's Neuroscience Institute, UPMC Children's Hospital of Pittsburgh, Pittsburgh, PA, USA. ✉email: jijing@njmu.edu.cn

Traumatic brain injury (TBI) is the leading cause of mortality and disability worldwide[1,2]. TBI develops as a two-stage process. The first stage is a mechanical injury of brain tissue that directly leads to neuronal death. At present, surgery is the first choice for treatment during this stage[3–5]. The second stage includes spreading of the damage and deterioration of the surrounding microenvironment caused by several mechanisms—pro-inflammatory response, oxidative stress, local hypoxia/reoxygenation, and accumulation of neurotoxic substances[6,7]—culminating in neuronal death due to temporally dependent activation of apoptosis or one of the necro-inflammatory programs such as necroptosis, pyroptosis, and autophagy[8–10]. Appreciation of the engagement of distinct death mechanisms has been an impetus for designing neuro-therapeutical approaches[11–13] albeit with limited success, thus indicating that additional yet to be deciphered mechanisms of neuronal death may be involved in TBI pathogenesis.

Among them is ferroptosis, a recently identified cell death program defined by the accumulation of phospholipid hydroperoxides, particularly HOO-arachidonoyl-PE, catalyzed by iron-dependent mechanisms and insufficiency of the thiol regulation[14–16]. Accordingly, acyl-CoA synthetase long-chain family member 4 (Acsl4), a key enzyme in arachidonic acid (AA)-PE biosynthesis, has been shown to contribute to the execution of ferroptosis[17,18]. Pathogenesis of several acute conditions (stroke and TBI)[19,20] and chronic neurodegenerative diseases (Parkinson's disease and Alzheimer's disease)[14,21–23] have been associated with ferroptosis.

Prokineticins are newly identified chemokines with a critical role in the immune system and inflammatory diseases[24] and likely participation in the pathogenesis of neurological disorders[25–27]. Prokineticin-2 (Prok2), a member of the prokineticin family, has been discovered as a component of black mamba venom and frog skin[28,29]. While initially its biological activity was associated with the regulation of gastrointestinal motility, later studies demonstrated that it was involved in olfactory bulb neurogenesis[30] and regulation of sleep and wake states in adult zebrafish[31]. Furthermore, Prok2 signaling plays a protective role in nigral dopaminergic neurons[32]. However, the function of Prok2 in TBI, particularly in the context of neuronal cell death, has not been studied.

In the present study, we investigate the regulatory role and mechanisms of Prok2 in neuronal ferroptosis. We demonstrate that Prok2 increases the level of Fbxo10, a ubiquitin ligase binding to Acsl4, thus promoting Acsl4 ubiquitination and degradation. This Prok2-driven cascade alleviates ferroptosis, preserves mitochondrial function, and protects neurons against TBI.

## Results

**Prok2 is upregulated after brain injury.** A total of three non-contusive brain tissues and five TBI tissues were obtained from the Tissue Bank of the First Affiliated Hospital of Nanjing Medical University, Nanjing, China, to perform transcriptome sequencing. Patient specimen information is shown in Table 1. The top 10 genes with the most robust differences based on the log2 (fold change) values were selected (Table 2). The difference in mRNA levels between the human control and TBI groups was confirmed by quantitative reverse transcriptase PCR (qRT-PCR) (Supplementary Fig. 1a). Six differentially expressed genes, including *IL1RL1, S100A8, S100A9, S100A12, PROK2*, and *CXCR1*, were screened out. To determine the neuronal association of the TBI-induced changes, we examined the expression levels of five of these genes—*Il1rl1, S100a8, S100a9, Prok2*, and *Cxcr1*—in different types of brain cells using the Brain RNA Data Base[33] (https://web.stanford.edu/group/barres_lab/brain_rnaseq.html) (Supplementary Fig. 1b–f). Prok2 was selected for further analysis because other genes were expressed at low levels in neurons.

Western blot assessments of Prok2 levels demonstrated that its expression was markedly increased after TBI (Fig. 1a, b). This was confirmed by the immunostaining (Fig. 1c). Using a mouse model of controlled cortical impact (CCI) (Fig. 1d), we assessed Prok2 expression in the peri-contusional area and found a time-dependent increase in the protein expression ($P < 0.05$, $n = 3$ mice per group) (Fig. 1e, f). Given that progression of injury was accompanied by upregulation of Prok2, we further explored its role in the brain injury.

Prok2 affects biological functions by interacting with its receptor, Prokr2. Therefore, we further examined the time course of Prokr2 expression in CCI tissues (Supplementary Fig. 2a). Decreased Prokr2 and NeuN expression was found after CCI. As Prokr2 was mainly expressed in neurons (Supplementary Fig. 2b), we performed dual immunofluorescence staining for Prokr2 (Green) and NeuN (Red) to specifically confirm that presence of Prokr2 was established almost exclusively in NeuN-positive cells. The intracellular fluorescence intensity of Prokr2 was not affected by CCI (Supplementary Fig. 2d). Furthermore, co-immunoprecipitation (Co-IP) assays showed that CCI did not impact the intracellular relationships between Prok2 and Prokr2, in spite of the changes in the average expression of the proteins in the brain tissue (Supplementary Fig. 2e). This is in line with the results of dual immunofluorescence staining for Prokr2 (Green) and Prok2 (Red), demonstrating that CCI caused increased Prok2 levels but no changes in Prokr2 expression in NeuN-positive neurons (Supplementary Fig. 2f). These results indicate that the decreased total expression of Prokr2 was due to the reduced number of NeuN-positive cells without changes in its intracellular expression in NeuN-positive cells.

**In vitro studies of ferroptosis-associated enhanced Prok2 expression in neurons.** Analysis of Prok2 in the brain tissue may be complicated by different levels of its expression in various types of cells. Immunofluorescence showed a more robust elevation of Prok2 in neurons (labeled by NeuN) 24 h after CCI than in astrocytes (labeled by GFAP) and microglia cells (labeled by

**Table 1 Demographic and clinical characteristics of TBI patients.**

| TBI patient | Accident type | Injury area | Gender | Approximate time delay between TBI and tissue cryopreservation (h) | Age | GCS scores |
|---|---|---|---|---|---|---|
| 1 | Traffic accident | Left temporal lobe | Male | 6.5 | 42 | 9 |
| 2 | Fall | Occipital lobe | Female | 9 | 70 | 8 |
| 3 | Traffic accident | Right temporal lobe | Male | 6 | 52 | 10 |
| 4 | Traffic accident | Left temporal lobe and occipital lobe | Female | 7 | 58 | 5 |
| 5 | Traffic accident | Left parietal lobe | Female | 5.5 | 60 | 7 |

**Table 2 The top 10 genes with the most significant differences were listed.**

| Number | Gene | log2 Fold change |
|---|---|---|
| 1 | Ficolin 1 (FCN1) | 3.7943 |
| 2 | Interleukin 1 receptor, type II (IL1R2) | 3.8505 |
| 3 | Interleukin 1 receptor-like 1 (IL1RL1) | 4.11 |
| 4 | Free fatty acid receptor 2 (FFAR2) | 6.098 |
| 5 | Osteomodulin (OMD) | −3.7845 |
| 6 | S100 calcium binding protein A8 (S100A8) | 3.5754 |
| 7 | S100 calcium binding protein A9 (S100A9) | 3.7295 |
| 8 | S100 calcium binding protein A12 (S100A12) | 4.071 |
| 9 | Prokineticin 2 (Prok2) | 3.4119 |
| 10 | Chemokine (C-X-C motif) receptor 1 (CXCR1) | 5.7263 |

iba-1). In fact, astrocytes and microglia showed low levels of Prok2 (Fig. 1g).

As increased Prok2 levels may be important contributors to the CCI-induced neuronal injury, we performed in vitro mechanical stretch experiments with primary cortical neurons (Fig. 1h, i). Primary neurons were identified by immunostaining assays using dual-labeling with NeuN and Map2. The cells were extracted from newborn mouse brain cortices and cultured for 5 days before use. An early (3 h) increase in Prok2 levels was detected after the stretch. At 9 h after stretch, increased levels of cleaved caspase-3 were also found (Fig. 1j, k). Further, several other insults were used to test their capacity to change Prok2 expression. LPS (2 μg/ml) (Supplementary Fig. 3a, b) or corticosterone (1 μM) (Supplementary Fig. 3c, d) did not alter Prok2 expression after treatments for 24 h. However, exposure to $H_2O_2$ (12 h) (Supplementary Fig. 3e–g) or excitotoxic glutamate (Supplementary Fig. 3h–j) increased Prok2 expression. Notably, both $H_2O_2$-induced oxidative stress and high levels of glutamate are known to trigger lipid peroxidation and cause cell ferroptosis[14,34]. Therefore, we speculated that Prok2 was involved in the regulation of ferroptosis. Notably, treatment of cells with a specific ferroptosis inducer, Erastin (24 h), increased Prok2 expression (Fig. 1l–n, Supplementary Fig. 4c, d). In contrast, the levels of cleaved caspase-3 were not altered during Erastin-induced ferroptosis, as demonstrated by western blots (Supplementary Fig. 4a, b) and immunostaining (Supplementary Fig. 4c–e). Interestingly, $H_2O_2$, excitotoxic glutamate and Erastin did not affect the expression of Prokr2 (Supplementary Fig. 4f–k), suggesting that ferroptosis induced increase in Prok2 levels was not driven by Prokr2-mediated regulation. Fer-1, a specific ferroptosis inhibitor, alleviated the Erastin-driven increase in *Prok2* mRNA levels and lactate dehydrogenase (LDH) release (Supplementary Fig. 5a–c). This was confirmed by immunostaining which also demonstrated that cleaved caspase-3 levels were not changed after Erastin treatment (Supplementary Fig. 5d–f). Fer-1 also prevented elevation of *Prok2* mRNA levels and cell death triggered by glutamate (Supplementary Fig. 5g–i). In contrast, Fer-1 did not affect the $H_2O_2$-induced enhancement of *Prok2* mRNA levels, but reduced LDH release (Supplementary Fig. 5j–l). Combined, these data are compatible with the involvement of Prok2 in regulation of ferroptosis in primary neurons.

**Upregulation of Prok2 decreases Erastin- or stretch-induced neuronal cytotoxicity and lipid peroxidation**. We further explored the role of Prok2 as a regulator of ferroptosis in primary neuronal cells. We used lentivirus containing Prok2 (Flagged) to increase *Prok2* mRNA levels and shProk2 to knock down *Prok2*. One out of three *Prok2*-interfering sequences tested was found to decrease *Prok2* mRNA expression most efficiently (Fig. 2a, b).

Terminal deoxynucleotidyl transferase dUTP nick-end labeling (TUNEL) assay indicated that Lv-Prok2 inhibited Erastin-induced cell damage, whereas shProk2 aggravated this effect. Fer-1 inhibited ferroptosis and prevented injury induced by Prok2 down-regulation (Fig. 2c, d). Stretch-induced cell death was also alleviated by Prok2 overexpression. Lowering Prok2 expression enhanced cell death after stretch, and this effect was inhibited by Fer-1 administration (Supplementary Fig. 6a, b). In primary neurons incubated for 24 h with Erastin, disruption and thinning of the neurites with large vacuoles and bright spots as well as decreased cytoplasm were observed. Cellular and neurite fragments were also detectable in the extracellular compartment. Lv-Prok2 administration resulted in morphologically distinct protection against Erastin-induced injury of primary cortical neurons (Fig. 2e). Both Erastin administration or stretch injury caused aggregation of $Fe3^+$ in the neurons, and this effect was suppressed by Prok2 overexpression. Low levels of Prok2 were associated with the increased contents of iron, preventable by Fer-1 treatment (Fig. 2f, g and Supplementary Fig. 6c, d). Exogenous Prok2 improved viability of cells treated with Erastin or exposed to stretch, as assayed by CCK-8. In contrast, shProk2-treated neuronal cells displayed the low viability, which could be recovered by Fer-1 (Fig. 2h and Supplementary Fig. 6e). Lv-Prok2 alleviated the Erastin- or stretch-induced LDH release. Downregulation of Prok2 was associated with additional cytotoxicity, which was reduced by Fer-1 (Fig. 2i and Supplementary Fig. 6f). Western blot results indicated that Erastin increased Acsl4 contents but reduced Gpx4 levels. After Erastin treatment, Lv-Prok2 decreased Acsl4 expression but increased Gpx4 expression. On the contrary, shProk2 significantly inhibited Gpx4 but promoted Acsl4 expression. Fer-1 inhibited shProk2-induced Gpx4 downregulation and Acsl4 upregulation (Fig. 2j). In contrast to Erastin, stretch did not change Gpx4 expression, but shProk2 decreased Gpx4 after stretch in a Fer-1 preventable way (Supplementary Fig. 6g). Furthermore, exogenous Prok2 reduced Erastin-induced or stretch-induced lipid peroxidation assessed by ROS generation, BODIPY 581/591 C11 oxidation, MDA content and GPX activity (Supplementary Fig. 7a–d and Supplementary Fig. 8a–e). Levels of oxygenated AA metabolites, 15-hydroxyeicosatetraenoic acid (15-HETE), and 12-hydroxyeicosatetraenoic acid (12-HETE), were significantly increased after Erastin or stretch treatment. Notably, Lv-Prok2 reduced the levels of these metabolites (Supplementary Fig. 7e, f and Supplementary Fig. 8f, g), whereas shProk2 enhanced lipid peroxidation. Inhibition of ferroptosis by Fer-1 prevented activation of lipid peroxidation.

We further examined whether Prok2 upregulation could change neuronal electrophysiological activity after exposure to Erastin. In the control group, action potentials, appearing as continued waves, were induced under appropriate stimulus (pA) (Supplementary Fig. 9a upper graph). Erastin-induced damage to

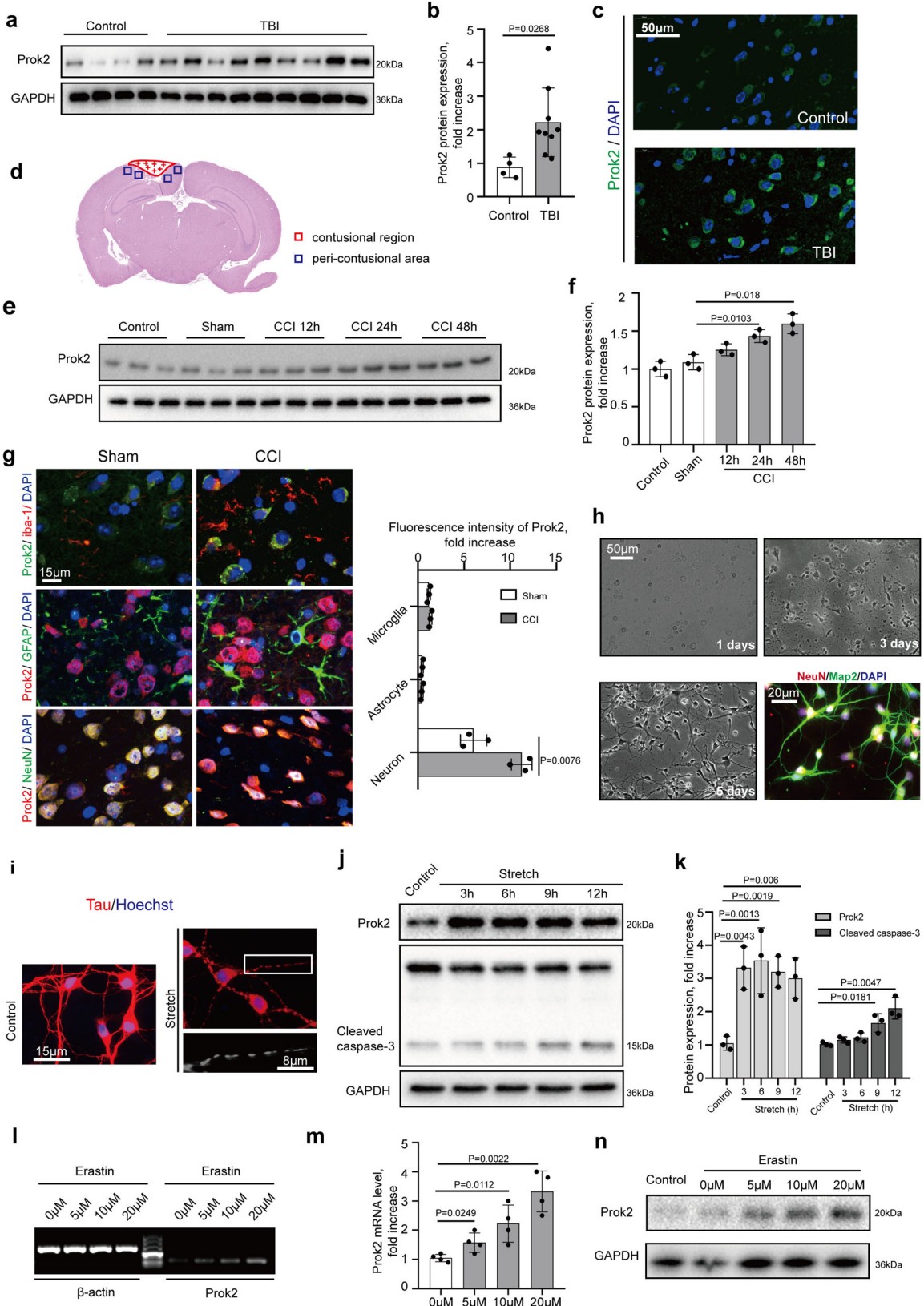

primary neurons resulted in a drastic decrease of action potentials in the patch clamp recordings. Prok2 overexpression caused alleviation of Erastin-induced damage and continued action potentials were detectable in the Erastin + Lv-Prok2 group (Supplementary Fig. 9a, b).

**Prok2-induced protection of mitochondrial functions is mediated by Acsl4.** Ferroptosis is commonly accompanied by mitochondrial injury[35,36]. We examined the morphology and distribution of mitochondria 24 h after treatment with Erastin using Mito-Tracker (Fig. 3a). In general, Mito-Tracker staining exhibited a fusiform structure, a small rod-like form in neurites,

**Fig. 1 Prok2 expression is increased after exposure to TBI, stretch, and Erastin. a, b** Western blot analysis and densitometric quantification of Prok2 expression by ImageJ in brain tissue of control ($n = 4$ samples) and TBI ($n = 9$ samples) patients. Data presented as mean ± SD. **c** Immunofluorescence assessment of Prok2 expression (green) in brain tissue from control and TBI patients. Scale bar is 50 µm. DAPI is used to label nucleus. **d** Schematic representation of the contusional region (red) and the peri-contusional area (blue) after CCI. Tissues from the peri-contusional area (blue) are collected for western blot and qRT-PCR analysis. **e, f** Western blot analysis and densitometric quantification of Prok2 expression by ImageJ in control, sham and CCI mouse brain tissue. GAPDH is used as a control. Data presented as mean±SD ($n = 3$ mice). **g** Dual immunofluorescence staining shows that Prok2 expression is most prominent in neurons (NeuN-labeled), whereas low levels of Prok2-staining are found in astrocytes (GFAP-labeled) and microglia (iba-1-labeled). Scale bar is 15 µm. Quantification of Prok2 fluorescence intensity by ImageJ is shown in the right panel. Data presented as mean±SD ($n = 3$). **h** Representative photomicrographs of NeuN (red) and Map2 (green) expressing primary cortical neurons utilized in the studies. Scale bar is 50 µm for light microscopy image and 20 µm for fluorescence image. **i** Stretch-induced neuronal injury manifests as the appearance of thin and disrupted neurites and loss of the cytoplasm. Immunofluorescence labeling of tau protein (red) is used to monitor the effects of mechanical stretch on neurites; Hoechst is used to stain cell nuclei. Scale bar is 15 and 8 µm, respectively. **j, k** Western blot analysis and quantification of Prok2 and cleaved caspase-3 expression by ImageJ in control and stretch groups. GAPDH is used as loading control. Data presented as mean ± SD ($n = 3$ experiments). **l, m** Detection and quantification of *Prok2* mRNA in primary cortical neurons exposed to Erastin for 24 h at different concentrations. β-actin is used as control. Data presented as mean ± SD ($n = 3$ experiments). **n** Erastin exposure increases Prok2 protein expression. GAPDH is used as control in western blot assays. For all panels, *n* indicates biologically independent repeats. *P* value was determined by a two-tailed unpaired Student's *t* test for comparations between two groups. Source data are provided as a Source Data file.

and an interconnected network in the cytoplasm, which were the dominant morphologies in the normal primary neurons. Upon Erastin treatment, the mitochondrial network looked disrupted and mitochondria appeared as small fragmented punctiform structures and small circles. Injury and disruption of neurites were accompanied by a decline in mitochondrial length as well. Prok2 overexpression attenuated Erastin-induced changes of mitochondrial morphology (Fig. 3b, c). Dual immunofluorescence labeling using Prok2 and Tomm20 indicated that Prok2 protected mitochondria and promoted their migration to neurites (Fig. 3d), which was also observed in the stretch model (Supplementary Fig. 10a). Transmission electron microscopy (TEM) studies revealed shrunken mitochondria, outer mitochondrial membrane (OMM) rupture, and the formation of light vacuoles likely related to the mitochondrial collapse in Erastin-treated cells, which are the characteristic changes in ferroptosis[17,37]; Prok2 overexpression prevented the appearance of these changes (Fig. 3e).

We observed that Acsl4 expression was decreased and Gpx4 expression was increased in ferroptotic cells overexpressing Prok2 however the underlying mechanism(s) of these effects remained unclear. In the CCI mouse model, injury increased Acsl4 levels but Gpx4 levels remained stable (Supplementary Fig. 11a–c), suggesting differences in regulation of Acsl4 and Gpx4 in this model vs. Erastin. And earlier studies established that Gpx4 expression was higher in *Acsl4* KO (*Acsl4*−/−) cells than in WT (*Acsl4*+/+) cells[17]. Therefore, we focused our efforts on exploring the role of Acsl4 as a possible mechanism of Prok2-mediated control of mitochondrial metabolism and ferroptosis.

Previous studies have shown that Acsl4 KO cells are resistant to RSL3-induced OMM rupture and lipid peroxidation compared to WT cells[17]. We chose to extend this work by using shCtrl and shAcsl4 in primary cortical neurons treated with Erastin or exposed to stretch. In Erastin treated or stretch exposed cells, we found a compensatory increase of Prok2. But overexpression of Prok2 did not affect Gpx4, Tomm20, Tfam, ATP and MT-ND1 levels in shAcsl4 cells (Fig. 3f–m (the left part) and Supplementary Fig. 10b–h). In shCtrl cells, Erastin-induced ferroptosis was inhibited by Lv-Prok2 treatment, associated with decreased Acsl4 and increased Gpx4 expression (Fig. 3f–i (the right part)). Gpx4 levels were not changed by a stretch but were elevated by Prok2 (Supplementary Fig. 10d). In addition, elevated levels of Tomm20, Tfam, MT-ND1, and ATP were found in shCtrl cells (Fig. 3j–m (the right part) and Supplementary Fig. 10e–h). No characteristic for ferroptosis morphological changes of mitochondria was observed in shAcsl4 primary neurons after Erastin

treatment (Fig. 3n). While stretch-induced mild mitochondrial swelling was detected by TEM, ferroptosis-related characteristic morphological changes were not observed (Supplementary Fig. 10i). These results suggest that Prok2-induced protection of mitochondrial functions is mediated by Acsl4.

**Prok2 promotes Acsl4 ubiquitination and degradation.** Assuming that Prok2-mediated decreased Acsl4 content was an important contributor to suppress neuronal ferroptosis, we explored possible mechanisms by which Prok2 downregulated Acsl4 in vitro. We assessed *Acsl4* mRNA stability in vector- and Lv-Prok2 cells treated with Erastin by blocking mRNA synthesis with a transcription inhibitor, actinomycin D. Prok2 overexpression had no effect on the steady-state levels of *Acsl4* mRNA (Fig. 4a). We next explored whether Prok2-mediated decrease of Acsl4 levels occurred via the protein degradation pathway. When CHX, a protein synthesis inhibitor, was added to vector- and Lv-Prok2 cells at different time points, western blot analysis revealed a higher Acsl4 degradation rate in Lv-Prok2 cells (Fig. 4b, c). Considering two major mechanisms of protein degradation—the autophagy-lysosomal pathway and the ubiquitin-proteasomal pathway—we pretreated primary cortical neurons with bafilomycin A1 (a specific lysosomal inhibitor) and bortezomib (a proteasomal inhibitor), either separately or in combination, and subsequently exposed them to Lv-Prok2 or vector. The CHX was used as a positive control in these experiments. Bortezomib alone or in combination with bafilomycin A1 largely blocked the Prok2-mediated degradation of Acsl4. On the contrary, treatment of cells with bafilomycin A1 alone had no effect (Fig. 4d, e). These data suggested that Prok2 primarily used the ubiquitin–proteasomal pathway for Acsl4 degradation. We co-transfected primary cortical neurons with a plasmid expressing Flagged-Acsl4 and a plasmid expressing an HA-Ubiquitin (Ub). In parallel, cells were transfected with empty vectors as negative controls. Whole-cell lysates were subjected to IP with anti-Flag or anti-HA conjugated to agarose beads. The anti-Flag and anti-HA immunoprecipitates were subjected to SDS–PAGE followed by visualization of blots with anti-HA, anti-Acsl4, and anti-Flag antibodies. Ubiquitinated species of Acsl4 were detected (Fig. 4f–h). Next, we used MG132 to examine the ubiquitination status of endogenous Acsl4 in vector-treated and Lv-Prok2-treated primary cortical neurons triggered to ferroptosis by Erastin. Based on the detection of Acsl4 and Ub in total lysates of cells treated with Erastin or exposed to stretch, we showed that Prok2 promoted Acsl4 ubiquitination/degradation,

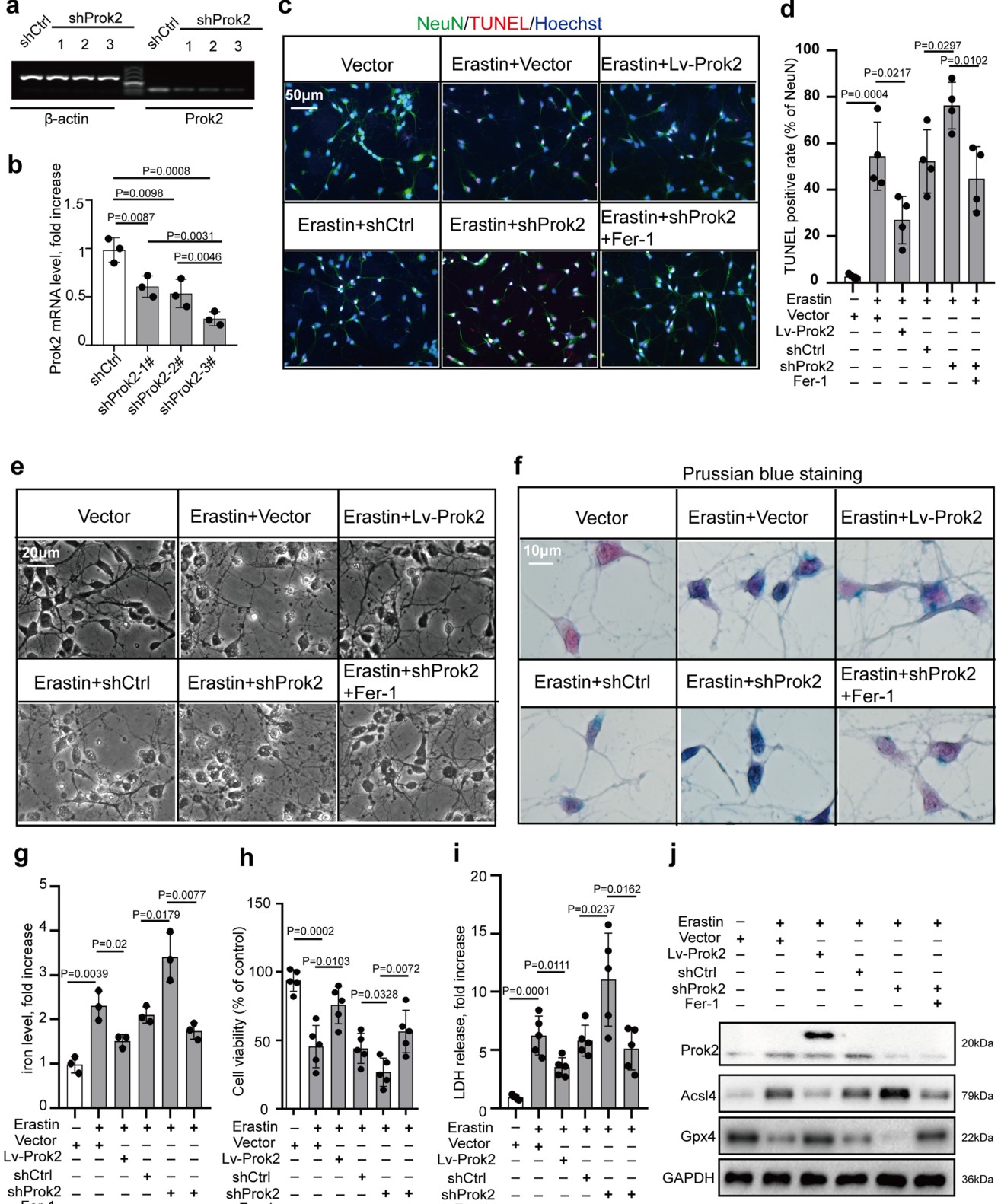

and this effect was abolished by MG132 (Fig. 4i and Supplementary Fig. 12a). Equal amounts of whole-cell lysate were subjected to IP with an anti-Acsl4 or -IgG, followed by western blotting using anti-Acsl4 or anti-Ub. Importantly, after Acsl4 IP, the amount of pulled-down non-ubiquitinated Acsl4 was lower in the precipitates of the Lv-Prok2 plus MG132 samples than in those of the vector-plus MG132 sample (Fig. 4j, compare lanes 3 vs. 2). However, the Lv-Prok2 plus MG132 cells clearly had a higher amount of polyubiquitinated Acsl4 than the vector-plus MG132 samples. In contrast, western blotting with anti-Acsl4 or anti-Ub did not reveal specific bands in the IgG immunoprecipitates. Prok2 upregulation also led to higher amounts of Acsl4 ubiquitination in the stretch model (Supplementary Fig. 12b). Altogether, these results provide

**Fig. 2 Overexpression of Prok2 attenuates Erastin-induced cytotoxicity.** Primary cortical neurons are treated with 20 μM Erastin for 24 h, lentivirus containing Prok2 (Lv-Prok2) or control (vector), control RNAi (shCtrl) or RNAi against Prok2 (shProk2), and ferrostatin-1 (Fer-1). **a** shProk2 decreases *Prok2* mRNA levels vs. shCtrl. One of the three Prok2-interfereing shRNAs, the sh-Prok2-3#, is found to decrease Prok2 mRNA expression most efficiently. **b** Quantification of *Prok2* mRNA levels by ImageJ. Data presented as mean ± SD ($n = 3$ experiments). **c**, **d** Cell death measured by TUNEL staining of primary neurons. Scale bar is 50 μm. Data presented as mean±SD ($n = 4$ experiments). **e** Representative photomicrographs of primary cortical neurons in different groups. Characteristic of neuronal injury are disruption and thinning of neurites with large vacuoles and bright spots as well as decreased cytoplasm. Cellular and neurite fragments are also observed in extracellular compartment. Scale bar is 20 μm. **f**, **g** Prussian blue staining shows that Erastin stimulates the formation of $Fe^{3+}$ (blue) in neurons. Lv-Prok2 suppresses this effect. Scale bar is 10 μm. The iron levels are determined based on the color intensities measured by ImageJ. Data presented as mean±SD ($n = 3$ independent experiments). **h**, **i** Cell viability analyzed by CCK-8. Cytotoxicity induced by Erastin is measured by LDH assay. Data presented as mean ± SD ($n = 5$ experiments). **j** Expression of Prok2, Gpx4 and Acsl4 in primary neurons under various treatment conditions. GAPDH is used as control in western blot assays. For all panels, *n* indicates biologically independent repeats. *P* value was determined by two-tailed unpaired Student's *t* test for comparations between two groups. Source data are provided as a Source Data file.

direct evidence that Prok2 overexpression enhanced Acsl4 ubiquitination and possibly channeled Acsl4 toward its proteasomal degradation.

**Fbxo10 is crucial for Prok2-induced Acsl4 ubiquitination/ degradation.** To directly examine the role of E3 ubiquitin ligase in the control of Acsl4 ubiquitination and degradation, we examined the proteins pulled down after IP of neuronal lysates with Acsl4 antibody using silver staining followed by LC–MS (Fig. 5a, b). Fbxo10 was the only ubiquitin ligase identified (Fig. 5b). Immunofluorescence staining confirmed that Acsl4 and Fbxo10 are co-localized in the mitochondria (Fig. 5c). Erastin caused upregulation of Acsl4 and a decrease of Fbxo10 staining. The intensity of mitochondrial staining decreased upon Erastin exposure, particularly in the neurites. Transfection with Myc-amplified Fbxo10 suppressed Acsl4 expression with or without Erastin stimulation and enhanced Ub binding to Acsl4 (Fig. 5d). Similar results were observed in the stretch model (Supplementary Fig. 12c). Fbxo10 has several functional domains which can interact with different signaling proteins. To determine the binding domain of Fbxo10 for Acsl4, three Myc-tagged Fbxo10 domains (aa 6–49, aa 460–867 and the whole aa 1–951; Fig. 5e, the upper part) were synthesized and integrated into plasmid, respectively. A plasmid carrying only Myc was used as a control. Specific protein–protein interaction was detected between the Fbxo10 domain (aa 460–867) and Acsl4 (Fig. 5e the lower part).

To further investigate the relationship between Prok2 and Fbxo10, we examined the expression of Fbxo10 in cells overexpressing Prok2. Elevated Fbxo10 levels were observed after Prok2 overexpression under the normal, Erastin-stimulated or stretch conditions (Fig. 5f and Supplementary Fig. 12d). Based on these results, we hypothesized that Fbxo10 may be involved in Prok2-induced Acsl4 ubiquitination and degradation. To further test this hypothesis, we knocked down Fbxo10 expression in Prok2-overexpressing cells and examined Acsl4 expression after vehicle (DMSO) control, Erastin, or stretch exposure (Fig. 5g, h and Supplementary Fig. 12e). Fbxo10 deficiency attenuated Acsl4 ubiquitination/degradation induced by Prok2 overexpression. Thus, Prok2 overexpression alleviates ferroptosis by promoting expression of Fbxo10 and accelerating Acsl4 ubiquitination/ degradation. Of note, in contrast to the effects of Erastin treatment or stretch exposure, upregulation of Fbxo10 by Prok2 was not equally efficient in the control (DMSO) group.

**Neuroprotective effects of intracerebroventricular (ICV) injection of adeno-associated virus (AAV)-Prok2 depend on Acsl4 regulation.** To examine the validity of the proposed mechanism in vivo, we performed AAV-Prok2 and AAV-shFbxo10 transfections before conducting CCI at the time points displayed in Supplementary Fig. 13a. Flag-conjugated AAV-Prok2 was injected into the lateral ventricle of the mouse

brain. Increased Prok2 expression was observed after 7 days (Fig. 6a, b). In order to exclude the effects of injection on neuronal mitochondria, we examined Tomm20 expression and mitochondrial morphology by immunostaining and TEM, respectively, and found no significant changes (Supplementary Fig. 13b–d). In the CCI model, Prok2 upregulation by AAV-Prok2 was associated with an increase in the levels of Prokr2 (Supplementary Fig. 13e, f).

Assuming that the effects of Prok2 were realized via regulation of Acsl4, we explored whether this mechanism occurred in vivo. We downregulated Acsl4 expression by intracranial injection of AAV-shAcsl4 tagged with HA. To monitor the transfection efficiency, we used immunofluorescence microscopy (Supplementary Fig. 14a), as well as immunoblotting (Supplementary Fig. 14b, c). Acsl4 content was reduced by AAV-shAcsl4 and remained low after CCI. In the AAV-shAcsl4 group, Gpx4 levels did not change after CCI but increased in the Prok2 overexpression group (Supplementary Fig. 14b–d). We found that overexpression of Prok2 prevented CCI-induced: (i) decreases in GPX activity, as well as total GSH levels and GSH:GSSG ratio, (ii) increases in MDA levels and the number of TUNEL- and NeuN-positive cells in an Acsl4-depended manner (Supplementary Fig. 14e–j).

**Effects of AAV-Prok2 and AAV-shFbxo10 on CCI-induced ferroptosis.** We further assessed in vivo effects of AAV-Prok2 and AAV-shFbxo10 on CCI-induced ferroptosis. To this end, mice were injected with luciferase-conjugated AAV-shFbxo10 and knockdown of Fbxo10 was confirmed (Fig. 6c, d). The contribution of ferroptosis was evaluated by the treatment with Fer-1 (1 mg/kg) which was administered i.p. before CCI injury once daily until euthanasia or the Morris water maze (MWM) test. Additionally, the expression of Gpx4 and Acsl4 was examined by western blot assays within 2 days post CCI (Fig. 6e–g). We observed that Acsl4 levels were increased in the CCI group. However, Gpx4 levels did not change significantly ($P = 0.7852$). AAV-Prok2 administration increased Gpx4 expression but decreased Acsl4 levels, and this effect was blocked by Fbxo10 knockdown. Fer-1 alleviated ferroptosis, suppressing Acsl4 and elevating Gpx4 levels, regardless of Fbxo10 deficiency. Immunohistochemical staining for Acsl4 and Gpx4 showed similar results (Fig. 6k). AAV-Prok2 reduced the lesion volume assessed by MRI. Furthermore, AAV-shFbxo10 administration increased the lesion volume even in the presence of enhanced Prok2 expression, and this effect was suppressed by Fer-1 (Fig. 6h, i). Prok2 overexpression or Fer-1 treatment protected against the mitochondrial shrinkage and outer membrane rupture after CCI (Fig. 6j). TUNEL staining showed that Prok2 overexpression attenuated, while Fbxo10 deficiency enhanced CCI-induced cell death. The increase in AAV-shFbxo10-induced cell death after CCI was decreased by Fer-1 administration (Fig. 6l, m). CCI resulted in ferroptotic changes as evidenced by decreased GSH, GSH levels,

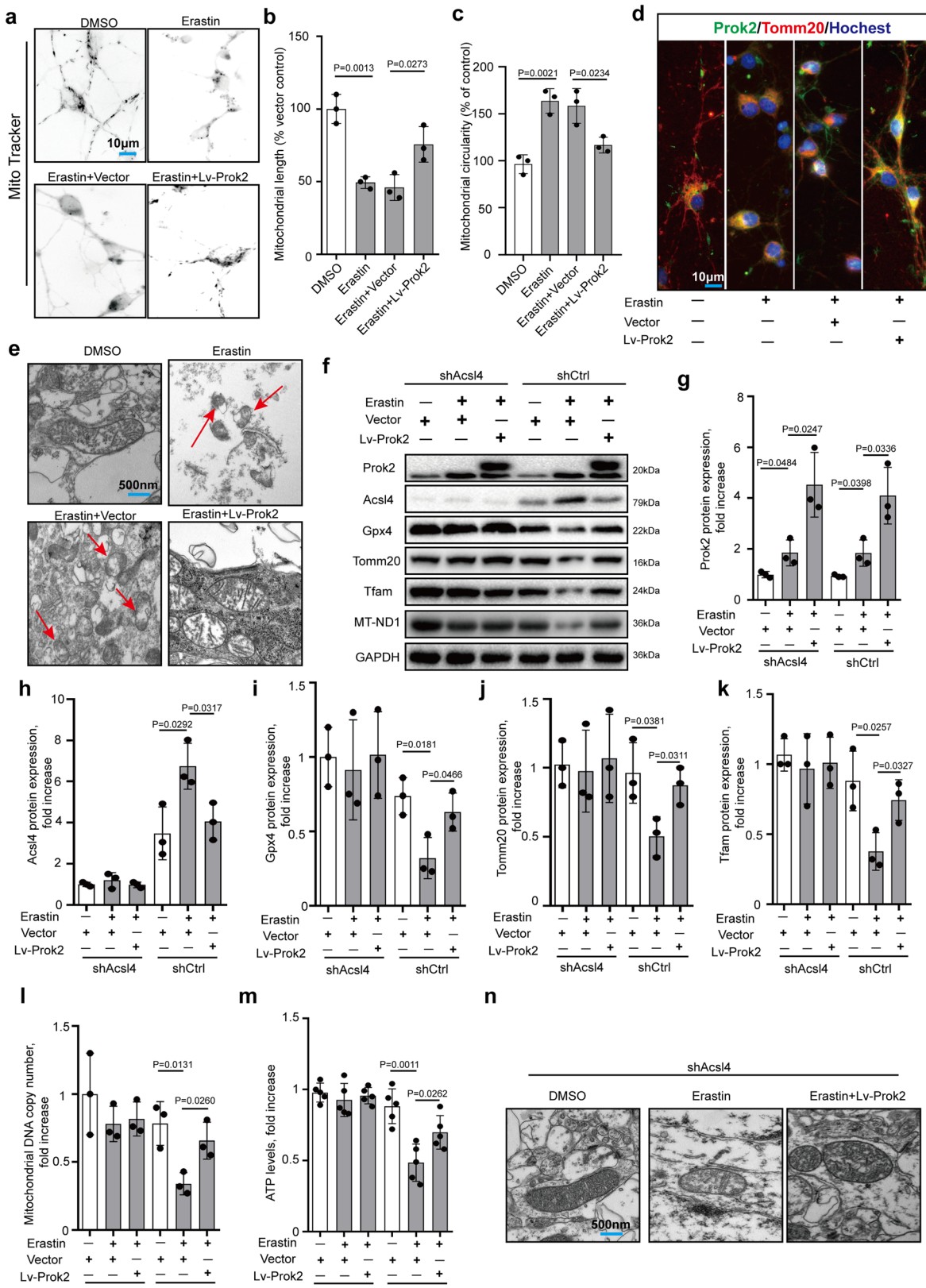

GSH:GSSG ratio, and GPX activity, and increased lipid perox-idation assessed by MDA, 12-HETE, and 15-HETE levels (Sup-plementary Fig. 15a–f). Pretreatment with AAV-Prok2 attenuated these changes in a Fbxo10 dependent manner. Detrimental effects of AAV-shFbxo10 co-treatment on GSH levels, GPx4 activity and lipid peroxidation after CCI were attenuated by Fer-1.

**Increased levels of Prok2 improve and AAV-shFbxo10 sup-presses motor ability and learning performance after CCI**. We further studied the effects of Prok2 in vivo using several neuro-cognitive tests. A schematic timeline of experiments is displayed in Fig. 7a. The rotarod test was employed to assess motor abilities 2 days after CCI. Motor activity of AAV-Prok2-injected mice

**Fig. 3 Prok2 protects mitochondrial function in an Acsl4-dependent manner.** 20 µM Erastin for 24 h is used to induce ferroptosis in these studies. **a** Mito-Tracker staining exhibits a fusiform structure, a small rod-like form in neurites, and an interconnected network in the cytoplasm in the normal primary neurons. Erastin treatment disrupts the mitochondrial network and mitochondria appear as small fragmented punctiform structures and small circles. Prok2 overexpression reduces mitochondrial circularity (**b**) and increases mitochondrial length (**c**). Data presented as mean ± SD ($n = 5$ experiments). **d** Immunofluorescence assessment of the expression and intracellular distribution of Tomm20 and Prok2. Prok2 overexpression increases Tomm20 positivity and promotes migration of mitochondria to neurites. Scale bar is 10 µm. **e** Representative electron microscopy images show shrunken mitochondria and outer membrane rupture upon exposure to Erastin (red arrow), which is inhibited by Prok2 overexpression. Scale bar is 500 nm. **f–l** Expression levels of Acsl4, Gpx4, Tfam, Tomm20, and MT-ND1 (mitochondrial DNA copy number) in Acsl4 deficient (shAcsl4) and control (shCtrl) primary neurons. GAPDH is used as control. Data presented as mean ± SD ($n = 3$ experiments). **m** ATP levels are decreased upon exposure to Erastin. Overexpression of Prok2 prevented the decrease in mtDNA copy number and ATP levels in Erastin-treated cells. Data presented as mean ± SD ($n = 5$ experiments). **n** Representative TEM images illustrating that Erastin administration does not cause marked changes of mitochondrial morphology in shAcsl4 primary neurons. Scale bar is 500 nm. For all panels, $n$ indicates biologically independent repeats. $P$ value was determined by two-tailed unpaired Student's $t$ test for comparisons between two groups. Source data are provided as a Source Data file.

exposed to CCI was markedly improved compared to the CCI-alone. This improvement was not observed in the CCI + AAV-Prok2+AAV-shFbxo10 group ($P = 0.0887$, $n = 10$ mice/group). Notably, Fer-1 effectively suppressed the harmful effects of AAV-shFbxo10 and enhanced motor abilities (Fig. 7b). The maximally tolerated rotation speed was not significantly different between the groups, with the notable exception of the sham and CCI groups (Fig. 7c). The MWM test was performed 14 days after CCI. Mice were subjected to 3 days of visible training sessions, during which the platform was on the surface of the water and indicated by black staining. Latency, distance, and swimming speed were equal in different groups of mice exposed to the treatments, suggesting that motor abilities did not interfere with the hidden training part (Fig. 7d–f). After elimination of the possibility of interferences due to motor differences, mice were trained to find a submerged platform during a 5-day hidden training session. The motion curves on the 5th day of hidden training session are shown in Fig. 7g. CCI-exposed mice spent more time and traveled longer distances to reach the platform than sham-operated mice during the training. AAV-Prok2 mice exhibited a decreased latency and distance as the training progressed. Mice injected with AAV-shFbxo10 demonstrated a significant decrease in their ability to learn the location of the submerged platform. However, Fer-1 administration significantly reduced the latency and distance required for searching for the hidden platform, in spite of the AAV-shFbxo10 administration (Fig. 7h, i). Different treatments did not cause changes in the swimming speed of mice in any of the tested groups during the hidden training sessions (Fig. 7j).

## Discussion

TBI is a mechanical injury resulting in immediate damage at the site of the impact due to direct trauma. Currently, surgery is still the first-line treatment for severe TBI in clinical practice, but there has been little progress in the development of drug-based therapies that can be used to inhibit cell death caused by the secondary injury, which leads to spreading of the lesion. This is important to improve the prognosis and the quality of life of patients with TBI. The current study identified a neuroprotective role of Prok2, a member of the Prokineticin family, in post-TBI pathophysiology. Here, we demonstrated that Prok2 expression was rapidly induced following TBI in humans as well as in mouse CCI model and in primary cortical neurons exposed to mechanical stretch and pro-ferroptotic neurotoxic agents. Our findings of the abundant expression of the Prok2 receptor, Prokr2, in neurons, were essential for developing the concept that targeting Prok2 signaling may lead to protective strategies affecting the neuronal activity. Mechanistically, we established that Prok2 can act as an anti-ferroptotic protein improving neuronal health through the regulation of iron and lipid

peroxidation. We further revealed that this protective strategy could be enhanced via downregulation of Acsl4, an important regulator of sensitivity to ferroptosis controlling the availability of polyunsaturated substrates required for the generation of ferroptotic death signals. Specifically, we discovered that Prok2 overexpression alleviated cell injury induced by Erastin and stretch injury by increasing Fbxo10 expression and promoting Acsl4 ubiquitination/degradation (Fig. 8).

Fbxo10 is a subunit of one of the large E3 ligase family and it is involved in the recognition of the Skp–Cullin1–F box (SCF) complex. Previous studies established the role of Fbxo10 in mediating ubiquitination and degradation of Bcl2 in mantle cell lymphoma[38]. Furthermore, Fbxo10 can alleviate the inflammatory response in acute lung injury by targeting Rage, a highly expressed cell membrane receptor serving to anchor lung epithelia to matrix components thus amplifying inflammatory signaling[39]. Our study discovered a potential protective role of Fbxo10 against TBI-induced ferroptosis. Using a mouse model, we demonstrated that Prok2 gene delivery with AAV-carrier effectively protected against CCI-induced brain lesion and behavioral deficits. On the contrary, inhibition of Fbxo10 by AAV-shFbxo10 administration suppressed the anti-ferroptotic effects of Prok2, which could be alleviated by a specific ferroptosis inhibitor, Fer-1. Our in vitro experiments established that Fbxo10 caused Acsl4 degradation both independently of Erastin or after Erastin treatment. However, in the control, DMSO group, only ~1.7-fold Fbxo10 upregulation was found after Prok2 overexpression. While mild Acsl4 degradation induced by Prok2 was found without Erastin treatment or stretch stimulation, Prok2 induced more robust Fbxo10 upregulation after these exposures. Therefore, Prok2 was more effective in promoting Acsl4 degradation after Erastin administration or stretch. Thus, regulators of Prok2 expression represent potential targets for the design of therapeutic approaches against CCI injury.

Ferroptosis has been proposed to play a pathogenic role in major neurodegenerative diseases, including Parkinson's disease, Alzheimer's disease, and stroke[14,40,41]. An increasing number of studies indicate that ferroptosis is involved in TBI pathogenesis as a mechanism of neuronal death. Gpx4 and Acsl4 represent two major regulators of lipid peroxidation in ferroptosis[18,42]. In a healthy cell, GSH is available as a source of reducing equivalents to effectively eliminate pro-ferroptotic hydroperoxy- phosphatidylethanolamines in a Gpx4-catalyzed reaction converting them into non-proferroptotic alcohols[14]. Depletion of GSH and/or inactivation of Gpx4 leads to the initiation and completion of the ferroptotic program. In the CCI model, we found that GPX4 activity was decreased and MDA levels were elevated within 2 days after CCI. However, Gpx4 protein expression remained unchanged. It has been reported that both Gpx4 expression and GPX4 activity were reduced after TBI in pediatric rats[43], but not in 8-week-old

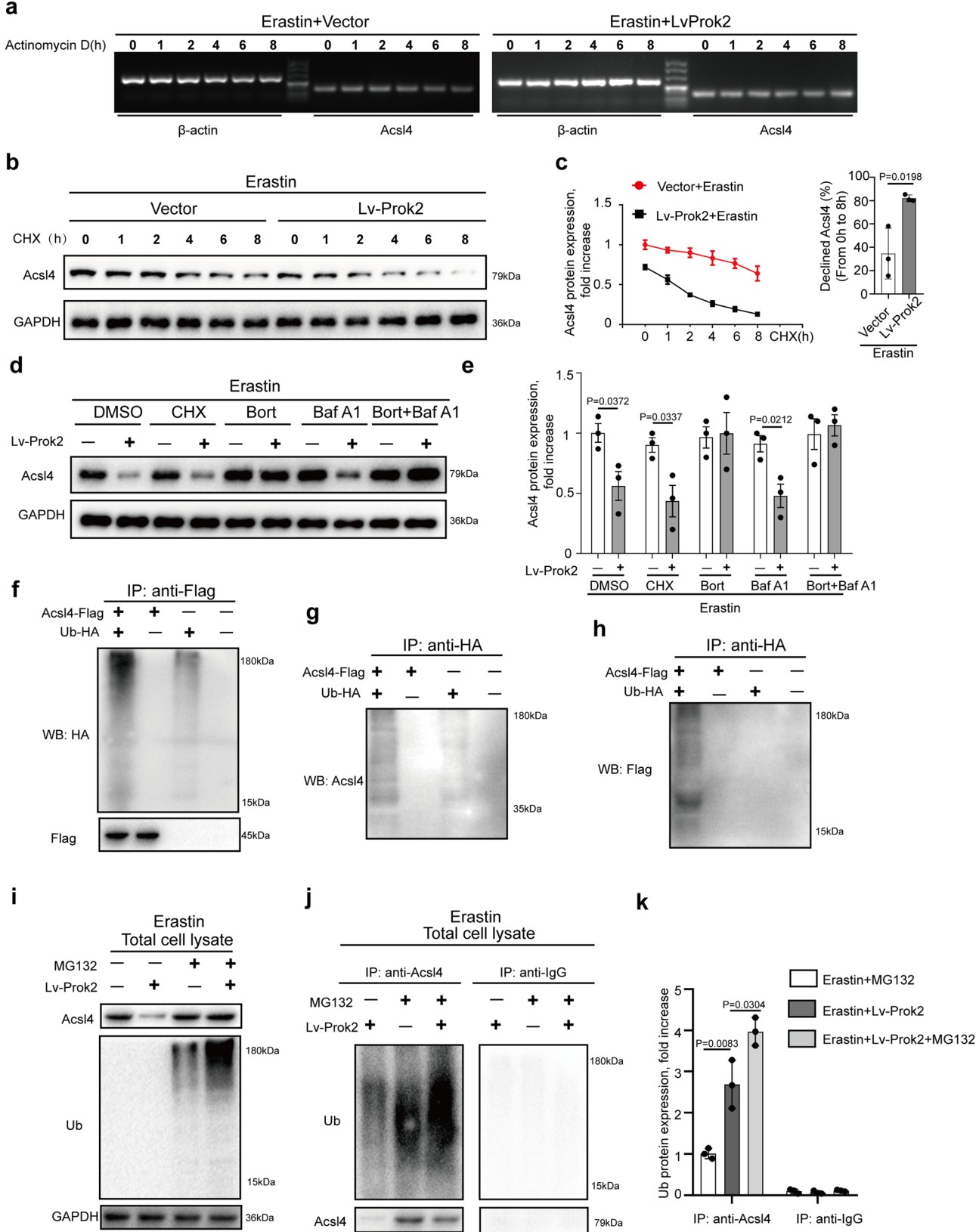

mice. This may be interpreted as an indication that stabilization of the Gpx4 protein in the CCI model might be a result of animal model differences and different ages. Several newly discovered protein regulators, such as PEBP1[43], BAP1[44], FSP1[45,46] the deubiquitinase OTUB1[47], and iPLA2 beta[48] have been associated with ferroptosis via the control of iron metabolism and lipid peroxidation. The machinery of autophagy can promote ferroptosis by the degradation of ferritin[49]. ALOX12 has been shown to participate in the p53-mediated ferroptotic pathway of tumor suppression[50]. Our study presents a neuro-ferroptosis regulatory

**Fig. 4 Prok2 promotes Acsl4 ubiquitination degradation. a** Vector and Prok2 overexpressing (Lv-Prok2) primary cortical neurons are treated with 20 μM Erastin for 24 h. Then, actinomycin D (Act D), a transcription inhibitor blocking mRNA synthesis, is added at a concentration of 6 μg/ml, and total RNA is isolated at the indicated time points for semi-qRT-PCR analysis of *Acsl4* and *β-actin*. Prok2 overexpression has no effect on the steady-state levels of *Acsl4* mRNA. **b** Vector and Prok2 (Lv-Prok2) overexpressing primary cortical neurons are treated with 20 μM Erastin for 24 h after which CHX, a protein synthesis inhibitor, is added to cells at a concentration of 5 μg/ml. Total cell lysates are isolated at the indicated times and subjected to western blotting. Bands are visualized using antibodies against Acsl4 and GAPDH. **c** The line graph (left panel) shows the expressions of Acsl4 analyzed by ImageJ and the bar graph (right panel) indicates a higher Acsl4 degradation rate observed in Lv-Prok2 cells vs. vector after Erastin treatment. Data are presented as mean values ± SD ($n = 3$ experiments). **d, e** Primary cortical neurons are treated with 5 μg/ml CHX, 200 nM bortezomib (Bort), 50 nM bafilomycin A1 (Baf A1), or a combination of Bort and Baf A1 for 1 h prior to the addition of 20 μM Erastin. Cell lysates are obtained after 24 h of Erastin administration and immunoblotted for Acsl4 and GAPDH. Bar graph shows Acsl4 expression normalized to GAPDH under different conditions. Bort alone or in combination with Baf A1 blocks Prok2-mediated degradation of Acsl4. Data are presented as mean values ± SD ($n = 3$ experiments). **f–h** Plasmids encoding Flag-tagged Acsl4 and HA-Ubiquitin (Ub) as well as empty vectors as controls are co-transfected into primary cortical neurons. Then cell lysates are immunoprecipitated with anti-Flag or anti-HA and western blot analysis is performed for Acsl4, Flag, and HA showing ubiquitinated species of Acsl4. **i** Primary cortical neurons expressing control empty vector or Lv-Prok2 are treated with 20 μM MG132 for 6 h to block proteasomal degradation and then are exposed to 20 μM Erastin for 24 h. Total lysates are analyzed for Acsl4 and ubiquitinated proteins by immunoblotting using anti-Acsl4 and anti-Ub antibodies. The decrease in Acsl4 protein observed upon Prok2 overexpression is abolished by MG132 and ubiquitination of Acsl4 is increased. **j, k** IP with Acsl4 antibody and western blot with anti-Ub show that ubiquitination of Acsl4 is higher in Prok2 overexpressing plus MG132-treated cells vs. control untreated cells as well as empty vector transfected plus MG132-treated cells. IgG is used as a negative control. Bar graph shows quantification of ubiquitinated Acsl4. Data are presented as mean values ± SD ($n = 3$ experiments). For all panels, *n* indicates biologically independent repeats. *P* value was determined by two-tailed unpaired Student's *t* test for comparations between two groups. Source data are provided as a Source Data file.

mechanism engaging the expression of Prok2, thus identifying a potential therapeutic target for the prevention of neuronal death.

Substantial mitochondrial damage occurs during the execution of the ferroptotic death program[51,52]. We revealed that Prok2 can intervene and rescue the ferroptosis-associated mitochondrial damage. In contrast to mitophagy, Prok2 protected mitochondria by preserving the number and function of healthy mitochondria by engaging the suppression of Acsl4. Expectedly, this mechanism was inefficient in Acsl4-deficient cells.

Two G-protein-coupled receptors (GPCRs) for Prok2, prokineticin receptors 1 and 2 (Prokr1/2), are ubiquitously expressed throughout the central nervous system. Prokr1 is present in glial cells, where its interaction with Prok2 promotes angiogenesis and blood flow recovery[53]. Here we documented the presence of Prokr2 and its essential Prok2-dependent functions in neurons. Interestingly, Prok2, but not Prokr2 expression, is stimulated in neurons after CCI. Kanthasamy et al. also detected elevated contents of Prok2 in dopaminergic neurons of Parkinsonian patients compared to those of healthy humans[32]. This work additionally reported that Prok2 played an anti-inflammatory role in stimulated astrocytes[54]. These researchers identified IS20, a non-peptide Prokr1 agonist, acting as a neuron-astrocyte signaling mechanism that promoted differentiation of astrocytes into an alternative A2 protective phenotype with a possible role in the development of therapeutic strategies. Similarly, our work suggests that the neuroprotective role of Prok2 may be utilized in the design and development of neuroprotectors against acute brain injury. We found that, in addition to Prok2, a number of other genes were over-expressed in TBI patients vs. healthy controls. However, these other genes were mainly expressed not in neurons but in other types of cells. Future studies will be necessary to explore the significance of the changes observed in these other genes after TBI.

Both in vitro and in vivo pro-ferroptotic exposures to Erastin, stretch, and CCI increased Prok2 expression, suggesting its sensitivity to ferroptotic stimuli. Further, Prok2 overexpression using AAV-mediated transfection attenuated CCI-induced neuronal death and functional deficits. We speculate that neurons upregulate Prok2 expression as a protective response to ferroptotic insults albeit the amounts of the produced protein are not sufficient to prevent the damage and death. Evidently, a stronger elevation of Prok2 achieved in mice after lentiviral transfection were required to alleviate CCI-induced neuronal injury. Based on

the reported Prok2 neurotoxic effects, several studies considered Prok2 as a risk factor after neuronal cell injury[55]. This discrepancy may be due to different concentrations of Bv8, a recombinant mouse Prok2, that has been used in primary neurons (0.01 and 0.1 nM)[56] in the previous study. Protective effects were obtained with nanomolar concentrations (10–100 nM). Indeed, Prok2 overexpression by viral transfection in our study could yield neuroprotective response. Thus, it is possible that the effects of Prok2 may be context-specific and concentration-dependent.

The use of AAV vector gene therapy has been reported in patients with aromatic l-amino acid decarboxylase (AADC) deficiency[57]. However, there are limitations for applications of AAV vector gene therapy in human TBI as the occurrence cannot be predicted. Patients seek treatment only after the injury. While we did not evaluate post-treatment potential of Prok2 overexpression in preventing ferroptotic neuronal death after TBI, recent studies showed that ferroptotic neuronal death is a viable therapeutic target after injury[19,58]. And our study also uncovered a molecular mechanism that could offer a potential target to suppress ferroptotic neuronal death.

## Methods

**RNA-sequencing in human brain tissues**. Human brain tissues were obtained from the First Affiliated Hospital of Nanjing Medical University, Nanjing, China. Five TBI tissues and three non-contusive tissues (control) were included. Detailed information of the five TBI tissues was displayed in Table 1. Three control tissues were obtained during surgery of arterial aneurysm and multiple cerebrovascular diseases. Dedicated personnel were alerted in advance and waited for the samples in the operating room to rapidly place the samples into liquid nitrogen. Dry ice was used for transportation of the samples to the lab. The use of human brain tissues was approved by the Research Ethics Committee of Nanjing Medical University (Nanjing, Jiangsu, China), and experiments were performed in accordance with the approved guidelines. Informed consents were obtained from the next of kin or patients. All sequencing procedures were performed by Novagene Co. Ltd (Nanjing, China).

**Design of animal experiments**. There was no significant difference in body temperature, weight, or food intake of mice between different groups before the experiment.

*Experimental design 1*. To explore the Prok2 levels at each time point after CCI, mice were randomly assigned into the following three groups: control ($n = 6$), sham ($n = 6$), CCI ($n = 18$). The CCI group was divided into 3 subgroups ($n = 6$ for each time point): 12, 24, and 48 h after CCI. Mice were sacrificed at the planned

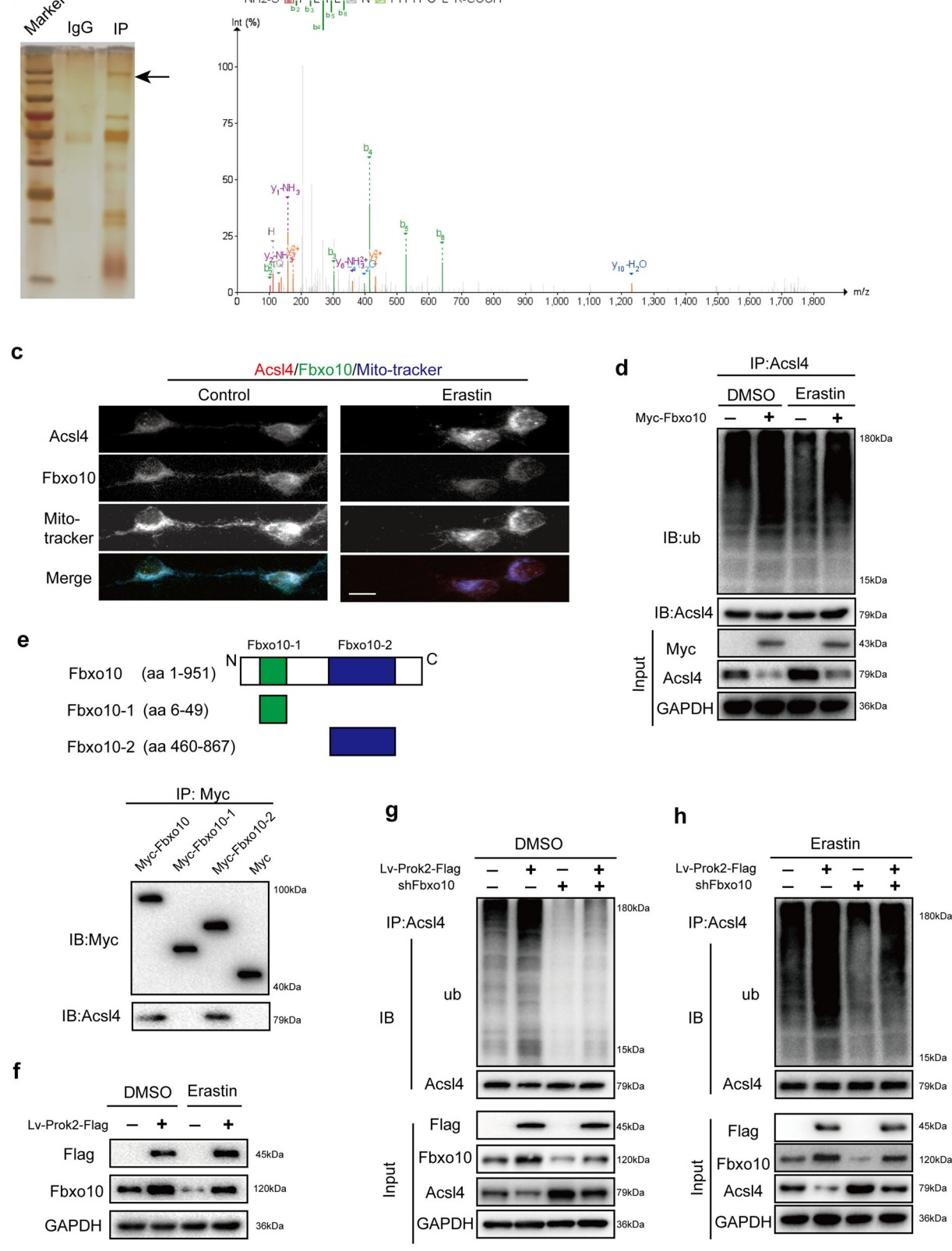

time points and tissue samples were collected for subsequent analysis such as western blot or qRT-PCR assays (Supplementary Fig. 16a).

*Experimental design 2.* To investigate whether Prok2 could reduce ferroptosis after CCI, we performed ICV injection with AAV-Prok2, AAV-shFbxo10, and AAV-shAcsl4. Five groups were designed to test for the optimal transfection time: AAV-

NC group, AAV-Prok2 group, AAV-shCtrl group, AAV-shFbxo10 group, and AAV-shAcsl4 group ($n = 3$ mice per group). And then, mice were randomly assigned into five groups: sham + AAV-NC group, CCI + AAV-NC group, CCI + AAV-Prok2 group, CCI + AAV-Prok2 + AAV-shFbxo10 group, and CCI + AAV-Prok2+AAV-shFbxo10+Fer-1 group ($n = 19$ mice per group). Then we detected the changes of cell death, ferroptotic markers, lesion volume and behavioral

**Fig. 5 Fbxo10 is crucial in Prok2-induced Acsl4 ubiquitination. a** Cell lysates obtained from primary cortical neurons are immunoprecipitated with Acsl4 antibody. Silver staining is used to reveal all proteins bound to Acsl4 antibody. Mass spectrometry analysis identified Fbxo10 as the only ubiquitin ligase in the extracted protein from primary cortical neurons. **b** Mass spectrogram of Fbxo10. **c** Immunofluorescence staining shows co-localization of Fbxo10 (green), Acsl4 (red) and Mito-tracker (blue) in primary cortical neurons. Scale bar is 10 μm. **d** IP with Acsl4 followed by western blot with anti-Ub shows that overexpression of Fbxo10 increases ubiquitination of Acsl4 in the presence or absence of Erastin. **e** Schematic representation of Fbxo10 fusion proteins (upper part). Interaction is detected between Fbxo10 domains (aa 460–867) and Acsl4 (lower part). **f** Overexpression of Prok2 in primary cortical neurons increases Fbxo10 protein levels in the presence or absence of Erastin. **g, h** Primary neurons are co-transfected with Lv-Prok2-Flag and Fbxo10 shRNA. Cell lysates are obtained and immunoprecipitated with Acsl4 antibody followed by western blot with anti-Ub. Prok2-induced ubiquitination of Acsl4 is decreased in Fbxo10-knockdown cells both in the presence and absence of Erastin. Acsl4 is used as a control in IP lysates; GAPDH is used as a control in input lysates. Source data are provided as a Source Data file.

outcomes via western blot, IF, IHC, micro-MRI, relevant kits and behavioral tests (Supplementary Fig. 16b).

**Primary cortical neuron extraction and cell culture**. Newborn mice were the source of primary cortical neurons. The newborn mice were first disinfected with 75% alcohol to avoid bacterial contamination. The mouse brain was harvested by decapitation. The skull and perichondrium were then dissected by forceps in a 4 °C pre-cooled buffer (10% horse serum and 2% penicillin and streptomycin in DMEM:F12 (1:1) (Gibco)). The brain tissue was then dissected and digested with 0.25% trypsin at 37 °C for 30 min. A strainer was used to filter and eliminate tissue fragments. And resultant solution contained various cells, which were pelleted by centrifugation at 200×g for 5 min. The supernatant was removed and cells were resuspended in a dish or in wells containing DMEM: F12 (1:1) supplemented with 10% horse serum and 2% penicillin and streptomycin (Gibco). The six-well plate contained 7 × 100,000 cells per well. At about 5 h after seeding, culture solution was removed and replaced with the neurobasal medium containing 2% B27 and 0.5 mM glutamate. Cells were kept at 37 °C with 5% $CO_2$.

**In vitro TBI model**. In vitro TBI was performed using an established model of mechanical stretch injury[11]. Briefly, primary cortical neurons were seeded (7 × 100,000 cells/well) on silicone membranes within custom-made stainless-steel wells 5 days before the stretch exposure. Wells were fitted into the stretch apparatus, and a severe stretch injury (strain rate, 10/s; membrane deformation, 50%; peak pressure, 3–4 psi) was applied to simulate a strain field similar to that in our in vivo TBI model.

**CCI model and drug administration**. All male C57BL/6 mice (Animal Core Facility of Nanjing Medical University, Nanjing, China) were kept in an SPF condition with a 12 h light/12 h dark circle, 20–25 °C ambient temperature and 40–70% humidity. Animal experiments were approved by the committee on Animal Care of Nanjing Medical University (IACUC-1706009) and performed in accordance with institutional guidelines. Eight-week-old male mice were subjected to severe CCI[59]. Anesthesia was induced with 3% isoflurane in nitrous oxide: oxygen (7:3) and maintained with 1.5% isoflurane via nose cone. Temperature was maintained at 37 ± 0.5 °C using a heating blanket. After anesthesia, mice were placed in a stereotaxic frame (R.W.D. Shenzhen, China). A 4-mm-diameter craniotomy was performed over the left parietal bone and the bone flap was removed for trauma. A vertically directed CCI was applied (6.0 ± 0.2 m/s, 50 ms dwell time, 1.4 mm depth) using an impactor (R.W.D., Shenzhen, China). After the injury, the skin incision was closed. Mice were monitored with supplemental oxygen (100%) for 1 h before returning to their cages. Fer-1 (1 mg/kg per day) was given i.p. at 7 days before CCI and once daily until euthanasia or the MWM test. Before the brain tissues were obtained, mice were perfused intracardially with 4 °C phosphate-buffer saline (PBS) solution.

**Lentiviral and plasmid transfection**. Prok2 mRNA sequence was obtained from NCBI: ATGGGGGACCCGCGCTGTGCCCCGCTACTGCTACTTCTGCTGCTA CCGCTGCTGTTCACACCGCCCGCCGGGGATGCCGCGGTCATCACCGGGG CTTGCGACAAGGACTCTCAGTGCGGAGGAGGCATGTGCTGTGCTGTCAG TATCTGGGTTAAGAGCATAAGGATCTGCACACCTATGGGCCAAGTGGGC GACAGCTGCCACCCCCTGACTCGGAAAAGTCATGTTGCAAATGGAAGGC AGGAAAGAAGAAAGGGCGAAGAGAAGAAAGAGGGAAGAAGGAGGTTCCAT TTTGGGGGCGGAGGATGCACCACACCTGCCCCTGCCTGCCAGGCTTGGC GTGTTTAAGGACTTCTTTCAACCGGTTTATTTGCTTGGCCCGGAAATGA. Lentiviruses carrying Prok2 (pLV-Ef1a-Prok2-3Flag-Puro) or vectors (pLV-Ef1a-3Flag-Puro) were obtained commercially from Genepharma (Shanghai, China).

The lentivirus-based short hairpin RNA (shRNA) targeting Prok2, LV2-pGLV-u6-shProk2-puro, with the Prok2 target sequence was purchased from Genepharma (Shanghai, China). 1#. 5′–3′ GCUGUCAGUAUCUGGGUUAUU; 2#. 5′–3′ GCGU GUUUAAGGACUUCUUUU; 3#. 5′–3′ GCUACUGCUACUUCUGCUGUU. As for the sh-Fbxo10, LV2-pGLV-u6-shFbxo10-puro, the target sequence was as follows: 5′-GGATCCTTCGCGGGACGTCCT-3′. For the sh-Acsl4, LV2-pGLV-u6-shAcsl4-puro, the target sequence was as follows: 5′-GAGGCUUCCUAUCUGAU

UATT-3′. To make lentivirus, the lenti- shProk2, Acsl4 or Fbxo10 plasmid and control plasmid were transfected into 293T cells using the Mission Lentiviral Packaging Mix from Sigma-Aldrich according to the manufacturer's instructions. The lentivirus was harvested at 48 h post transfection and titers were measured using the Lenti-X p24 Rapid Titer Kit (Clontech).

For stable lentiviral transfection in primary neurons, cells were grown in six-well plates in neurobasal media, and lentivirus was added to the media at a MOI of 100. After 72 h, fresh media supplemented with puromycin (50 μg/ml) was added to the cells for stable cell selection. The effects of gene overexpression or interference were verified using qRT-PCR and western blot assays. The Fbxo10-overexpression or special domain plasmids carrying Myc were generated and purchased from Genepharma (Shanghai, China). All plasmids were transfected into cells using Lipofectamine 2000 Transfection Reagent (Invitrogen) according to the manufacturer's instructions.

**TUNEL assay**. A TUNEL assay (C1089, Beyotime, China) was used to test cell death according to the manufacturer's instructions. In brief, cells or 12-μm tissue sections were prepared and fixed in 4% paraformaldehyde (PFA). And then they were incubated in PBS with 0.3% Triton X-100 for 1 h. TUNEL detection solution was incubated in the dark (37 °C) for 1 h and DAPI was used to visualize cell nuclei. Images were obtained using a Nikon Eclipse E600 microscope (Nikon, Melville, NY).

**Western blot assay**. Proteins were extracted from tissues or cells in protein lysis buffer (Keygen Biotech, Nanjing, China). The proteins were separated by 8%, 10% or 12% SDS–PAGE and then transferred to PVDF membranes (Merck Millipore). The membranes were blocked in 5% skim milk for 2 h and then membranes were incubated overnight at 4 °C with antibodies against Prok2 (Abcam, ab76747, rabbit polyclonal, 1:1000), NeuN (Abcam, ab279296, mouse monoclonal, 1:1000), Acsl4 (Santa Cruz, sc-365230, mouse monoclonal, 1:400), Gpx4 (Santa Cruz, sc-166570, mouse monoclonal, 1:400), Tomm20 (Abcam, ab56783, mouse monoclonal, 1:1000), Tfam (Abcam, ab131607, rabbit polyclonal, 1:1000), MT-ND1(Abcam, ab181848, rabbit monoclonal, 1:1000), GAPDH (YI FEI XUE BIOTECH, YFMA0037, mouse monoclonal, 1:1000), Flag (Beyotime, China, AF519, mouse monoclonal, 1:1000), HA (Beyotime, China, AH158, mouse monoclonal, 1:1000), Fbxo10 (Novus, NBP1-91889, rabbit polyclonal, 1:1000) followed by incubation with horseradish peroxidase-conjugated secondary antibody (Beyotime, China, A0208, A0216, 1:5000) for 2 h. After washing with PBST, protein bands were visualized using SuperSignal® Maximum Sensitivity Substrate (Thermo Fisher Scientific). Samples derive from the same experiment and blots are processed in parallel. Uncropped and unprocessed scans of blots are included in a Source Data file.

**RNA isolation, semi-quantitative reverse transcription (RT) PCR, and quantitative real-time RT-PCR**. Total RNA from cells was extracted using TRIzol Reagent (Life Technologies) following the manufacturer's protocol. Semi-quantitative RT PCR was performed according to previous study[60]. For semi-quantitative RT PCR, 400 ng RNA from each sample was used for cDNA synthesis, which was carried out using the RT reagents (TaKaRa RNA PCR Kit (AMV) Ver.3.0, China) according to the manufacturer's instructions. Polymerase chain reaction conditions are as follows: 94 °C, 3 min; 94 °C, 30 s; 60 °C, 30 s; 72 °C, 35 s, and an additional 35 cycles (for *Prok2*), 30 cycles (for *Acsl4*) or 24 cycles (for *β-actin*); 72 °C, 7 min. RT-PCR products were analyzed by 1 % agarose gel electrophoresis and visualized with ethidium bromide under UV light. For the quantitative real-time RT-PCR, 500 ng RNA from each sample was used for cDNA synthesis, which was carried out using the RT reagents (TaKaRa, Dalian, China) according to the manufacturer's instructions. The cDNAs were amplified by qRT-PCR using TB Green™ Premix Ex Taq™ II (Takara, Dalian, China) on a StepOnePlus Real-Time PCR System, Data were analyzed using the $2^{-\Delta\Delta Ct}$ method, and *BETA-ACTIN* RNA was used as endogenous control. All primers used were listed in Supplementary Table 1.

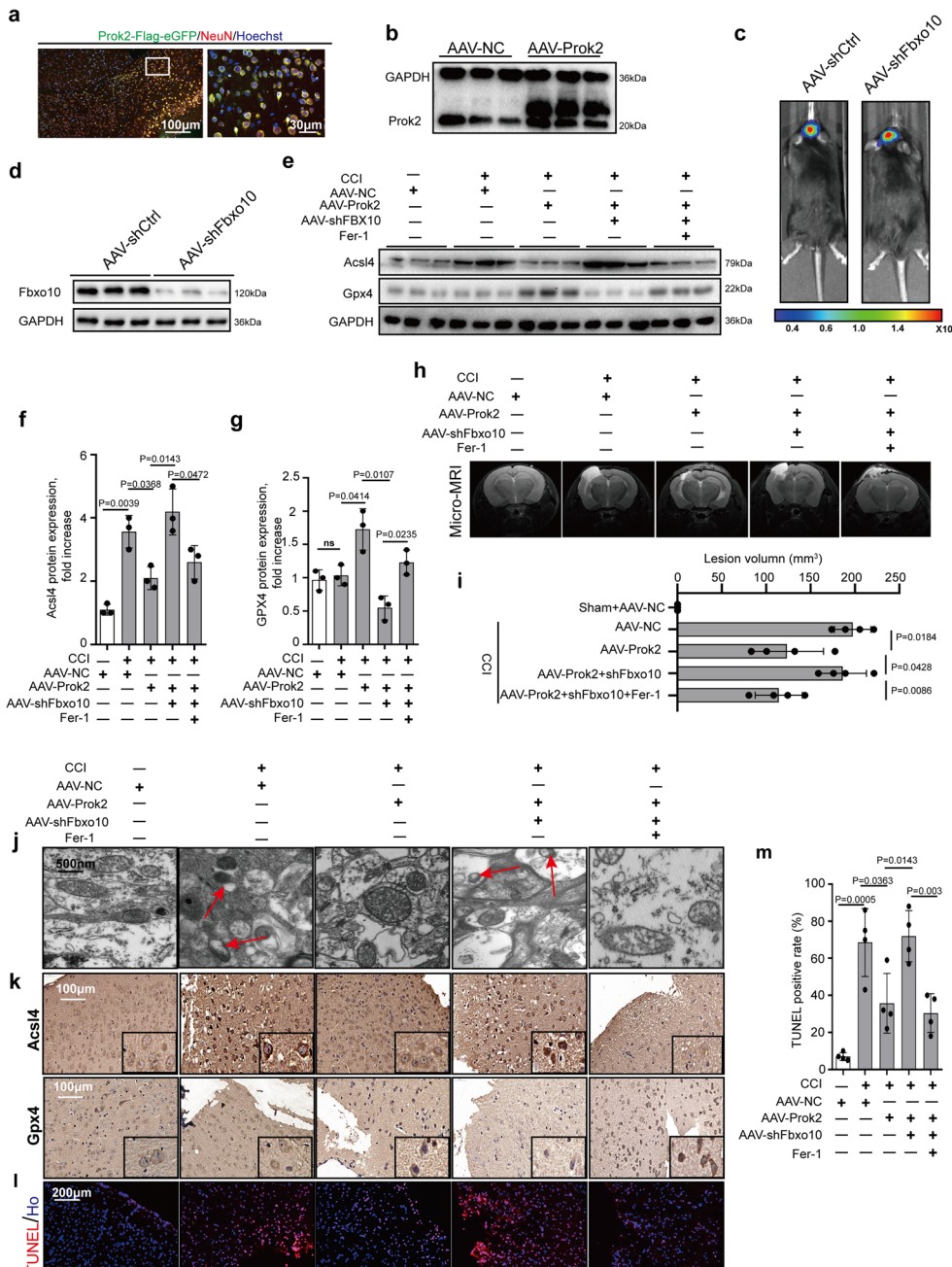

**Fig. 6 AAV-Prok2 intracerebroventricular injection (i.c.v) decreases CCI-induced lesion volume in a Fbxo10-dependent way. a** GFP-tagged Prok2-AAV is injected into mouse brain at 1 week before CCI. Dual-labeled immunofluorescence staining with Prok2-eGFP (green) and NeuN (red) is used to test the efficiency of AAV transfection in neurons. Scale bar is 100 μm (left) and 30 μm (right). **b** Increased brain tissue Prok2 expression is detected by western blot at 7 days after i.c.v. injection of GFP-tagged Prok2-AAV. GAPDH is used as control. **c, d** AAV-shFbxo10 carrying luciferase is injected into mouse brain tissue. Cri Maestro In-vivo Imaging Systems is used to screen for successful transfection. Fbxo10 knockdown in brain tissue is confirmed by western blot assays. GAPDH is used as control. **e–g** 2 days after CCI, protein expression of Gpx4 and Acsl4 proteins is examined by western blot analysis. Fer-1(1 mg/kg per day) is given i.p. once daily for 7 days before CCI and continued until euthanasia. While Acsl4 levels increases. Gpx4 levels do not change after CCI. AAV-Prok2 administration increases Gpx4 but decreases Acsl4 expression, which is blocked by Fbxo10 knockdown after CCI. Fer-1 administration suppresses CCI-induced increases in Acsl4 levels and alleviates AAV-shFbxo10-induced decrease in Gpx4 expression. Data are presented as mean values ± SD (*n* = 3 mice per group). **h** and **i** Representative T2 weighted MR images showing lesion volume in mouse brain after CCI in different experimental groups. While AAV-Prok2 transfection reduces lesion volume, co-transfection of AAV-Prok2 and AAV-shFbxo10 abolishes this effect. Administration of Fer-1 on the other hand decreases lesion volume in CCI mice expressing AAV-Prok2 and AAV-shFbxo10. Data are presented as mean values ± SD (*n* = 4 mice per group). **j** Mitochondrial morphology under different conditions is evaluated by electron microscopy. CCI-induced shrunken mitochondria and rupture of OMM, ferroptosis-related morphological changes of mitochondria (red arrow), are prevented by AAV-Prok2. **k** Immunostaining is used to examine the spatial distribution Gpx4 and Acsl4 expression in pericontusional area and shows similar treatment effect that is seen in western blot analysis observed in panels **f**, **g**. **l**, **m** Cell death response in the pericontusional area is quantified using TUNEL. Data are presented as mean values ± SD (*n* = 4 mice per group). For all panels, *n* indicates biologically independent repeats. *P* value was determined by two-tailed unpaired Student's *t* test for comparisons between two groups. Source data are provided as a Source Data file.

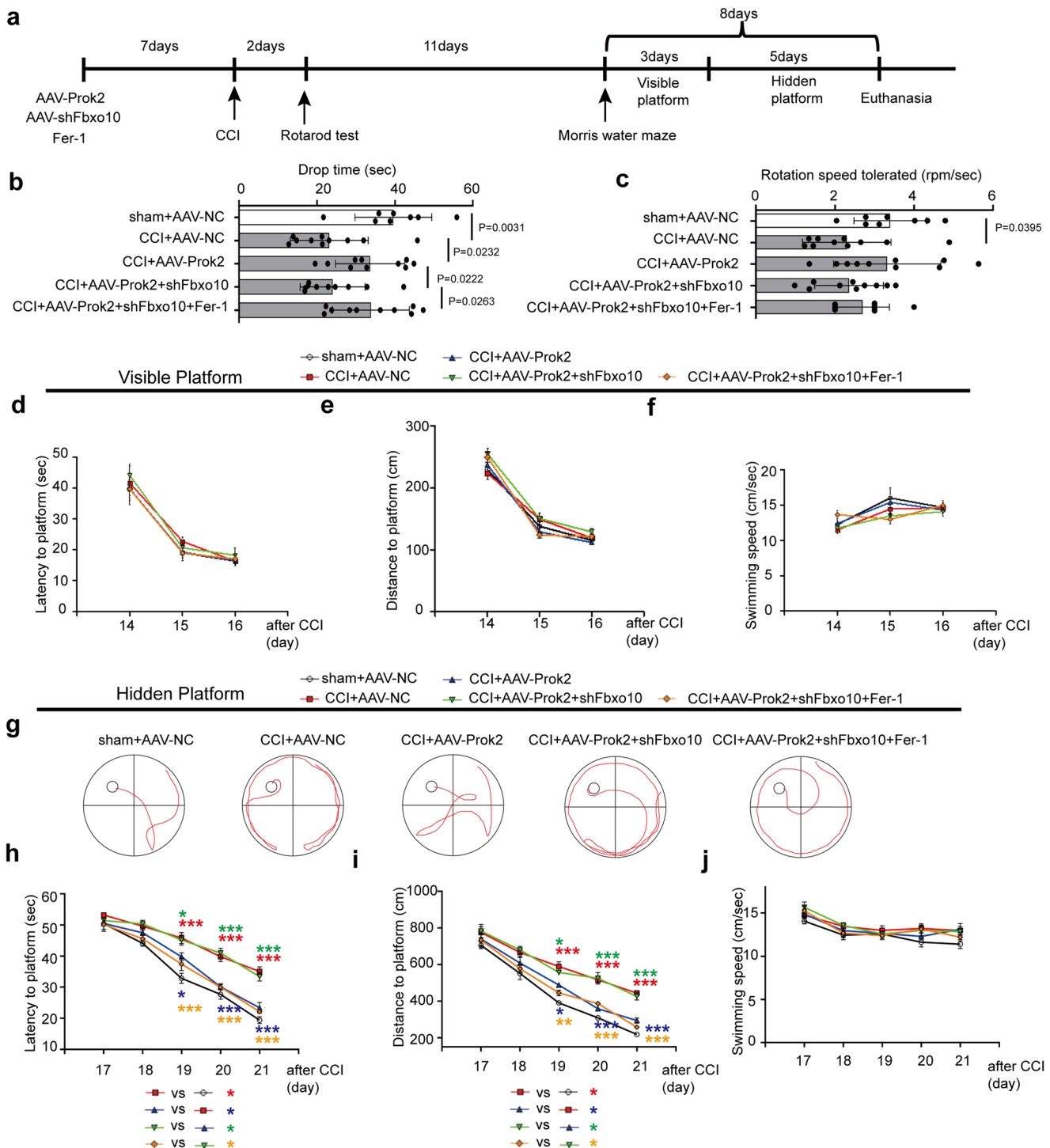

**Prussian blue staining assay**. Cellular iron accumulation was detected by Prussian Blue Stain (60533ES20, Yeasen, China) according to the manufacturer's instructions. Cells were washed three times by ddH$_2$O and fixed by PFA for 15 min. Prussian stain A (potassium ferrocyanide solution) and B (acid solution) were mixed 1:1 and added onto cells. Blue stain was visible after 30 min, and iron levels were evaluated. After that, samples were washed three times by ddH$_2$O and all cells were stained for 15 min with Nuclear Fast Red solution. Cells were imaged under a light microscope (Leica). The iron levels were determined based on the color intensities measured by ImageJ.

**ROS and lipid peroxidation assay**. The Cellular ROS Assay Kit (ab186027, Abcam) was used to quantify ROS generation in cells following the manufacturer's protocol with a fluorescence microplate reader. Excitation and emission wavelengths were 520 and 605 nm, respectively. Lipid ROS levels were measured using cell analysis reagent BODIPY 581/591 C11 (D3861, Invitrogen). Cells were

incubated with C11-BODIPY (581/591) (1 μM) for 30 min at 37 °C in an incubator before trypsinization. Subsequently, cells were resuspended in 500 μl of fresh PBS (Gibco), strained through a 35-μM cell strainer (Falcon tube with cell strainer CAP) and analyzed by flow cytometer (FACS Canto II, BD Biosciences) using the 488-nm laser for excitation. At least 10,000 cells were analyzed per sample. Data analysis was conducted using the FlowJo Software.

Lipid peroxidation was detected by the lipid peroxidation (MDA) Assay Kit (ab118970, Abcam) following the manufacturer's protocol. In brief, TBA solution was added to samples and standards, which were incubated at 95 °C for 60 min, cooled in an ice bath for 10 min, transferred to wells of the microplate and analyzed with a microplate reader.

**Cell viability and LDH assay**. Cell viability was examined by a Cell Counting Kit-8 (CCK-8, CK04, Dojindo, Tokyo, Japan) assay. In brief, cells were cultured in a 96-well plate, and after the reagent was added, the plates were incubated at 37 °C for 2 h.

**Fig. 7 AAV-Prok2 improves neurobehavioral outcome after CCI. a** Schematic outlining the timeline for the neurobehavioral testing. Fer-1 (1 mg/kg) is given i.p. once daily for 7 days before CCI and continued until euthanasia or the MWM test. Motor function is evaluated using Rotarod 2 days after CCI. MWM is utilized to examine spatial memory acquisition. MWM consisted of **d–f** visible platform testing for 3 days (4 trials per day) to assess motor and visual capabilities, followed by **h–j** hidden platform testing for 5 days (4 trials per day) to assess spatial learning ability. **b** A two-tailed unpaired Student's $t$ test and one-way ANOVA plus Tukey's test revealed motor activity of inured AAV-Prok2-injected mice is improved versus CCI-alone ($P = 0.0232$). Addition of AAV-shFbxo10 abolishes this effect ($P = 0.0222$). Fer-1 attenuated the negative effect of AAV-shFbxo10 and enhanced motor function ($P = 0.0263$). Data are presented as mean values ± SD ($n = 8$ mice in sham group and 10 mice per group in other groups). **c** The rotation speed tolerated is not significantly different between groups except between sham + AAV-NC and CCI + AAV-NC. $P = 0.0395$ versus CCI + AAV-NC group. Data are presented as mean values ± SD ($n = 8$ mice in sham + AAV-NC group; $n = 10$ mice in CCI + AAV-NC group). **d–f** Latency to platform, distance to platform and swimming speed in the visible platform testing. Data are presented as mean values ± SD ($n = 8$ mice in sham + AAV-NC group; $n = 10$ mice per group in other groups). **g** Representative swimming tracks of the mice in all five groups on the 8th day of the MWM task. **h–j** During the hidden platform testing, time spent to reach the platform (**h**), swimming distance (**i**) and swimming speed (**j**) are recorded. One-way ANOVA followed by Tukey post hoc test for different groups on the same time point are carried out. Among of them, *(red) means CCI + AAV-NC group versus sham + AAV-NC group; *(blue) means CCI + AAV-Prok2 group versus CCI + AAV-NC group; *(green) means CCI + AAV-Prok2 + shFbxo10 group versus CCI + AAV-Prok2 group; * (orange) means CCI + AAV-Prok2 + shFbxo10+Fer-1 group versus CCI + AAV-Prok2 + shFbxo10 group. Mice in CCI group spend more time ($P_{(19d)} < 0.0001$, $P_{(20d)} < 0.0001$, and $P_{(21d)} < 0.0001$) and travel longer distances ($P_{(19d)} < 0.0001$, $P_{(20d)} < 0.0001$, and $P_{(21d)} < 0.0001$) to reach the platform than sham. AAV-Prok2 mice exhibit a decrease in latency ($P_{(19d)} = 0.0199$, $P_{(20d)} < 0.0001$, and $P_{(21d)} < 0.0001$) and distance ($P_{(19d)} = 0.0126$, $P_{(20d)} < 0.0001$, and $P_{(21d)} < 0.0001$) as the training progressed. Mice injected with AAV-shFbxo10 group exhibits a significant decline in the ability to learn the spatial location of the submerged platform ($P_{(19d)} = 0.0499$, $P_{(20d)} < 0.0001$ and $P_{(21d)} < 0.0001$ for latency; $P_{(19d)} = 0.0409$, $P_{(20d)} < 0.0001$, and $P_{(21d)} = 0.0004$ for distance). However, Fer-1 administration significantly reduces the latency and distance spent on searching for the hidden platform despite AAV-shFbxo10 administration ($P_{(19d)} = 0.0009$, $P_{(20d)} < 0.0001$ and $P_{(21d)} < 0.0001$ for latency; $P_{(19d)} = 0.004$, $P_{(20d)} = 0.0002$, and $P_{(21d)} < 0.0001$ for distance). Swimming speed is not different between the groups. *$P < 0.05$, **$P < 0.01$, and ***$P < 0.001$. A two-way ANOVA with repeated measures followed by Tukey post hoc test is used for the whole groups, which reveals group by day interaction effect in latency to platform ($F_{16,215} = 3.524$, $P < 0.0001$), swimming distance ($F_{16,215} = 1.998$, $P = 0.0145$) and swimming speed ($F_{16,215} = 0.6165$, $P = 0.8702$) during hidden test. For MWM analysis, data are presented as mean values ± SD ($n = 8$ mice in sham + AAV-NC group; $n = 10$ mice per group in other groups).

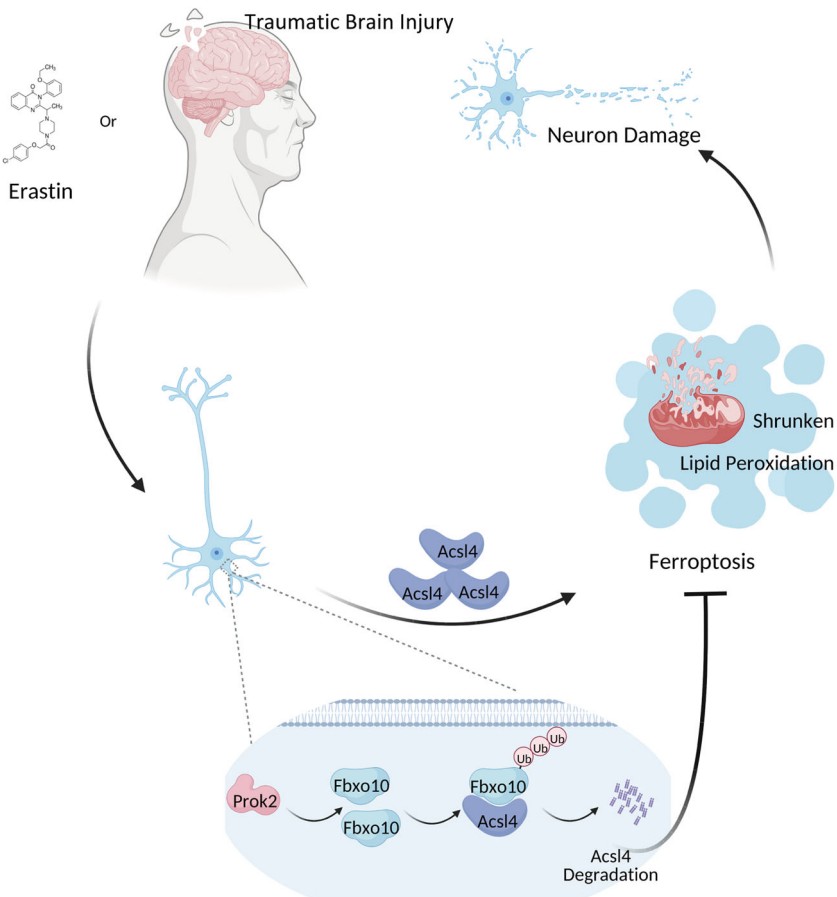

**Fig. 8 A schematic diagram showing the neuroprotective and anti-ferroptotic effects of Prok2.** TBI and Erastin in primary neurons trigger ferroptotic process, resulting in lipid peroxidation, shrunken mitochondrial morphology and neuronal damage. Overexpression of Prok2 is anti-ferroptotic and neuroprotective by upregulating the expression of Fbxo10, a E3 ubiquitin ligase that binds to Acsl4, and thereby inducing ubiquitination degradation of Acsl4.

Optical density (OD) values were measured at 450 nm by a Thermo Multiskan FC microplate photometer.

Cellular injury-induced cytotoxicity was quantified by measuring released LDH activity using the Cytotoxicity Detection Kit (C0017, Beyotime Biotech, China) according to the manufacturer's instructions.

**Mitochondrial morphology assays**. Mito-Tracker Red or Blue dye was added to each cell dish at a final concentration of 50 nM. The cells were incubated with the dye for 15–20 min at 37 °C. After incubation, cells were fixed with 4% PFA for 1 h and then washed three times with PBS. Images were obtained using a Nikon Eclipse E600 microscope (Nikon, Melville, NY). Mitochondrial area and form (length and circularity) were analyzed by ImageJ. TEM was used to analyze mitochondrial morphology. Cells were fixed in 2.5% glutaraldehyde in 0.1 M sodium cacodylate buffer pH 7.4, postfixed in 2% aqueous osmium tetroxide, dehydrated in gradual ethanol (30–100%) and propylene oxide, embedded in Epon (Merck, Darmstadt, Germany) for 24 h at 60 °C. Ultra-thin sections were cut and stained with 0.5% uranyl acetate and 3% lead citrate at 20 °C for 30 and 7 min, respectively before TEM analysis (Zeiss Libra 120 Plus, Carl Zeiss NTS GmbH, Oberkochen, Germany). Pictures were taken using a Slow Scan CCD-camera and iTEM software (Olympus Soft Imaging Solutions, Münster, Germany).

**Electrophysiology**. In brief, electrical signals were recorded using a Multiclamp 700B amplifier and a Digidata 1550 digitizer under the control of pCLAMP 10 software (all from Molecular Devices). Recording pipettes were pulled from borosilicate glass using a horizontal puller (Sutter Instrument) to give resistance ranging from 3 to 8 MU when filled with intracellular recording solution containing 135 mM K-gluconate, 3.5 mM $MgCl_2$, 0.1 mM $CaCl_2$, 4 mM Na2ATP, 5 mM EGTA, and 5 mM HEPES, at a pH of 7.2–7.4, in voltage and dynamic clamps. For the voltage clamp, voltages ranging between −60 and +20 mV in 10 mV increments with a holding potential of −60 mV were delivered to the cells. For the current clamp, cells of interest were stimulated by currents (−100 to +500 pA in 50 pA increments). Primary cortical neurons were visualized and targeted for recording by their morphology.

**GSH assays**. Cells or homogenized TBI tissues were treated with protein removal reagent and centrifuged, and the supernatant was collected for GSH analysis using a GSH and GSSG Assay Kit (S0053, Beyotime, China).

**12-HETE and 15-HETE levels**. 12/15-HETE levels were assessed using 12/15-HETE ELISA kits (ab133034/ab133035, Abcam) according to the manufacturer's instructions.

**Protein silver staining and qualitative analysis of protein**. For silver staining, gels were stained with Fast Silver Stain Kit (Beyotime Institute of Biotechnology, Shanghai, China) according to the manufacturer's instructions. Qualitative analysis of protein combined with Acsl4 was performed by the Beijing Genomics Institution (BGI). In brief, protein gel strips were obtained by gel electrophoresis. Then procedures including protein digestion, high performance liquid chromatography (HPLC) and mass spectrometer were made in sequence according to the manufacturer's instructions. In the bioinformatics pipeline, experimental MS/MS data were aligned with theoretical MS/MS data from the database to obtain protein identification results. The whole process started from converting raw MS data into a peak list and then searching matches in the database. The search results were subjected to strict filtering and quality control, and possible protein identifications were produced.

**ICV injection with AAV**. In vivo gene overexpression was achieved by AAV2 vectors. AAV serotype 2 (AAV2) vectors carrying Prok2 or GFP with a hSyn promoter (AAV2-hSyn-MCS-EGFP-3Flag-SV40-Prok2) were manufactured by GeneChem Co., Ltd (Shanghai, China). Vector No. is GV466. AAV2-hSyn-MCS-EGFP served as negative control. Adeno-associated shFbxo10 and sh-Acsl4 viruses were chemically synthesized by Genechem (Shanghai, China). The virus serotype is 2. The vectors used were AAV2-U6-MCS-Lucif-CAG-puro (shFbxo10) and AAV2-U6-MCS-HA-CAG-puro (shAcsl4). U6 is the promoter. AAV-shCtrl was generated after cloning short-hairpin RNA (shRNA) fragments into the AAV vector GV478 (Shanghai Genechem Company Limited). AAV packaging was performed by cotransfecting AAV-293 cells with the recombinant AAV vector, pAAV-RC vector, and pHelper vector. AAVs were collected from the AAV-293 cell supernatant, condensed, and purified for further animal experiments. The titer of the virus was $1 × 10^{13}$. ICV injection was performed as described previously[61]. Mice were anesthetized with 2% isoflurane. They were placed in a stereotaxic apparatus (RWD, Shenzhen, China). The stereotactic coordinates were as follows: anteroposterior (AP), −0.4 mm; mediolateral (ML), −1.0 mm; dorsoventral (DV), −3.0 mm from the bregma for injection into the left lateral ventricle. Animals were injected with 10 μl using a syringe with a 0.52 mm needle (Ito Co., Shizuoka, Japan). Virus was injected over 10 min, and the needle was left in place for 10 min prior to withdrawal. For mice that received two different AAV injections, the second injection was performed at 7 days after the first injection. During this period, mouse behavior was closely observed. Mice with increased intracranial pressure induced by ICV injection or related dysphoria were not included in the experiment. 7 days were needed for successful transfection before CCI.

**Immunostaining assay**. In line with the planned time course study, appropriate anesthesia was performed, and mice were intracardially perfused with pre-cooled PBS solution followed by pre-cooled 4% PFA. The brain tissues obtained were placed in 4% PFA for 24 h followed by dehydration in graded sucrose solution series of 10%, 20% and 30%. Sections (12 μm) were prepared to perform immunofluorescence and DAB-immunohistochemistry assays. For immunofluorescence assays, sections were permeabilized with 0.1%Triton X-100 for 15 min, and blocked with 5% bovine serum albumin in PBS at 37 °C for 1 h. Samples were incubated with the specific primary antibodies at 4 °C overnight, washed three times with PBS, and incubated with Alexa Fluor 488- or $Cy^{TM}3$-conjugated secondary antibodies (Jackson, USA, 1:500) at 37 °C for 1 h. After additional PBS washes, Hoechst nuclear staining (C1018, Beyotime, China) was performed at room temperature for 10 min. A laser scanning confocal microscope (TCS SP5II, Leica, Wetzlar, Germany) was used to acquire immunofluorescence images, and signal intensities were quantified by ImageJ. For immunohistochemistry assays, the sections were incubated with the secondary antibody (ab6112, Abcam, 1:200) for 30 min at 37 °C, rinsed with PBS, and incubated with DAB for 15 min at 37 °C. Slides were imaged under a light microscope (Leica).

Above protocols of immunofluorescence were applied to brain tissue sections. For immunofluorescence staining of cells, a different protocol was utilized. Briefly, primary neuronal cells were plated on glass slides precoated with poly-lysine (PLL). After experimental treatments, 4% PFA was used for 1 h to perform fixation of cells. The rest of the steps was the same as those for the brain tissue immunofluorescence assay.

The following primary antibodies were used to perform immunostaining: anti-Prok2 (Abcam, ab76747, rabbit polyclonal, 1:200), anti-Prokr2 (Santa Cruz, sc-365696, 1:50), anti-NeuN (Abcam, ab104224, mouse monoclonal, 1:500), anti-GFAP (Abcam, ab190288, Mouse monoclonal, 1:500), anti-iba-1 (Santa Cruz, sc-32725, mouse monoclonal, 1:50), anti-Acsl4 (Abcam, ab155282, rabbit monoclonal, 1:200), anti-Gpx4 (Abcam, ab125066, rabbit monoclonal, 1:200), anti-HA (Beyotime, China, AH158, Mouse monoclonal, 1:200), and anti-Tomm20 (Abcam, ab56783, mouse monoclonal, 1:500).

**Co-immunoprecipitation**. Co-IP was performed as described previously[62]. In brief, total cell lysates from brain tissues were harvested using weak RIPA lysis buffer (Cell Signaling Technology, Danvers, MA, USA), and were pre-cleared with 50% protein A/G agarose for 1 h. Then 500 ml of extracted proteins were incubated with 2 mg primary antibody overnight at 4 °C. The immune complexes were pulled down with protein A/G agarose for 4 h in a 4 °C shaker. Microbeads were collected and washed, and then proteins were eluted through boiling in 1 × loading buffer followed by immunoblotting analysis.

**Micro-MRI assay**. An MRI apparatus for small animals at 11.7T (Bruker, AVANCE 500WB) at the Animal Core Facility of Nanjing Medical University was used for the experiments. Mice were anesthetized with isoflurane and MRI was performed under the following conditions: T2-weighted fast spin echo (RARE) sequence (TR/TE = 5000/35.4 ms; FOV = 25 × 25 mm; matrix size = 256 × 256; slice thickness = 0.5 mm; NS = 8; total scan time = 10 min).

**Rotarod test and MWM test**. In brief, mice were placed on a rotating drum with speed accelerating from 5 to 50 rpm within 5 min, and the time animal drops off the drum was recorded. The training began 1 day before TBI, and the mice underwent three trials. Then, the rotarod test was performed at 2 days post-CCI by an investigator blinded to the animal group assignments.

For MWM test, we adapted this method from Sakakibara's study[63]. All experiments were performed in a white circular pool (2.0 m in diameter) with a light intensity above the center of the pool of ~2.0 m. The pool was divided into four quadrants and the average of the data in four quadrants was denoted as the data of the sample. Water temperature was maintained at 22 ± 1 °C and its color was made opaque using nontoxic white paint during hidden training sessions. First, mice (n = 8–10/group) were individually handled for 1 day before starting the experiment to acclimate them to the introduction and removal from the pool. After the habituation period, mice were subjected to 3-day visible training sessions (four trials per day), in which a platform (10 cm in diameter) was made visible by attaching a black cubic landmark. The aim of the visible training was to exclude mice with motor, visual or motivational impairments. If the mouse found the platform within a 1-min time limit, the mouse remained there for 20 s. If not, the mouse was gently guided towards the platform before staying on it. After the 20-s period on the platform, mice were placed in the cage heated by a heating pad to dry and then transported back to their home cage. The start position was changed randomly in each trial. For each trial, latency to reach the platform (s), swimming distance (cm) and swimming speed (cm/s) were measured using Time MWM software (TOPSCAN G3; ANY-MAZE 6.0.). Following the visible training sessions, the mice were subjected to 5-day hidden training sessions (four trials per day), in which the platform was placed 0.5–1.0 cm below the water surface. Four distinct

objects of different geometry were used around the pool as spatial cues. If the mouse had found the platform, the data was recorded. When mice could not find the platform within 60 s, mice were guided towards the platform and stayed on it for 10 s and the latency to reach the platform was recorded as 60 s. Then, mice were placed in the cage heated by a heating pad to dry before returning to their home cage. The start position was changed randomly to avoid track memorization, while the location of the platform was fixed throughout the experiment. Latency (s) and distance traveled (cm) to reach the platform and swimming speed (cm/s) were also measured by the software.

**Statistics and reproducibility**. Data were analyzed with GraphPad 8.0 Software and reported as the mean ± SD. A two-tailed unpaired Student's $t$ test was used to analyze the data obtained from western blot, qRT-PCR, TUNEL, CCK-8, ROS, GSH, and lipid peroxidation assays, the HETE test, action potential detection at each stimulus (pA) and the behavioral tests at each experimental time point to compare the differences between two groups. For behavioral tests, including the MWM assays, data of the whole group were analyzed using two-way ANOVA with repeated measures followed by the Tukey post hoc test. For all assays, significant differences were set at $P < 0.05$. The experiments have been repeated at least three times with similar results and representative data was shown.

**Reporting summary**. Further information on research design is available in the Nature Research Reporting Summary linked to this article.

## Data availability

The authors declare that all data supporting the findings of this study are available within this article, its supplementary information files, and source data. The source data underlying Figs. 1–7, Supplementary Figures 1–15 are provided as a Source Data file. The data of RNA-sequencing generated in this study have been deposited in the SRA database under accession code PRJNA730111. The brain data base [https://web.stanford.edu/group/barres_lab/brain_rnaseq.html] or GSE 52564 was used to identify the cell types where genes were mainly located. The datasets generated during and/or analyzed during the current study are available from the corresponding author on reasonable request. Source data are provided with this paper.

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

## Acknowledgements

This study was supported by grants from National Natural Science Foundation of China (Grant Nos. 81972153, 81471269 and 81901955), the Priority Academic Program Development of Jiangsu Higher Education Institutions (PAPD, Grant No. JX10231803), and Natural Science Foundation of Jiangsu Province (Grant No. BK20191066).

## Author contributions

J.J., N.L., X.M.W., Y.L., V.E.K., and H.B. designed and interpreted experiments. Z.Y.B., C.L., Z.M., S.M.L., G.C.S. and P.Z.Z. performed experiments. H.L.C., B.L.C., Y.F.Y., Y.M.T., Y.L.Y. and X.P.X. analyzed the data. Z.Y.B., J.J., V.E.K., and H.B. wrote the manuscript. All authors critically read the manuscript.

## Competing interests

The authors declare no competing interests.
