## [Peer Review File · Nature Communications]

Reviewers' comments:

Reviewer #1 (Remarks to the Author):

The manuscript submitted by Bao et al., highlight the pivotal anti-ferroptotic role of prokineticin 2 (Prok2) in traumatic brain injury. The authors demonstrate that Prok2 elicits its anti-ferroptotic role via induction of Fbxo10- mediated degradation of Acls4 in a ubiquitin dependent manner. In the proposed studies, the authors employ both in vivo and in vitro approaches to demonstrate the critical neuro rescue effects of ProK2 in experimental TBI. To further confirm the critical neuroprotective role of Prok2 in TBI associated neuronal injury the authors use an intracerebral viral mediated Prok2 gene delivery methodology prior to CCI to demonstrate a marked reduction in lesion volume and improvement in neurobehavioral deficits in mice subjected to CCI.

"The strengths of the manuscript include the incorporation of multiple complementary approaches (overexpression and genetic depletion) to build a convincing case for the pivotal neuroprotective role of Prok2 in experimental models of TBI. Importantly, by utilizing an in vivo model of TBI the authors are able to demonstrate that Fbxo10 driven degradation of Acls4 through a ubiquitin dependent degradation mechanism may at least in part explain the neuroprotective effects of ProK2 in this model. Despite this, the manuscript suffers from several deficiencies, which should be addressed.

1. A relevant model to test the neuroprotective role of Prok2 in relation to TBI induced brain injury would be to incorporate a mechanical stretch injury model that has been routinely used to model TBI-induced neuronal injury in vitro. The proposed in vitro primary culture model stimulated with erastin is less likely to mimic the actual brain injury associated with TBI. Thus, a more refined cell culture model as stated above would be more relevant to better understand the influence of Prok2 during TBI-associated ferroptosis-induced neuronal injury.

2. Fig.1C Please indicate the brain region from which the levels of ProK2 was assessed in the CCI mice? Is this the pericontusional area? Fig 1h. Please show double immunolabeling for ProK2 and caspase-3 in erastin treated cells as compared to vehicle treated cortical neurons and to what extent fer-1 modulates the colocalization. Does Fer-1 afford neuroprotection in erastin stimulated cortical neurons.

3. Please provide the rationale for using the proposed concentrations of erastin in the primary cortical cell culture model.

4. In the supplementary fig.2 the authors demonstrate a steady time dependent decline in the expression of the receptor for ProK2 (PKR2) in the CCI mouse brains. Since previous studies have shown that an increase in ProK2 expression levels is associated with increased PKR2 expression levels please provide WB data showing the impact of ProK2 overexpression in the CCI mouse model.

5. In Fig.2, please quantify the iron levels in primary cortical neurons cell culture exposed to erastin as compared to other treatments, including vehicle treated cells.

6. In the overall scheme the authors hypothesize that Acls4 localization to mitochondria in neurons undergoing erastin-induced ferroptosis, thus In fig.3 IHC studies please provide IHC data that Acls4 and Fbxo10 are colocalized to the mitochondria in erastin treated primary cortical neurons, this would further strengthen the stated hypothesis. Does shRNA mediated knock down of Acls4 impact erastin-induced changes in mitochondrial morphology? What is the time point at which erastin-induced mitochondrial dysfunction was studied? Please indicate it in the figure legend.

7. Fig4i the Western blot data showing accumulation of ubiquitinated aggregates is highly pixelated please replace it with a better quality blot. Please quantify the levels of ubiquitin in fig.

4j. What is the rationale for using HT22 cells in Fig.4 d and e when primary cortical neurons in culture were used in the rest of the studies? Please repeat the studies performed in HT22 cell line in primary cortical neurons, successful completion of the studies would bear more relevance to the hypothesis under study.

8. Please provide details regarding the order and the amount of viral mediated intracerebral gene delivery because subjecting the CCI mice to additional intracerebral infusion might exert additional stress on the animal. Is mitochondrial function altered in these mice? please indicate. (Fig. 6).

9. It would be beneficial to test whether prokineticin-2 agonist, IS 20 can afford neuroprotection in addition to improving neurological outcome in the proposed CCI mouse model because it may lead

to the identification of novel therapeutic avenues for the treatment of TBI-induced brain injury.

10. In the CCI mouse model are the levels of astrocytic or microglial Prok2 levels altered? Please provide the relevant data.

11. The manuscript would benefit from a thorough grammar check.

Reviewer #2 (Remarks to the Author):

This study by Bao et al. shows that prokineticin-2 (Prok2), a chemokine expressed in the injured mouse and human CNS and involved in neuroinflammation, also influences the ability of neurons to survive via its type 2 receptor, Prokr2. Notably, the authors report that Prok2 suppresses lipid peroxidation, as shown by a reduction in levels of oxygenated metabolites of arachidonic acid and lipid ROS, via augmented Fbxo10-driven ubiquitination and degradation of Acsl4, a key enzyme in the biosynthesis of arachidonic acid. AAV-mediated delivery of Prok2 or Acsl4 knockdown by means of shRNA both reduced neuronal death in vitro in response to the ferroptosis-inducing agent Erastin, as well as in vivo in the controlled cortical impact (CCI) model of traumatic brain injury (TBI). This study is therefore relevant and important as it reveals that Prok2 has the capacity to prevent ferroptosis-associated mitochondrial damage in neurons in the context of CNS injury. Although the experimental flow of the paper follows a logical path, the manuscript suffers from many problems in data analysis, conclusions and writing. From an overall perspective, the basic in vitro findings are interesting, useful and potentially important. However, the in vivo work does not appear as reliable (see comments below). Also, English-editing is required to improve the quality of the paper and facilitate the reading flow. In addition, there are quite a number of serious concerns that need to be dealt with.

Major points to be addressed:

- In Table 1, please provide the approximate time delay between TBI and tissue recovery/cryopreservation.

- In Fig. 1c, it is quite difficult to distinguish the brown from the blue color following DAB-based immunostaining of Prok2 in human brain tissue sections. Why performing counterstaining only in the control sections? Why not using immunofluorescence instead, as this would have allowed co-localization studies using neuronal and glial markers? Please provide high magnification images to be able to determine the intracellular localization of the Prok2 protein.

- In Fig. 1c & g, the Prok2 immunostaining in human versus mouse brain sections looks very different. Fig. 1c shows what seems to be mainly a nuclear signal, whereas Fig. 1g shows a cytoplasmic signal. Fig. 1g shows a relatively intense Prok2 signal in basically all neurons in sham-operated mice, suggesting a constitutive expression of the protein. In contrast, although it is difficult to say with certainty because of the presence of the counterstaining, the Prok2 protein seems less abundant in tissue sections from control human subjects (Fig. 1c). Please explain how was the specificity of the primary antibodies tested in both human mouse brain tissue sections? The same comment applies to Fig. 6l with regard to the specificity of the anti-Acsl4 and anti-Gpx4 antibodies. There is no Immunostaining subsection in the Methods section.

- In Suppl. Fig. 2, please indicate that Prok2 expression in CCI tissues was determined at the protein level using an immunoblotting approach. Also, please provide quantification for these data. Based on the image provided in Suppl. Fig. 2a, it seems that NeuN is sensitive to the neurodegenerative insult, which would either mean that protein expression is downregulated or that the number of neurons decreased because the cells died as a result of the CCI. To determine which one of these two processes is responsible for the decrease in Prokr2 expression, immunofluorescence studies will be needed.

- In Fig. 1h, the purity of the primary cortical neuronal cultures is impossible to determine based on the provided images.

- At page 6, lines 150-151, the conclusion that Prok2, but not Prokr2, plays a role in ferroptosis-related pathological changes is a hasty and controversial one. At this point of the paper, this conclusion is purely based on the fact that expression levels of the receptor do not seem to fluctuate in the pathological conditions tested in vitro. However, as the authors are aware, expression levels are not the only mechanism that determines receptor function.

- At page 6, lines 157-159, the authors state that "Out of three Prok2-interfering sequences tested, one was found most efficient in decreasing Prok2 expression", but Suppl. Fig. 5a-b shows no statistical difference between shRNAs in terms of knockdown efficiency.

- In Fig. 2b, high magnification images are required to be able to fully visualize the extent of the colocalization between TUNEL and NeuN stainings.

- In Fig. 2d, what is perceived as a morphological sign of neuronal injury should be described in more detail in the Results section.

- In Suppl. Fig. 7a, the action potentials that are shown in the middle graph for the cortical neurons treated with Erastin+Vector are extremely difficult to appreciate.

In Fig. 3a, the authors claim that mitochondrial injury can be detected using the Mito-Tracker. Please describe in the Results section (page 7) the criteria that were used to define "mitochondrial injury".

- In Figs. 3e & 6k, transmission electron microscopy (TEM) was used to reveal the presence of ruptures in the outer mitochondrial membrane of Erastin-treated neurons, but again this is far from clear in the provided images without any explanation whatsoever.

- In Fig. 5h, the schematic diagram provided is not enough detailed to understand the complexity of the dataset and overall perspective.

- In Figs. 6-7, mice were injected with AAV-Prok2 before CCI instead therapeutically, which I suppose is because AAVs take time to transduce brain cells efficiently in vivo following i.c.v. administration. Perhaps this should be discussed and put into relation with the progression of secondary damage in the CCI model (in terms of timing) and treatment issues and challenges in human TBI.

- In Fig. 6, the quality of the images provided in panel "b" showing eGFP signal in the brain of mice injected with AAV-Prok2-eGFP viruses is insufficient. High magnification images are required to be able to visualize the cells that were transduced. From what can be seen in the low magnification images provided, eGFP appears to be expressed in virtually all cells but neurons. Therefore, the target cell specificity (i.e. neurons according to Fig. 1) was probably not conferred by the AAV-Prok2-eGFP viruses.

- The provenance, background, sex and age of mice is not indicated in the Methods.

- Logically, the training on the rotarod probably started before the CCI injury. Therefore, please validate whether the following information is correct or not (lines 753-754): "The training began 1 days after TBI, and the mice underwent three trials every day, which made mice adapt to the instruction."

Minor points:

1) Several abbreviations were used in the Abstract without being defined.

2) Page 5, line 136: The time of exposure to H₂O₂ is 12 hours, not 24 hours.

3) Page 7, lines 185-187: The following sentence is incomplete: "Because healthy electrophysiological activities are necessary for neurons to exert their biological functions."

4) In Fig. 6, panel "a" is dispensable and could be removed.

5) Throughout the text, please replace "CCI treatment" by "CCI injury".

6) Some sentences throughout the text, more especially in the Discussion section, are just unreadable. Here are a few examples: Lines 450-451: "We investigated Prok2 ipregulation after CCI was source from feedback signal, though preventing cell death despite in vain." Lines 456-458: "Through comparison their study, we found Prok2 neurotoxic effects were obtained with picomolar concentrations (0.01 and 0.1 nM) they offered." Line 448: "...which would be explored and preceded in our further study". Line 516: "Remove the supernatant, re-suspend the precipitate with buffer, and lay the board." Lines 510-511: "...and their heads were quickly severed." Lines 517-518: "After 5 hours, buffer was instead by the neurobasal medium containing...." Lines 533-534: "PBS was used to pure from the left ventricle..."

7) Please indicate the source and model of the impactor used for CCI.

8) Please indicate the source of the TUNEL detection kit.

9) Line 672: the abbreviation for "transmission electron microscopy" should be corrected from TAM to TEM.

Reviewer #3 (Remarks to the Author):

Review Manuscript NCOMMS-20-08461

In the present work, the authors provide an interesting link between Prok2 and its protective effect in models of traumatic brain injury. Mechanistically the authors propose that Prok2 drives Fbxo10 expression, which ultimately decreases the expression of ACSL4. Downregulation of ACSL4 has been associated with increased protection against ferroptosis. This concept is novel and of broad interest. The study is exciting and expands our understanding of ferroptosis in neuropathologies.

I have a few remarks the authors might want to address

The authors propose a role for Prok2 to regulate ACSL4 levels, but this is not clear from their study. Does Prok2 overexpression lead to ACSL4 degradation per-se? As it is shown the results only point to this activity upon Erastin treatment.

Similarly, does the overexpression of Fbxo10 is sufficient to decrease the ACSL4 downregulation without prior stress?

Also, its unclear if the downregulation of ACSL4 (roughly 50%) is sufficient to modify the neuronal "lipidome". The authors might need to comment on this.

The authors mostly focused on stress-induced by Erastin, yet this is not relevant in the context of neurons given their extremely low expression of SLC7A11. It would be essential to demonstrate if triggers such as GPX4 inhibitors could phenocopy these effects.

Moreover given the broad importance of this regulatory link the authors could address if this is limited to neurons or could be generally applicable to different cell types.

Might be unrealistic for the revision – but a key experiment defining this link would have been to demonstrate if AVV delivery of Prok2 effect is lost in an ACSL4 knockout background. Perhaps given this lack of "hard-evidence" the authors might wish to tone down some of their statements.

Minor –

The manuscript could benefit from professional editing; some sentence are repetitive or grammatically wrong which make the reading tedious.

Dear Reviewers:

RE: manuscript NCOMMS-20-08461A-Z “Deciphering molecular anti-ferroptotic mechanisms of Prok2 in traumatic brain injury: suppression of lipid peroxidation via accelerated Fbxo10-driven ubiquitination/degradation of Acls4” by Bao et al.

We would like to thank you for the very helpful and constructive comments. The manuscript has been revised according to the reviewers’ critiques and suggestions. We performed new experiments requested by the reviewers and included the results in the revised manuscript. Our point-by-point responses to the critics of the reviewers are presented below.

Reviewer #1:

The manuscript submitted by Bao et al., highlight the pivotal anti-ferroptotic role of prokineticin 2 (Prok2) in traumatic brain injury. The authors demonstrate that Prok2 elicits its anti-ferroptotic role via induction of Fbxo10-mediated degradation of Acls4 in a ubiquitin dependent manner. In the proposed studies, the authors employ both *in vivo* and *in vitro* approaches to demonstrate the critical neuro rescue effects of Prok2 in experimental TBI. To further confirm the critical neuroprotective role of Prok2 in TBI associated neuronal injury the authors use a intracerebral viral mediated Prok2 gene delivery methodology prior to CCI to demonstrate a marked reduction in lesion volume and improvement in neurobehavioral deficits in mice subjected to CCI. The strengths of the manuscript include the incorporation of multiple complementary approaches (overexpression and genetic depletion) to build a convincing case for the pivotal neuroprotective role of Prok2 in experimental models of TBI. Importantly, by utilizing an *in vivo* model of TBI the authors are able to demonstrate that Fbxo10 driven degradation of Acls4 through a

ubiquitin dependent degradation mechanism may at least in part explain the neuroprotective effects of ProK2 in this model. Despite this, the manuscript suffers from several deficiencies, which should be addressed.

Response: We appreciate the reviewer's positive comments on our work. Those comments were all valuable and served as an important guidance for us to revise and improve our manuscript. Our point-by-point responses to the critics of the reviewer are presented below.

1. A relevant model to test the neuroprotective role of Prok2 in relation to TBI induced brain injury would be to incorporate a mechanical stretch injury model that has been routinely used to model TBI-induced neuronal injury *in vitro*. The proposed *in vitro* primary culture model stimulated with Erastin is less likely to mimic the actual brain injury associated with TBI. Thus, a more refined cell culture model as stated above would be more relevant to better understand the influence of Prok2 during TBI-associated ferroptosis-induced neuronal injury.

Response: As recommended by the reviewer, we performed a series of new experiments using a mechanical stretch injury to cortical neurons *in vitro*. Anti-tau antibody was used for staining of neurons. Mechanical stretch injury caused thinning and disruption of neurites (Fig. 1i) accompanied by a rapid increase of Prok2 expression (Fig. 1j and k). Furthermore, 9 h after stretch, upregulation of the cleaved caspase-3 was observed (Fig. 1j and k), which was not found in Erastin exposure (Supplementary Fig. 4a-d). This indicated that stretch injury was also associated with the activation of caspase-dependent apoptotic cell death occurring at a later time point than the Prok2 mediated ferroptosis. Therefore, primary neurons were evaluated 6h after stretch for activation of ferroptosis, because cleaved caspase-3 was not increased at that time point. (Page 5-6)

Fig. 1 (i) Immunofluorescence assay (labeling of tau protein, red) is used to monitor mechanical stretch; Hoechst staining is used to stain cell nuclei. Scale bar is 15µm and 8µm, respectively.

Fig. 1 (j and k) Western blot analysis and quantification of Prok2 and cleaved caspase-3 expression in control and stretch groups. GAPDH is used as loading control. Error bars represent S.D. ($n = 3$). * $P < 0.05$ and ** $P < 0.01$ versus control group.

Supplementary Fig. 4 (a and b) Erastin (within the concentration range from 5 to 20 μM), does not significantly affect the levels of cleaved caspase-3 (GAPDH is used as a control). Error bars represent S.D. ($n = 3$ experiments). **(c-e)** Immunostaining is used to detect the expression of Prok2 (green) and cleaved caspase-3 (red). Hoechst is employed to stain cell nucleus. Scale bar is 20 μm. The fluorescence intensity is measured by ImageJ. Error bars represent S.D. ($n = 3$ experiments). * $P < 0.05$ versus 0 μM Erastin group.

We found that overexpression of Prok2 in primary cortical neurons alleviated stretch induced ferroptotic lipid peroxidation (as evidenced by accumulation of 12-HETE, 15-HETE and MDA as well as enhanced C11-BODIPY response, and ROS production,) and cellular injury (documented by TUNEL staining, LDH release, Prussian blue staining and cell viability assays) (Supplementary Fig. 6 and Supplementary Fig. 8). (Page 7-8)

Supplementary Fig. 6 Overexpression of Prok2 attenuates stretch-induced cell death. Primary cortical neurons are exposed to stretch. The following characteristics are assessed at 6h after stretch: **(a-b)** Cell death assessed by TUNEL staining. Scale bar is 50µm. * $P < 0.05$ and *** $P < 0.001$. Error bars represent S.D. ($n = 3$ experiments). **(c-d)** Prussian blue staining shows that stretch promotes aggregation of Fe^{3+} (blue) in neurons and Lv-Prok2 inhibits this process. Scale bar is 15 µm. The relative iron levels are assessed by quantitation of color intensity using ImageJ. Error bars represent S.D. ($n = 3$ experiments). * $P < 0.05$ and ** $P < 0.01$. **(e-f)** Cell viability is analyzed by CCK-8. Cytotoxicity induced by stretch is measured by LDH assay. Error bars represent S.D. ($n = 4$ experiments). * $P < 0.05$, ** $P < 0.01$ and *** $P < 0.001$. **(g)** Expression of Prok2, Gpx4 and Acsl4 is normalized to GAPDH levels used as controls in Western blot assays.

Supplementary Fig. 8 Prok2 overexpression alleviates stretch induced lipid peroxidation. (a) ROS generation (b-c) Lipid peroxidation detected by BODIPY 581/591 C11 staining (green). Hoechst is used to stain cell nuclei. Scale bar is 50µm. The percentage of BODIPY 581/591 C11-positive cells is presented. (d) GPX activity. (e) Lipid peroxidation assessed by MDA assay. (f, g) 15-HETE and 12-HETE generation quantified by corresponding kits. * $P < 0.05$, ** $P < 0.01$ and *** $P < 0.001$. Error bars represent S.D. ($n = 4$ experiments); $n = 3$ experiments for measurements of BODIPY 581/591 C11-positive cells; $n = 5$ experiments for GPX activity assays.

Notably, neuroprotective effect of Prok2 after mechanical stretch could not be realized in the presence of shACSL4 (Supplementary Fig. 10). Similar to CCI *in vivo*, Gpx4 levels were not changed in neurons after stretch injury but were upregulated by Prok2. (Page 9)

Supplementary Fig. 10 Prok2 protects mitochondrial function in stretch model of neuronal injury in an Acsl4 dependent manner. (a) Expression and intracellular distribution of Tomm20 (red) and Prok2 (green) are shown by immunofluorescence. Hoechst stains cell nuclei. Scale bar is 15 μ m. (b-g) Expression levels of Acsl4, Gpx4, Tomm20, Tfam and MT-ND1 in Acsl4 deficient (shAcsl4) and control (shCtrl) primary neurons. GAPDH is

used as a control. * $P < 0.05$, ** $P < 0.01$ and *** $P < 0.001$. (h) ATP levels are decreased upon stretch exposure. Overexpression of Prok2 prevents the decrease in ATP levels in stretched cells. * $P < 0.05$ and ** $P < 0.01$. Error bars represent S.D. ($n = 3$ experiments). (i) Representative electron microscopy images show that stretch does not cause mitochondrial shrinking in shAcs14 primary neurons. Scale bar is 500nm.

Finally, overexpression of Prok2 enhanced levels of Fbxo10, promoting Acs14 ubiquitination and degradation in stretch-exposed cells (Supplementary Fig. 12).

Prok2-induced ubiquitination of Acs14 was decreased in Fbxo10 deficient cells (Page 11 and 12).

Supplementary Fig. 12 Prok2 promotes Acs14 ubiquitination and degradation in stretch model. (a) Primary cortical neurons expressing control empty vector or Lv-Prok2 are treated with MG132 (20 μ M) to block proteasomal degradation for 6 h and then exposed to mechanical stretch. Total lysates are analyzed for Acs14 and ubiquitinated proteins by immunoblotting using anti-Acs14 and anti-Ub antibodies. The decrease in Acs14 protein observed upon Prok2 overexpression is abolished by MG132 and whereas ubiquitination of Acs14 is increased. (b)

IP with Acsl4 antibody and western blot with anti-Ub antibody show that Acsl4 ubiquitination is higher in Prok2 overexpressing plus MG132 treated cells versus control untreated cells as well as empty vector transfected plus MG132 treated cells. IgG is used as a negative control. (c) Immunoprecipitation with Acsl4 followed by western blot to Ub shows that overexpression of Fbxo10 increases Acsl4 ubiquitination in the presence or absence of stretch. (d) Overexpression of Prok2 in primary cortical neurons increases the level of Fbxo10 in the presence or absence of stretch. (e) Primary neurons are co-transfected with Lv-Prok2-Flag and Fbxo10 shRNA. Lysates of primary neurons exposed to stretch are immunoprecipitated with Acsl4 antibody followed by western blot with anti-Ub. Prok2-induced ubiquitination of Acsl4 is decreased in Fbxo10 deficient cells.

Overall, these new data are compatible with the notion that the neuroprotective effects of Prok2 in stretch-induced injury are realized via its effects on ferroptosis. These new data are included in the Results section of the revised manuscript.

Detailed description of stretch model used is presented in the Methods section of the revised manuscript. (page 21)

In vitro TBI (stretch model) was performed using a well-established model of mechanical stretch injury as previously described¹. Briefly, primary cortical neurons were seeded (7×100,000 cells/well) on silicone membranes within custom-made stainless-steel wells 5 days before the stretch exposure. Wells were fitted into the stretch apparatus, and a severe stretch injury (strain rate, 10/s; membrane deformation, 50%; peak pressure, 3–4 psi) was applied to simulate a strain field similar to that in our *in vivo* TBI model.

2. Fig.1C Please indicate the brain region from which the levels of Prok2 was assessed in the CCI mice? Is this the pericontusional area?-Fig 1h. Please show double immunolabeling for Prok2 and caspase-3 in erastin treated cells as compared to vehicle treated cortical neurons and to what extent fer-1 modulates the colocalization. Does Fer-1 afford neuroprotection in erastin stimulated cortical neurons.

Response: Brain tissue samples for testing Prok2 levels were obtained from the peri-contusional area. Diagrammatically, this is shown in Fig.1d.

Fig.1 (d) Schematic representation of the contusional region (red) and the peri-contusional area (blue) after CCI. Tissues from peri-contusional area (blue) are collected for western blotting and qRT-PCR analysis.

The concentration of Erastin used ranged from 5 μ M to 20 μ M. We performed dual immunofluorescence staining with Prok2 and caspase-3 in Erastin treated cells and the results are shown in Supplementary Fig. 4c-e. Additionally, western blot assays were performed to quantitatively assess the levels of Prok2 and caspase-3 (Supplementary Fig. 4a and b). We found that expression of Prok2 rapidly increased after Erastin administration; however, the levels of cleaved caspase-3 remained unchanged. (Page 6)

Supplementary Fig. 4 (a and b) Erastin (within the concentration range from 5 to 20 μ M), does not significantly affect the levels of cleaved caspase-3 (GAPDH is used as controls). Error bars represent S.D. ($n = 3$ experiments). (c-e) Immunostaining is used to detect the expression of Prok2 (green) and cleaved caspase-3 (red). Hoechst is

employed to stain cell nucleus. The fluorescence intensity is measured by ImageJ. Error bars represent S.D. ($n = 3$ experiments). * $P < 0.05$ versus 0 μM Erastin group.

The concentrations of Fer-1 used ranged from 0.5 μM to 1.5 μM . We found that 1 μM Fer-1 could inhibit increases in *Prok2* expression (Supplementary Fig. 5a-b and d-e) and cellular cytotoxicity (Supplementary Fig. 5c) induced by Erastin in primary cortical neurons. This is in line with previous studies showing neuroprotective effects of Fer-1^{2,3}. (Page 6)

Supplementary Fig. 5 (a-c) Fer-1 decreases *Prok2* mRNA and LDH induced by Erastin. *** $P < 0.001$ versus Con group; # $P < 0.05$, ## $P < 0.01$ and ### $P < 0.001$ versus Erastin group. Error bars represent S.D. ($n = 3$ experiments). **(d-f)** Dual-immunofluorescence staining with *Prok2* (green) and cleaved caspase-3 (red) is used to detect their expressions after Fer-1 administration. Hoechst is used for staining cell nuclei. The fluorescence intensity is measured by ImageJ. Error bars represent S.D. ($n = 3$ experiments). *** $P < 0.001$ versus 0 μM Erastin group; # $P < 0.05$ and ## $P < 0.01$ versus 20 μM Erastin group. NC (negative control) refers to Erastin (20 μM) group.

3. Please provide the rationale for using the proposed concentrations of Erastin in the primary cortical cell culture model.

Response: The choice of Erastin concentration was based on the previously published work with neuronal cells from other labs. Wang et al. showed a protective role of mitochondrial ferritin on Erastin-induced ferroptosis in SH-SY5Y cells⁴. It was shown that 10 μ M Erastin (24 h) induced cytoplasmic ROS production, increased intracellular labile iron pool (LIP) levels and cell death. Considering that SH-SY5Y cells are human neuroblastoma-derived immortalized cell line, they may have different tolerance to Erastin compared to mouse primary cortical neurons used in our study. In a recent study by Zhang et al⁵ on primary cortical neurons, several Erastin concentrations (1, 10, 20 or 50 μ M, 5 days after seeding) were tested and 50 μ M Erastin was selected as the optimized effective concentration. In our study, we found 20 μ M Erastin could trigger neuronal ferroptosis, including increased LDH content, MDA level and TUNEL-positive rate. Erastin at 5, 10 and 20 μ M and these concentrations were lower than that used by Zhang et al most likely due to a different cell density of primary cortical neurons seeded. Wu et al.⁶ reported that at higher cell density, increased E-cad expression and enrichment at the sites of cell-cell contacts, suppressed ferroptosis.

4. In the supplementary fig.2 the authors demonstrate a steady time dependent decline in the expression of the receptor for ProK2 (PKR2) in the CCI mouse brains. Since previous studies have shown that an increase in ProK2 expression levels is associated with increased PKR2 expression levels please provide WB data showing the impact of ProK2 overexpression in the CCI mouse model.

Response: Indeed, the data on Supplementary Fig. 2a, showed the decline of Prokr2 and NeuN expression after CCI. As Prokr2 was mainly expressed in neurons, we performed dual immunofluorescent staining with Prokr2 (Green) and NeuN (Red) to more specifically address this point (Supplementary Fig. 2c). We found the presence of Prokr2 almost exclusively in NeuN-positive cells. And the fluorescence intensity of

Prokr2 per NeuN-positive cell was not altered after CCI (Supplementary Fig. 2d). Furthermore, Co-IP assays showed that CCI did not impact the intracellular relationships between Prok2 and Prokr2, while the averaged expression of the proteins in the brain tissue changed (Supplementary Fig. 2e). This is in line with the immunofluorescent dual-labeling with Prokr2 (Green) and Prok2 (Red) (Supplementary Fig. 2f and g). Thus, the decreased total expression of Prokr2 was most likely due to the reduced numbers of NeuN-positive cells. (Page 5)

Supplementary Fig. 2 (a) Expression of Prokr2 and NeuN tested after CCI. GAPDH is used as a control ($n = 3$ mice per group). (b) Brain RNA Data base indicates that the Prokr2 expression is specifically located in neurons. (c and d) Immunofluorescence staining shows that Prokr2 is expressed predominantly in NeuN-positive cells.

Scale bar is 20 μ m and 10 μ m. DAPI is used to stain cell nucleus. Prokr2 fluorescence intensity per NeuN-positive cell is measured by ImageJ. Error bars represent S.D. ($n = 4$ in sham group and $n = 3$ in CCI group). (e) Co-immunoprecipitation assays document that interactions between Prok2 and Prokr2 are not changed after CCI. Note the changes in the total expression of two proteins after CCI. (f and g) Immunofluorescence studies with dual-labeling for Prok2 (red) and Prokr2 (green). Scale bar is 20 μ m. Prok2 and Prokr2 fluorescence intensities per cell are measured by ImageJ. *** $P < 0.001$ versus sham group in Prok2 expression. Error bars represent S.D. ($n = 7$ in sham group and $n = 5$ in CCI group).

As recommended by the Reviewer, we performed new experiments to examine the effect of Prok2 expression on the levels of Prokr2. We employed western blot assays to assess the expression of Prokr2 after AAV-Prok2 administration in CCI model (Supplementary Fig. 13 e and f). (Page 13)

Supplementary Fig. 13 (e-f) Western blot assays documenting Prokr2 contents after Prok2 overexpression in CCI model. Error bars represent S.D. ($n = 3$ mice per group). * $P < 0.05$.

We found Prok2 overexpression attenuated CCI-induced decrease in Prokr2 levels. This was likely due to decreased lesion volume and TUNEL staining observed with Prok2 overexpression after CCI (Fig. 6 h, i and m).

5. In Fig.2, please quantify the iron levels in primary cortical neurons cell culture exposed to Erastin as compared to other treatments, including vehicle treated cells.

Response: As recommended the quantification of the Prussian blue staining was performed to determine the iron levels in primary cortical neurons. Cellular iron accumulation was detected by Prussian Blue Iron Stain Kit (60533ES20, Yeasen,

China). The depth of blue is measured by ImageJ. We showed that Erastin increased iron levels in primary cortical neurons, and this effect was suppressed by Prok2 overexpression (Fig. 2f and g). Low levels of Prok2 were associated with the increased contents of iron preventable by Fer-1 treatment. In addition, data on the iron staining and analysis in stretch model were presented in Supplementary Fig. 6c and d. (Page 7)

Fig. 2 (f and g) Prussian blue staining shows that Erastin stimulates the formation of Fe³⁺ (blue) in neurons. Lv-Prok2 suppresses this effect. Scale bar is 10 µm. The iron levels are determined based on the color intensities measured by ImageJ. Error bars represent S.D. (*n* = 3 experiments). * *P* < 0.05 and ** *P* < 0.01.

Supplementary Fig. 6 (c-d) Mechanical stretch leads to increase in intracellular Fe³⁺ (blue) levels assessed by Prussian blue staining in primary cortical neurons. Lv-Prok2 attenuates this increase. Scale bar is 10 µm. Relative iron level is analyzed according to color intensity in ImageJ. Error bars represent S.D. (*n* = 3 experiments). * *P* < 0.05 and ** *P* < 0.01.

6. In the overall scheme the authors hypothesize that Acsl4 localization to mitochondria in neurons undergoing erastin-induced ferroptosis, thus In fig.3 IHC studies please provide IHC data that Acsl4 and Fbxo10 are colocalized to the mitochondria in erastin treated primary cortical neurons, this would further strengthen the stated hypothesis. Does shRNA mediated knock down of Acsl4 impact erastin-induced changes in mitochondrial morphology? What is the time point at which erastin-induced mitochondrial dysfunction was studied? Please indicate it in the figure legend.

Response: We thank the Reviewer for this excellent suggestion that will strengthen and improve our study. Accordingly, we performed three-colour fluorescence staining of Acsl4 (red), Fbxo10 (green) and mitochondria (Mito-tracker, blue) and confirmed the Acsl4 and Fbxo10 colocalization to the mitochondria in Erastin treated primary cortical neurons (Fig. 5 c in the revised manuscript). Erastin caused upregulation of Acsl4 and decrease of Fbxo10 staining. The intensity of mitochondrial staining was decreased particularly in the neurites upon Erastin exposure.

Fig. 5 (c) Immunofluorescence staining shows co-localization of Fbxo10 (green), Acsl4 (red) and mitochondria (Mito-tracker, blue) upon exposure to Erastin in primary cortical neurons. Scale bar is 10 μ m.

TEM was used to detect the changes in mitochondrial morphology (Fig. 3 n). Erastin had little impact on mitochondrial morphology in shAcsl4 treated primary neurons: shrunken mitochondria and outer membrane rupture were not observed. Similar results were reported in our earlier study⁷ in which RSL3 induced mitochondrial shrinkage and outer membrane rupture observed in Acsl4 WT pfa1 cells, was not

observed in *Acsl4*^{-/-} cells. Erastin-induced mitochondrial dysfunction was assessed at 24h. It is true for all the others measures, too, such as Tomm20, mtDNA, and ATP levels.

n

Fig. 3 Prok2 protects mitochondrial function in an *Acsl4* dependent manner. 20 μ M Erastin for 24 h is used to induce ferroptosis. **(n)** Representative TEM images illustrating that Erastin treatment does not cause marked changes of mitochondrial morphology in shAcsl4 primary neurons. Scale bar is 500nm.

7. Fig4i the Western blot data showing accumulation of ubiquitinated aggregates is highly pixelated please replace it with a better quality blot. Please quantify the levels of ubiquitin in fig. 4j. What is the rationale for using HT22 cells in Fig.4 d and e when primary cortical neurons in culture were used in the rest of the studies? Please repeat the studies performed in HT22 cell line in primary cortical neurons, successful completion of the studies would bear more relevance to the hypothesis under study.

Response: We apologize for the insufficient quality of the images of the western blots presented on previous Fig. 4i showing the accumulation of ubiquitinated aggregates. We replaced this with high-quality images of the blots.

IP with *Acsl4* antibody and western blot with anti-Ub antibody showed Ub binding to *Acsl4*, indicating the level of ubiquitination of *Acsl4*. The ubiquitination of *Acsl4* was higher in Prok2 overexpressing plus MG132 treated cells versus control untreated cells as well as empty vector transfected plus MG132 treated cells. IgG is used as a

negative control (Fig. 4j). The level of ubiquitination of Acsl4 was analyzed in Fig. 4k.

Fig. 4 (i) Primary cortical neurons expressing control empty vector or Lv-Prok2 are treated with MG132 (20 μ M for 6 h) to block proteasomal degradation and then exposed to 20 μ M Erastin for 24h. Total lysates are analyzed for Acsl4 and ubiquitinated proteins by immunoblotting using anti-Acsl4 and anti-Ub antibody. The decrease in Acsl4 protein observed upon Prok2 overexpression is abolished by MG132 and ubiquitination of Acsl4 is increased. (j and k) IP with Acsl4 antibody and western blot with anti-Ub antibody show that ubiquitination of Acsl4 is higher in Prok2 overexpressing plus MG132 treated cells versus control untreated cells as well as empty vector transfected plus MG132 treated cells. IgG is used as a negative control. The level of ubiquitination of Acsl4 is analyzed. * $P < 0.05$ versus Erastin+Lv-Prok2 group; ** $P < 0.01$ versus Erastin+MG132 groups. Error bars represent S.D. ($n = 3$ experiments).

As requested by the Reviewer, we repeated the experiments related to Fig. 4d using primary cortical neurons and presented the new results in the revised manuscript (Page 10-11). Primary cortical neurons were treated with 5 μ g/ml CHX, 200 nM bortezomib (Bort), 50 nM bafilomycin A1 (Baf A1), or a combination of Bort and Baf A1 for 1 h prior to the addition of 20 μ M Erastin. Cell lysates were isolated after 24 h of Erastin treatment and immunoblotted for Acsl4. We found that Bortezomib alone or combined with bafilomycin A1 largely blocked the Prok2-mediated degradation of Acsl4 while bafilomycin A1 alone had no effect (Fig. 4d and e) suggesting that inhibition of ubiquitin- proteasome pathway was mostly responsible for Acsl4 degradation after Prok2 overexpression. The autophagy-lysosome pathway likely played a minor role in degradation of Acsl4 upon Prok2 overexpression.

Fig. 4 (d, e) Primary cortical neuronal cells are treated with 5 $\mu\text{g/ml}$ CHX, 200 nM bortezomib (Bort), 50 nM bafilomycin A1 (Baf A1), or a combination of Bort and Baf A1 for 1 h prior to the addition of 20 μM Erastin. Cell lysates are obtained after 24 h of Erastin treatment and immunoblotted for Acsl4 and GAPDH. Bar graph shows Acsl4 expression normalized to GAPDH under different conditions. Acsl4 signal corresponding to Lv-Prok2 condition is employed as 100%. Bort alone or in combination with Baf A1 blocks Prok2-mediated degradation of Acsl4. * $P < 0.05$ versus respective control groups. Error bars represent S.D. ($n = 3$ experiments).

8. Please provide details regarding the order and the amount of viral mediated intracerebral gene delivery because subjecting the CCI mice to additional intracerebral infusion might exert additional stress on the animal. Is mitochondrial function altered in these mice? please indicate. (Fig. 6).

Response: As requested, the time course of intracerebral injections is shown in Supplementary Fig. 13a of the revised manuscript (Page 12-13).

Supplementary Fig. 13 (a) The schedule of intracranial gene and shRNA delivery AAV-Prok2 or -shFbxo10 are designed for injection into mouse brain prior to CCI for effective transfection. At 7 days before CCI, AAV-Prok2 is injected. As for injection both AAV-Prok2 and -shFbxo10, the first injection, AAV-shFbxo10, is performed at 14 days before CCI. **(b-c)** Immunofluorescence staining of Tomm20 (red) and Prok2 (green) is used to test for possible effects of AAV injections on the mitochondria. No significant differences in Tomm20 staining between control and AAV-Prok2 are found. Scale bar is 20 μ m. Error bars represent S.D. ($n = 3$ mice per group). ** $P < 0.01$ versus control group. **(d)** Representative TEM images of mitochondria in control and AAV-Prok2 groups. Scale bar is 500nm.

Mice were anesthetized persistently by 2% isoflurane. They were placed in a stereotaxic apparatus (RWD, Shenzhen, China). The stereotactic coordinates were as follows: anteroposterior (AP), -0.4 mm; mediolateral (ML), -1.0 mm; dorsoventral (DV), -3.0 mm from the bregma for injection. Animals were injected with 10 μ l AAV using a syringe with a 0.52 mm needle (Ito Co., Shizuoka, Japan). For the optimized gene delivery and protein expression, mice received the intracranial injection of AAV 7 days before CCI injury. For both AAV-Prok2 and AAV-shFbxo10 co-transfection, the first injection was performed 14 days before CCI (Supplementary Fig. 13a). The virus (with the titer of 1×10^{13} in 10 μ l) was injected using a syringe with a 0.52 mm needle (Ito Co., Shizuoka, Japan). The injection time was 10 min, and the needle was left at the injection site for additional 10 min prior to withdrawal.

To test whether the injection impacted the mitochondrial function due to additional intracerebral infusion and stress, we performed dual immunofluorescence staining to examine the mitochondria (Tomm20, red) and Prok2 (green) after second injection. No changes of Tomm20 were found (Supplementary Fig. 13b and c). TEM revealed no changes in mitochondrial morphology (Supplementary Fig. 13d). Thus, we

concluded that the procedure of intracerebral injections of AAV altering target gene expression did not affect mitochondria. Detailed description of the procedures is presented in the Methods section (Page 30).

9. It would be beneficial to test whether prokineticin-2 agonist, IS 20 can afford neuroprotection in addition to improving neurological outcome in the proposed CCI mouse model because it may lead to the identification of novel therapeutic avenues for the treatment of TBI-induced brain injury.

Response: The Reviewer is correct: it has been shown that IS20 administration increased Prok2 function and improved neurological outcome acting via multiple mechanisms such as ⁸: i) enhanced astrocyte migration accompanied by a shift in mitochondrial energy metabolism, ii) reduction in pro-inflammatory factors, iii) increased expression of the antioxidant genes, Arginase-1 and Nrf2, iv) increased glutamate uptake due to the upregulation of GLAST. Collectively, their results revealed that IS20 regulated a novel neuron-astrocyte signaling mechanism by promoting an alternative *A2 protective phenotype in astrocytes*, which could be exploited for the development of novel therapeutic strategies for PD and other related chronic neurodegenerative diseases. It is likely that IS20 is a potentially promising agent for treatment of TBI.

In spite of these important features related to astrocytes ⁸, it is not clear whether IS20 can realize its protective potential against ferroptosis in neuronal cells. Our data suggest the effects of AAV-Prok2 are related to its expression in neuronal cells. In the CCI model, Prok2 overexpression upregulated Fbxo10 in neurons, triggering Acsl4 ubiquitination/ degradation and suppression of lipid peroxidation and neuronal cell damage and death. Furthermore, Dr. Arthi Kanthasamy's team in their study⁸ demonstrated that they found IS20 is a non-peptide Prokr1 agonist and almost complete Prokr1 KD (by a CRISPR/Cas9 lentiviral technique) completely abolished the IS20-mediated increase in the anti-inflammatory arginase-1 and Nrf2, without

affecting iNOS gene expression. Cheng et al ⁹ showed that *Prokr1* mRNA expression was high in microglia, whereas *Prokr2* mRNA expression was robust in neurons (please, see Fig. 2B in their study). In CCI mouse model, we would not be able to distinguish the source of the IS20 neuroprotective response: anti-ferroptotic action in neuronal cells vs. anti-inflammatory effects induced by A2 astrocytes or microglia. Based on these facts and our data on neuron-centric mechanisms of Prok2's protective action, we believe that the beneficial effects of IS20 are not likely to be related to the central subject of our study.

10. In the CCI mouse model are the levels of astrocytic or microglial Prok2 levels altered? Please provide the relevant data.

Response: On average, an increased Prok2 expression was found in CCI tissues. To specifically address the question on the protein expression in neurons, astrocytes and microglia, we performed dual-labeling immunofluorescence assay (Fig. 1g left panel). Prok2 was mainly expressed in neurons (labelled by NeuN, green). Astrocytes (labelled by GFAP, green) and microglia (labelled by iba-1, red) showed low levels of Prok2. CCI stimulated Prok2 expression in NeuN-positive neurons. Slight changes were found in astrocytes and microglia in the CCI model (Fig. 1g right panel) (Page 5).

Fig.1 (g) Dual-labeling immunofluorescent staining indicates that Prok2 expression is most prominent in neurons (NeuN-labelled), whereas low levels of Prok2-staining are found in astrocytes (GFAP-labelled) and microglia (iba-1-labelled). Scale bar is 15µm. Quantification of Prok2 relative fluorescence intensity is shown in the right panel. Error bars represent S.D. ($n = 3$). ** $P < 0.01$ versus sham.

11. The manuscript would benefit from a thorough grammar check.

Response: We are thankful to the Reviewer for this comment. The entire manuscript has been thoroughly edited and proofread.

Reviewer #2:

This study by Bao et al. shows that prokineticin-2 (Prok2), a chemokine expressed in the injured mouse and human CNS and involved in neuroinflammation, also influences the ability of neurons to survive via its type 2 receptor, Prokr2. Notably, the authors report that Prok2 suppresses lipid peroxidation, as shown by a reduction in levels of oxygenated metabolites of arachidonic acid and lipid ROS, via augmented Fbxo10-driven ubiquitination and degradation of Acs14, a key enzyme in the biosynthesis of arachidonic acid. AAV-mediated delivery of Prok2 or Acs14 knockdown by means of shRNA both reduced neuronal death *in vitro* in response to the ferroptosis-inducing agent Erastin, as well as *in vivo* in the controlled cortical impact (CCI) model of traumatic brain injury (TBI). This study is therefore relevant and important as it reveals that Prok2 has the capacity to prevent ferroptosis-associated mitochondrial damage in neurons in the context of CNS injury. Although the experimental flow of the paper follows a logical path, the manuscript suffers from many problems in data analysis, conclusions and writing. From an overall perspective, the basic *in vitro* findings are interesting, useful and potentially important. However, the *in vivo* work does not appear as reliable (see comments below). Also, English-editing is required to improve the quality of the paper and facilitate the reading flow. In addition, there are quite a number of serious concerns that need to be dealt with.

Response: We thank the reviewer's positive assessment of our work. Those comments were all valuable and very helpful and served as an important guidance for us to revise and improve our manuscript. Our point-by-point responses to the critics of the reviewer are presented below.

Major points to be addressed:

1- In Table 1, please provide the approximate time delay between TBI and tissue recovery.

Response: We collected the data on the approximate time delay between TBI and tissue cryopreservation. The time delays were estimated as 6.5h, 9h, 6h, 7h and 5.5h, respectively for the 5 TBI patients. Table1 was updated to present this information in the revised manuscript.

Detailed description of the procedure employed for procurement of the samples, including the time delays, is included in the revised manuscript (Page 19).

Table 1. Demographic and clinical characteristics of TBI patients.

TBI Patient	Accident type	Injury area	Gender	Approximate time delay between TBI and tissue cryopreservation (h)	Age	GCS scores
1	Traffic accident	Left temporal lobe	Male	6.5	42	9
2	Fall	Occipital lobe	Female	9	70	8
3	Traffic accident	Right temporal lobe	Male	6	52	10
4	Traffic accident	Left temporal lobe and occipital lobe	Female	7	58	5
5	Traffic accident	Left parietal lobe	Female	5.5	60	7

2- In Fig. 1c, it is quite difficult to distinguish the brown from the blue color following DAB-based immunostaining of Prok2 in human brain tissue sections. Why performing counterstaining only in the control sections? Why not using immunofluorescence instead, as this would have allowed co-localization studies using neuronal and glial markers? Please provide high magnification images to be able to determine the intracellular localization of the Prok2 protein.

Response: In the previously submitted manuscript, Fig. 1c, DAB-based immunostaining in sham and TBI group was performed under the same conditions. We apologize for presenting the images of insufficiently high quality. As suggested by the Reviewer, we conducted immunofluorescence staining to reveal the expression of Prok2 in Fig. 1c.

Fig. 1 (c) Immunofluorescence assessment of Prok2 expression (green) in Control and TBI brain tissue. Scale bar is 50µm. DAPI is used to label nucleus.

Co-localization studies were also performed using NeuN (a marker of neuron), GFAP (a marker of astrocyte) and iba-1(a marker of microglia) (Fig. 1g). The results demonstrated that CCI significantly increased Prok2 mainly in neurons.

Fig.1 (g) Dual-labeling immunofluorescent staining indicates that Prok2 expression is most prominent in neurons (NeuN-labelled), whereas low levels of Prok2-staining are found in astrocytes (GFAP-labelled) and microglia (iba-1-labelled). Scale bar is 15µm. Quantification of Prok2 relative fluorescence intensity is shown in the right panel. Error bars represent S.D. ($n = 3$). ** $P < 0.01$ versus sham.

3- In Fig. 1c & g, the Prok2 immunostaining in human versus mouse brain sections looks very different. Fig. 1c shows what seems to be mainly a nuclear signal, whereas Fig. 1g shows a cytoplasmic signal. Fig. 1g shows a

relatively intense Prok2 signal in basically all neurons in sham-operated mice, suggesting a constitutive expression of the protein. In contrast, although it is difficult to say with certainty because of the presence of the counterstaining, the Prok2 protein seems less abundant in tissue sections from control human subjects (Fig. 1c). Please explain how was the specificity of the primary antibodies tested in both human mouse brain tissue sections? The same comment applies to Fig. 6l with regard to the specificity of the anti-Acsl4 and anti-Gpx4 antibodies. There is no Immunostaining subsection in the Methods section.

Response: We apologize for failing to provide sufficient information and description of the immunostaining protocols in the Methods section. The same primary Prok2 antibody was used to perform immunostaining in human and mouse brain tissues (cat No. ab76747, Abcam). The manufacturer's instructions indicate the species reactivity of the primary antibody of Prok2 and include human, mouse and rat. We presented two references related to the use of ab76747.

In the first one¹⁰, the primary anti-Prok2 (ab76747) was used to perform immunofluorescence assay in mouse brain tissues. The second one¹¹ demonstrated that the anti-Prok2(ab76747) could be also used for performing IHC assays in human liver tissues. Based on these data, we concluded that using anti-Prok2 (ab76747) for IHC assays both in human and mouse tissues was appropriate.

To exclude the interference of the secondary antibodies and DAB staining, we conducted IHC staining again with and without Prok2 primary antibody (not shown in revised Figure).

(a) Immunohistochemical staining for Prok2 in human brain tissues. Scale bar is 50 µm. **(b)** Immunohistochemical stain for Prok2 in mouse brain tissues. Scale bar is 50 µm.

The images obtained displayed blue cell nuclei and light background, which was used as a negative control. Human brain tissues might have contained blood clots resulting in a slightly yellowish background in negative controls. Thus, the secondary antibody and DAB did not interfere with the assay suggesting that the brown staining in cells was due to Prok2 expression. The same dilutions of the antibodies (primary antibody 1:200, secondary antibody 1:500) were used with human (control and TBI) and mouse (sham and CCI) brain tissues. We found markedly increased intracellular brown positive coloring in TBI or CCI tissues. Notably, the background in human tissues was more clear and lighter than darker coloring in mouse tissues.

In response to the Reviewer's comments, we removed IHC staining by DAB and showed immunofluorescence images in the revised manuscript. Specifically, in Fig. 1g, double labelling with Prok2 and cell markers was performed.

In Fig. 6, anti-Acs14 (ab155282, Abcam) and anti-Gpx4 (ab125066, Abcam) were used to perform IHC. They both reacted with target proteins in mouse tissues as

described in the manufacturers' instructions. The protocol for the use of secondary antibodies remained unchanged.

In the revised manuscript, we included the immunostaining assay protocols in the *Methods* section. In line with the planned time course study, appropriate anesthesia was performed, and mice were intracardially perfused with pre-cooled phosphate buffered saline solution followed by pre-cooled 4% paraformaldehyde. The brain tissues obtained were placed in 4% paraformaldehyde for 24 h followed by dehydration in graded sucrose solution series of 10%, 20% and 30%. 12 μ m sections were prepared to perform immunofluorescence and DAB-immunohistochemistry assays. For immunofluorescence assays, sections were permeabilized with 0.1% Triton X-100 for 15 min, and blocked with 5% bovine serum albumin in PBS at 37 °C for 1 h. Samples were incubated with the specific primary antibodies at 4 °C overnight, washed three times with PBS, and incubated with Alexa Fluor 488- or CyTM3-conjugated secondary antibodies (Jackson, USA) at 37 °C for 1 h. After additional PBS washes, Hoechst nuclear staining (C1018, Beyotime, China) was performed at room temperature for 10 min. A laser scanning confocal microscope (TCS SP5II, Leica, Wetzlar, Germany) was used to acquire immunofluorescence images, quantified by ImageJ. For immunohistochemistry assays, the sections were incubated with the secondary antibody (ab6112, Abcam) for 30 min at 37°C, rinsed with phosphate-buffered saline, and incubated with DAB for 15 min at 37°C. Slides were imaged under a light microscope (Leica).

Above protocols of immunofluorescence are applied to brain tissue sections. As for cell immunofluorescence, some different processes are noted. Primary neuronal cells were plated on glass slide precoated with poly-lysine (PLL). After relative treatments, 4% paraformaldehyde was used to perform fixation of cells for 1h. The following protocols were the same as brain tissue immunofluorescence assay.

4- In Suppl. Fig. 2, please indicate that Prokr2 expression in CCI tissues was determined at the protein level using an immunoblotting approach. Also, please provide quantification for these data. Based on the image provided in Suppl. Fig. 2a, it seems that NeuN is sensitive to the neurodegenerative insult, which would either mean that protein expression is downregulated or that the number of neurons decreased because the cells died as a result of the CCI. To determine which one of these two processes is responsible for the decrease in Prokr2 expression, immunofluorescence studies will be needed.

Response: The recommended changes have been made in the revised manuscript. We have indicated that Prokr2 expression in CCI tissues was determined by western blotting (Fig.1 e). The quantification of these data was displayed on Fig. 1f which demonstrated that Prokr2 expression increased over time after the CCI.

Fig. 1 (e, f) Western Blot analysis and densitometric quantification of Prokr2 expression in Con, sham and CCI mouse brain tissue. GAPDH is used as a control. Error bars represent S.D. ($n = 3$ mice). * $P < 0.05$ versus sham group.

To clarify whether downregulation of Prokr2 occurred in neuronal cells, we conducted immunofluorescence studies with dual-labeling for Prokr2 and NeuN (Supplementary Fig. 2c and d). In the sham group, Prokr2 was readily detectable in neurons (NeuN positive) – in line with the results shown on Supplementary Fig. 2b. In CCI model, neurons suffering the trauma-induced death and non-neuronal cells were identified as NeuN-negative cells. Surrounding tissues were impacted by trauma but not damaged. Prokr2-positivity was detected in NeuN-positive neurons surrounding the contusion (Supplementary Fig. 2c, the area right to the yellow line) and the

intracellular expression of Prokr2 was not significantly changed after CCI (Supplementary Fig. 2d). Therefore, the total expression of Prokr2- and NeuN-positivity was decreased (Supplementary Fig. 2a) indicating the CCI-induced decrease of the number of viable neurons. These descriptions are included in revised manuscript (Page 5).

Supplementary Fig. 2 (c and d) Immunofluorescence staining showed that Prokr2 is expressed predominantly in NeuN-positive cells. Scale bar is 20µm and 10µm. DAPI is used to stain cell nucleus. Prokr2 fluorescence intensity per NeuN-positive cell is quantified by ImageJ. Error bars represent S.D. ($n = 4$ in sham group and $n = 3$ in CCI group).

5- In Fig. 1h, the purity of the primary cortical neuronal cultures is impossible to determine based on the provided images.

Response: A new higher quality image clearly demonstrating the high purity of primary cortical neurons has been displayed in the revised manuscript (Fig. 1h). To study the purity of the neuron cultures, cells were immunostained with an antibody specific to neuronal marker protein microtubule-associated protein 2 (MAP2, green) and neuronal marker (NeuN, red) (Figure 1h).

Fig.1 (h) Representative photomicrographs of NeuN (red) and Map2 (green) expressing primary cortical neurons used in the studies. Scale bar is 50 μm for white light image and 20 μm for fluorescence image.

6- At page 6, lines 150-151, the conclusion that Prok2, but not Prokr2, plays a role in ferroptosis-related pathological changes is a hasty and controversial one. At this point of the paper, this conclusion is purely based on the fact that expression levels of the receptor do not seem to fluctuate in the pathological conditions tested in vitro. However, as the authors are aware, expression levels are not the only mechanism that determines receptor function.

Response: In our study, Prokr2 was not affected in various pathological conditions. We agree with the Reviewer that Prokr2 expression levels are not sufficient for defining its function. Evaluation of the Prok2 binding to the Prokr2 is critically important for enacting the biological responses. Therefore, we studied the interactions of Prok2 and Prokr2 using Co-IP assay (Supplementary Fig. 2e). We found that CCI increased the level of Prok2 and decreased the amount of Prokr2 in the total lysates. However, the binding of Prokr2 with Prok2 was not changed by CCI. Reciprocally, Prok2 binding to Prokr2 was also unaffected (Supplementary Fig. 2e). We also found that intracellular Prokr2 level per cell was not significantly changed (Supplementary Fig. 2f and g). These results are compatible with the notion that the total Prokr2

expression decline was due to the reduced number of neuronal (NeuN-positive) cells (Page 5).

Supplementary Fig. 2 (e) Co-immunoprecipitation assays show that interactions between Prok2 and Prokr2 are not changed after CCI. Note the changes in the total expression of two proteins after CCI. (**f** and **g**)

Immunofluorescence studies with dual-labeling for Prok2 (red) and Prokr2 (green). Scale bar is 20 μ m. Prok2 and Prokr2 fluorescence intensities per cell are measured by ImageJ. *** $P < 0.001$ versus sham group in Prok2 expression. Error bars represent S.D. ($n = 7$ in sham group and $n = 5$ in CCI group).

7- At page 6, lines 157-159, the authors state that “Out of three Prok2-interfering sequences tested, one was found most efficient in decreasing Prok2 expression”, but Suppl. Fig. 5a-b shows no statistical difference between shRNAs in terms of knockdown efficiency.

Response: In revised manuscript, former Supplementary Fig. 5a-b are now presented as Fig. 2 a and b. (Page 6-7)

Three Prok2-interfering sequences were tested and shown to decrease Prok2 expression. shProk2-3 showed significantly higher knockdown efficiency vs. shProk2-1 and shProk2-2. This difference in the interfering efficiency was marked with an asterisk but the group comparisons were not clearly indicated in the original submission. Group comparisons are now clearly shown by a horizontal line between groups on the bar graph in Fig. 2b.

Fig. 2 (a and b) shProk2 decreased *Prok2* mRNA and one of them (3#) is the most efficient. * $P < 0.05$, ** $P < 0.01$ and *** $P < 0.001$. Error bars represent S.D. ($n = 3$ experiments).

8- In Fig. 2b, high magnification images are required to be able to fully visualize the extent of the colocalization between TUNEL and NeuN stainings.

Response: We apologize for presenting low-quality and low-magnification images in the previously submitted version of the manuscript. We have included high magnification images in Fig.2c of the revised manuscript.

Erastin treatment increased the number of TUNEL- and NeuN-positive cells, which was alleviated by Prok2. However, decreased Prok2 expression further enhanced Erastin-induced neuronal death preventable by Fer-1. Similarly, stretch injury model resulted in increased TUNEL positivity of primary cortical neurons. While Prok2 overexpression decreased, Prok2 shRNA treatment increased stretch-induced primary neuronal death assessed by TUNEL. These new data are presented in Supplementary Fig. 6a and b in the revised manuscript.

Fig. 2 (c and d) Cell death measured by TUNEL staining of primary neurons. Scale bar is 50µm. * $P < 0.05$ and *** $P < 0.001$. Error bars represent S.D. ($n = 4$ experiments).

Supplementary Fig. 6 (a-b) Primary cortical neurons are exposed to mechanical stretch injury and cell death is assessed by TUNEL staining at 6h after stretch. Scale bar is 50µm. * $P < 0.05$ and *** $P < 0.001$. Error bars represent S.D. ($n = 3$ experiments)

9- In Fig. 2d, what is perceived as a morphological sign of neuronal injury should be described in more detail in the Results section.

Response: We apologize for insufficient details in the description of neuronal injury corresponding to the data in previous Fig. 2d. Moreover, additional modified images are shown in Fig. 2e.

In the primary neurons exposed to Erastin for 24h, disruption and thinning of the neurites with large vacuoles and bright spots as well as decreased cytoplasm were observed. Cellular and neurite fragments were also observed in extracellular compartment (Fig. 2e). In the revised manuscript, the stretch model has also been performed in response to the comment 1 of Reviewer 1. In this case, the neuronal injury was shown by labelling with anti-Tau. Stretch induced neuronal injury manifested as the appearance of thin and discontinuous neurites and loss of the cytoplasm (Fig. 1i).

e

Fig. 2e Representative photomicrographs of control and Erastin treated primary cortical Prok2-deficient or Prok2- overexpressing neurons. Characteristic of neuronal injury are disruption and thinning of the neurites with large vacuoles and bright spots as well as decreased cytoplasm. Cellular and neurite fragments are also observed in extracellular compartment. Scale bar is 20 µm.

10- In Suppl. Fig. 7a, the action potentials that are shown in the middle graph for the cortical neurons treated with Erastin+Vector are extremely difficult to appreciate.

Response: We used registration of the electrophysiological activity as a functional test for characterization of the treatments with Erastin + vector on primary cortical neurons. The cells were isolated from newborn mouse brain tissues and placed on the dish to be maintained alive *in vitro*. In the control group (Supplementary Fig. 9a upper graph), the action potentials, appearing as continued waves, were induced under appropriate stimulation (pA). Erastin-induced damage to primary neurons resulted in

a drastic decrease of action potentials in the patch clamp recordings. Prok2 overexpression caused alleviation of Erastin-induced damage and continued action potentials were seen in Erastin+Lv-Prok2 group. In Erastin + vector group, conducting electrophysiological activity measurements were technically more difficult because of the shrinkage of the cytoplasm caused by Erastin. This resulted in the electrode slipping easily and breaking the membrane. When the electrode was stable, the recording of action potentials was feasible.

We replaced with new representative image in revised Supplementary Fig. 9. Overall, a low-frequency and a smaller number of action potentials were found in Erastin + vector group.

11-In Fig. 3a, the authors claim that mitochondrial injury can be detected using the Mito-Tracker. Please describe in the Results section (page 7) the criteria that were used to define “mitochondrial injury”.

Response: We employed Mito-Tracker staining to examine morphology and distribution of mitochondria exposed to Erastin (24 h after treatment, Fig. 3a) in the cytoplasm and neurites. Most cells were represented by a fusiform structure, a small rod-like form in neurites and an interconnected network in cytoplasm, which were the dominant morphologies revealed by the Mito-Tracker staining in normal primary neurons. Upon Erastin treatment, mitochondrial network looked disrupted and mitochondria appeared as small fragmented punctiform structures and small circles. Injury and disruption of neurites was accompanied by a decline in mitochondrial length as well. Mitochondrial circularity and length were presented as indices of mitochondrial injury (Fig 3b, c) as described previously¹². This description has been included in the revised manuscript (Page 8).

12- In Figs. 3e & 6k, transmission electron microscopy (TEM) was used to reveal the presence of ruptures in the outer mitochondrial membrane of Erastin-treated neurons, but again this is far from clear in the provided images without any explanation whatsoever.

Response: As recommended by the Reviewer, we provided a more detailed presentation of changes in the morphology of mitochondria revealed by TEM with references to the previous studies which provided good description of the ferroptosis-related mitochondrial changes. These changes include mitochondria shrinking as a distinctive morphological feature of Erastin-treated cells¹³ and outer mitochondrial membrane rupture was found to be characteristic of RSL3-treated Acsl4 WT cells⁷

These ferroptotic features were also observed in our experiments. We detected mitochondrial shrinking, outer membrane rupture and the formation of light vacuoles likely related to the mitochondria collapse. We presented these descriptions in the revised manuscript (Page 8-9) and included arrows pointing to these ferroptotic features of mitochondria on the images (Figs. 3e and 6j). The figure legends were also revised accordingly.

13- In Fig. 5h, the schematic diagram provided is not enough detailed to understand the complexity of the dataset and overall perspective.

Response: We developed a new schematic diagram that combines the characteristic features of ferroptotic pathways triggered by TBI and Erastin in primary neurons. The diagram includes the major pro- and anti-ferroptotic regulatory mechanisms leading to lipid peroxidation and mitochondrial damage. It also emphasizes the role of Prok2 overexpression in attenuating neuronal ferroptosis via accelerated Fbxo10-driven ubiquitination and degradation of Acsl4. This newly modified schema is displayed in Fig.8 of the revised manuscript.

Fig. 8 A schematic diagram showing the neuroprotective and anti-ferroptotic effects of Prok2. TBI and Erastin in primary neurons trigger ferroptotic process, resulting in lipid peroxidation, shrunken mitochondrial morphology and neuronal damage. Overexpression of Prok2 is anti-ferroptotic and neuroprotective by upregulating the expression of Fbxo10, a E3 ubiquitin ligase that binds to Acsl4, and thereby inducing ubiquitination degradation of Acsl4.

14- In Figs. 6-7, mice were injected with AAV-Prok2 before CCI instead therapeutically, which I suppose is because AAVs take time to transduce brain cells efficiently in vivo following i.c.v. administration. Perhaps this should be discussed and put into relation with the progression of secondary damage in the CCI model (in terms of timing) and treatment issues and challenges in human TBI.

Response: The Reviewer is correct: AAV-driven gene transfection takes time. In our experiments, 7 days were necessary to reach efficient transfection. After this period,

stable transfection persisted for many days, up-to three months (results are not shown). According to our experimental design, the behavioral tests, including rotarod test and Morris Water Maze test, were performed within 1 month after the transfection.

TBI causes primary mechanical injury to the brain tissue leading to immediate necrotic neuronal death, which is not amenable to therapy but can only be prevented. Clinically, the necrotic core of a contusion resulting from brain trauma can be removed surgically. Secondary injury cascades follow primary injury and involve mitochondrial dysfunction, lipid peroxidation and inflammation accompanied by delayed neuronal death particularly in the regions surrounding the contusion. The secondary injury mechanisms leading to delayed neuronal death can be targeted therapeutically. Our treatment strategy with AAV-mediated overexpression of Prok2 in target brain regions prior to CCI was effective in attenuating CCI-induced neuronal death and improving neurocognitive outcomes indicating that this pathway is amenable for therapeutic targeting.

Use of AAV vector gene therapy for aromatic l-amino acid decarboxylase (AADC) deficiency has been reported in patients¹⁴. Implementation of this approach is impractical in human TBI as patients seek for treatment only after TBI as the occurrence of cannot be predicted. While we did not evaluate post-treatment potential of Prok2 overexpression in preventing ferroptotic neuronal death after TBI, recent studies showed that ferroptotic neuronal death is a viable therapeutic target after TBI^{15,16}. And our study also uncovered a molecular mechanism that could offer a potential target to suppress ferroptotic neuronal death.

This discussion has been summarized in the revised manuscript (Page 18).

15- In Fig. 6, the quality of the images provided in panel “b” showing eGFP signal in the brain of mice injected with AAV-Prok2-eGFP viruses is insufficient. High magnification images are required to be able to visualize the cells that were transduced. From what can be seen in the low magnification images provided, eGFP appears to be expressed in virtually all cells but neurons. Therefore, the target cell specificity (i.e. neurons according to Fig. 1) was probably not conferred by the AAV-Prok2-eGFP viruses.

Response: In response to the Reviewer’s critique, high magnification images are presented in the revised manuscript. Dual-label immunofluorescent staining was used to detect the Prok2 overexpression and location. We found that the Prok2 overexpression (green) was mainly attributable to neuronal cells (NeuN, red). In addition to the previously shown split images, the merged image was included in the revised Fig. 6.

Fig. 6 (a) GFP-tagged Prok2-AAV is injected into mouse brain 1 week before CCI. Dual-labeling immunofluorescence staining with Prok2-eGFP (green) and neurons (NeuN, red) is used to test the efficiency of AAV transfection. Scale bar is 30µm.

16- The provenance, background, sex and age of mice is not indicated in the Methods.

Response: We apologize for this omission. The requested information has been included in the Methods section of the revised manuscript (Page 21).

8-week male C57BL/6 mice (Animal Core Facility of Nanjing Medical University, Nanjing, China) were subjected to severe CCI.

17- Logically, the training on the rotarod probably started before the CCI injury. Therefore, please validate whether the following information is correct or not (lines 753-754): “The training began 1 days after TBI, and the mice underwent three trials every day, which made mice adapt to the instruction.”

Response: We apologize for the typo. This has been corrected in the revised manuscript (Page 30-31).

The training began 1 day *before* TBI, and the mice underwent three trials, which made mice adapt to the testing instructions. Then, the rotarod test was performed on day 2 post-CCI by an investigator blinded to the experimental group assignments.

Minor points:

1) Several abbreviations were used in the Abstract without being defined.

Response: This has been corrected. In the revised manuscript, all abbreviations are defined at first mention.

2) Page 5, line 136: The time of exposure to H₂O₂ is 12 hours, not 24 hours.

Response: The typo has been corrected: Primary neurons were exposed to H₂O₂ for 12 h.

3) Page 7, lines 185-187: The following sentence is incomplete: “Because healthy electrophysiological activities are necessary for neurons to exert their biological functions.”

Response: We further examined whether Prok2 upregulation could change neuronal electrophysiological activity after exposure to Erastin. In the control group, action potentials, appearing as continued waves, were induced under appropriate stimulus (pA) (Supplementary Fig. 9a upper graph). Erastin-induced damage to primary neurons resulted in a drastic decrease of action potentials in the patch clamp recordings. Prok2 overexpression caused alleviation of Erastin-induced damage and continued action potentials were detectable in the Erastin + Lv-Prok2 group (Supplementary Fig. 9a and b).

Incomplete sentence is deleted and the above description has been summarized in the revised manuscript.

4) In Fig. 6, panel “a” is dispensable and could be removed.

Response: This panel has been removed as recommended.

5) Throughout the text, please replace “CCI treatment” by “CCI injury”.

Response: As recommended, we replaced “CCI treatment” by “CCI injury” throughout the text. in reviewed manuscript.

6) Some sentences throughout the text, more especially in the Discussion section, are just unreadable. Here are a few examples: Lines 450-451: “We investigated Prok2 upregulation after CCI was source from feedback signal, though preventing cell death despite in vain.” Lines 456-458: “Through comparison their study, we found Prok2 neurotoxic effects were obtained with picomolar concentrations (0.01 and 0.1 nM) they offered.” Line 448: “...which would be explored and preceded in our further study”. Line 516: “Remove the supernatant, re-suspend the precipitate with buffer, and lay the board.” Lines 510-511: “...and their heads were quickly severed.” Lines 517-518: “After 5 hours, buffer was instead by the neurobasal medium containing...” Lines 533-534: “PBS was used to pure from the left ventricle...”

Response: We apologize for these mistakes. The entire text has been thoroughly proofread and corrected, including the unreadable sentences mentioned above.

① In the revised manuscript, we corrected the sentence as follows: “

Both *in vitro* and *in vivo* pro-ferroptotic exposures to Erastin, stretch and CCI increased Prok2 expression, suggesting its sensitivity to ferroptotic stimuli. Further, Prok2 overexpression using AAV-mediated transfection attenuated CCI-induced neuronal death and functional deficits. We speculate that neurons upregulate Prok2 expression as a protective response to ferroptotic insults albeit the amounts of the produced protein are not sufficient to prevent the damage and death. Evidently, a stronger elevation of Prok2 achieved in mice after lentiviral transfection were required to alleviate CCI-induced neuronal injury.” (Page 18)

②Based on the reported Prok2 neurotoxic effects, several studies considered Prok2 as a risk factor after neuronal cell injury (Cheng et al., 2012). This discrepancy may be due to different concentrations of Bv8, a recombinant mouse Prok2, that has been used in primary neurons (0.01 and 0.1 nM) (Severini et al., 2015) in the previous

study. Protective effects were obtained with nanomolar concentrations (10–100 nM). Indeed, Prok2 overexpression by viral transfection in our study could yield neuroprotective response. Thus, it is possible that the effects of Prok2 may be context-specific and concentration-dependent. (page 18)

③ Future studies will be necessary to explore the significance of the changes observed in these other genes after TBI. (page 18)

④ A strainer was used to filter and eliminate tissue fragments. And resultant solution contained various cells, which were pelleted by centrifugation at 1000 rpm for 5 min. The supernatant was removed and cells were resuspended in a dish or in wells containing DMEM: F12 (1:1) supplemented with 10% horse serum and 2% penicillin and streptomycin (Gibco). (Page 20)

⑤ In the section of primary cortical neuronal culture, we mentioned that newborn mice were the source of primary cortical neurons. The newborn mice were first disinfected with 75% alcohol to avoid bacterial contamination. The mouse brain was harvested by decapitation. The skull and perichondrium were then dissected by forceps in a 4 °C pre-cooled buffer. (Page 20)

⑥ At about 5 hours after seeding, DMEM: F12 (1:1) culture solution was removed and replaced with the neurobasal medium containing 2% B27 and 0.5mM glutamate. (Page 20)

⑦ Mice were perfused intracardially with 4 °C phosphate buffer saline (PBS) solution. (Page 20)

7) Please indicate the source and model of the impactor used for CCI.

Response: After anesthesia, mice were placed in a stereotaxic frame (R.W.D. Shenzhen, China) and a 4-mm diameter craniotomy was performed over the left parietal bone and the bone flap was removed for trauma. A vertically directed CCI was delivered (6.0 ± 0.2 m/s, 50 ms dwell time, 1.4 mm depth) using an impactor (R.W.D., Shenzhen, China). After injury, the skin incision was closed. Mice were monitored with supplemental oxygen (100%) for 1 hour before returning to their cages.

The missing information for the impactor has been included in the revised manuscript (Page 21-22).

8) Please indicate the source of the TUNEL detection kit.

Response: The TUNEL detection kit was purchased from Beyotime, China. The Cat No. is C1089. This information was included in the revised Methods section (Page 23).

9) Line 672: the abbreviation for “transmission electron microscopy” should be corrected from TAM to TEM.

Response: Thank you - this has been corrected.

Reviewer #3:

In the present work, the authors provide an interesting link between Prok2 and its protective effect in models of traumatic brain injury. Mechanistically the authors propose that Prok2 drives Fbxo10 expression, which ultimately decreases the expression of ACSL4. Downregulation of ACSL4 has been associated with increased protection against ferroptosis. This concept is novel and of broad interest. The study is exciting and expands our understanding of ferroptosis in neuropathologies. I have a few remarks the authors might want to address.

Response: We appreciate the reviewer's positive assessment of our work. Those remarks were all valuable and served as an important guidance for us to revise and improve our manuscript. Our point-by-point responses to the remarks of the reviewer are presented below.

1. The authors propose a role for Prok2 to regulate ACSL4 levels, but this is not clear from their study. Does Prok2 overexpression lead to ACSL4 degradation per-se? As it is shown the results only point to this activity upon Erastin treatment.

Similarly, does the overexpression of Fbxo10 is sufficient to decrease the ACSL4 downregulation without prior stress?

Response: The Reviewer is right: This work demonstrated that Erastin-induced increases in Acs14 expression could be alleviated by Prok2 overexpression whereby Fbxo10 acted as the key E3-ubiquitin ligase, which was upregulated by Prok2. We also agree with the Reviewer further support is needed to ascertain whether this mechanism is also working in the absence of Erastin stimulation.

We performed additional experiments examining the effect of Prok2 overexpression on Acs14 without Erastin stimulation (Fig. 5g).

We demonstrated that overexpression of Fbxo10 was *sufficient* to decrease Acs14. We found that Fbxo10 was able to directly ubiquitinate Acs14 resulting in its degradation with or without Erastin stimulation (Fig. 5d). We also found that Prok2 overexpression led to ~50% elevation of Fbxo10 expression and ~40% decrease in Acs14 levels. We believe that this magnitude of Acs14 downregulation is not sufficient for affecting neuronal survival without Erastin. Consistent with this, assessments of the MDA level(**a**), LDH level(**b**) and cell viability(**c**) without Erastin administration showed that Prok2 overexpression did not change either the MDA contents or cell viability (results are not shown in the revised manuscript).

(a) Lipid peroxidation tested by MDA levels. (b) Cytotoxicity tested by LDH. (c) Cell viability tested

by CCK-8. Data represents the mean \pm S.D. $n = 3$. ns means no significance.

Therefore, we concluded that Prok2-induced decrease of Acsl4 did not act anti-ferroptotically in neurons without Erastin stimulation. Although increased Fbxo10 caused Acsl4 degradation with or without Erastin, Prok2 induction of Fbxo10 upregulation was more robust and effective in combination with the Erastin treatment. In other words, Erastin was a necessary “second hit” required for the initiation of pro-ferroptotic response. It should be noted, however, that the delayed secondary phase of TBI pathogenesis includes severe oxidative stress and depletion of GSH^{15, 17, 18} that may act similar to Erastin-induced GSH exhaustion.

2. Also, it is unclear if the downregulation of ACSL4 (roughly 50%) is sufficient to modify the neuronal “lipidome”. The authors might need to comment on this.

Response: This is a very important point made by the Reviewer. In order to examine whether that Prok2-induced downregulation of Acsl4 (roughly 50%) could modify the neuronal “lipidome”, we performed the LC-MS/MS analysis of PE species in primary cortical neurons. The major PE species are shown as a heatmap (**a**). Notably, molecular species of PE (18:0/20:4) and PE (18:0/22:4) representing known substrates for oxidation during ferroptosis⁷ are downregulated by Prok2 overexpression (**b** and **c**).

LC-MS/MS analysis of PE species in primary cortical neurons. **(a)** Heatmap of all major PE species of the vector and Lv-Prok2 treated neurons. The log₂ values are calculated for each sample by normalizing to counting number reads alone. **(b and c)** Effects of Prok2 on two major molecular species of PE (18:0/20:4) and PE (18:0/22:4) representing substrates for oxygenation during ferroptosis. The contents of PE (18:0/20:4) and PE (18:0/22:4) are normalized to total protein. Data represents the mean ± S.D. $n = 3$, $*P < 0.05$ and $***P < 0.001$.

Here, we showed experimental methods to the lipidome assay (performed by LipidALL Technologies Company Limited, Changzhou 213022, Jiangsu Province, China).

Lipidomics Analyses

Lipid were extracted from approximately 1×10^6 cells using a modified version of the Bligh and Dyer's method. Briefly, cells were incubated in 750 μ L of chloroform:methanol 1:2 (v/v) with 10 % deionized water for 30 min. At the end of the incubation, 350 μ L of deionized water and 250 μ L of chloroform were added. The samples were then centrifuged and the lower organic phase containing lipids was extracted into a clean tube. Lipid extraction was carried out twice and the lipid extracts were pooled into a single tube and dried in the SpeedVac under OH mode. Samples were stored at -80 °C until further analysis. Phosphatidylethanolamines (PEs) were analyzed using an Exion UPLC system coupled with a triple quadrupole/ion trap mass spectrometer (6500 Plus Qtrap; SCIEX) as described previously¹⁹. Separation of lipids by normal phase (NP)-HPLC was carried out using a Phenomenex Luna 3 μ -silica column (internal diameter 150 \times 2.0 mm) with the following conditions: mobile phase A (chloroform: methanol: ammonium hydroxide, 89.5:10:0.5) and mobile phase B (chloroform: methanol: ammonium hydroxide: water, 55:39:0.5:5.5). MRM transitions were set up for analysis of various PEs. Individual PEs were quantified by using DMPE from Avanti Polar Lipids as an internal standard.

Indeed, incomplete elimination of Acsl4 is likely insufficient for triggering the anti-ferroptotic program without TBI, stretch or Erastin stimulus. However, the pro-inflammatory secondary response to TBI is associated with severe shift in the redox balance and deficiency of reducing equivalents, including GSH. This pro-oxidant environment stimulates peroxidation of polyunsaturated phospholipids such as arachidonoyl-PE, that act as pro-ferroptosis signals²⁰. Further detailed mechanistic studies will elucidate whether site-specific exhaustion of PUFA-PLs along with other oxidative stress factors act synergistically in neuronal compartments where the ferroptosis is initiated. The current work demonstrate that Prok2 is one of the important regulators that causes Acsl4 ubiquitination/degradation thus contributing to the creation of anti-ferroptotic environment. This interpretation is supported by our data demonstrating decreased lipid peroxidation (MDA levels) in Erastin + Lv-Prok2

group (Supplementary Fig.7d). The mean value for the Erastin group was 1.12 ($\mu\text{M}/\text{mg}$) but 0.70 in the Erastin +Lv-Prok2 group ($P=0.023$). Thus, about 38% lipid peroxidation was blocked by downregulation of Acsl4 (roughly 50%) induced by Prok2. Similar results were also obtained in neuronal stretch model in vitro (Supplementary Fig. 8e). The mean MDA value for the stretch group was 1.115 vs. 0.775 ($\mu\text{M}/\text{mg}$) for the stretch+Lv-Prok2 group ($P=0.0298$). This represents $\sim 32\%$ suppression of lipid peroxidation. Accompanied by decreased MDA levels after Prok2 overexpression, cell viability was increased (from 45.56% to 75.89% in Erastin administration and from 45.72% to 74.85% in stretch model). Therefore, Prok2-induced Acsl4 downregulation (roughly 50%) could decrease the levels of neuronal lipid peroxidation and improve cell viability after Erastin or stretch exposure.

In CCI model, our data also demonstrated attenuation of lipid peroxidation (MDA levels) in CCI + AAV-Prok2 group (Supplementary Fig. 15d). The mean value for the CCI group was 6.72 ($\mu\text{M}/\text{mg}$) but 4.22 ($\mu\text{M}/\text{mg}$) in the CCI +AAV-Prok2 group ($P=0.037$). Thus, about 38% lipid peroxidation was blocked. In the assessment of 12-HETE and 15-HETE, the metabolites of arachidonic acid, we found 43% and 38% reduction in CCI+AAV-Prok2 group, respectively. Therefore, Prok2-induced downregulation of ACSL4 (roughly 50%) is also sufficient to modify the neuronal lipid peroxidation induced by CCI.

3. The authors mostly focused on stress-induced by Erastin, yet this is not relevant in the context of neurons given their extremely low expression of SLC7A11. It would be essential to demonstrate if triggers such as GPX4 inhibitors could phenocopy these effects.

Response: This is an interesting point for discussion. SLC7A11 (also called system *xc-*) is present in neurons and astrocytes in the mouse and human brain as well as in the border areas between the brain and the blood or cerebrospinal fluid including

vascular endothelial cells, ependymal cells, choroid plexus, and leptomeninges²¹. Low expression of SLC7A11 in neurons suggests that these cells are vulnerable to depletion of GSH - one of the hallmarks of the delayed responses to TBI^{15, 17, 18}. In cultures, primary astrocytes and neurons or neuronal and glial cell lines, the dependence on system *xc-* for GSH synthesis is strong because of the highly pro-oxidant conditions of the media favoring the conversion of the extracellular cysteine to cystine²². Therefore, in neural cells, Erastin-induced GSH depletion readily creates pro-ferroptotic environment. Several earlier studies have documented the importance of Erastin-induced pro-ferroptotic conditions in neuronal injury, including HT22 cell line and primary cortical neurons^{5, 23}. In our study, we found that CCI did not affect the expression of GPX4 *in vivo*. Similarly, stretch injury had little effect on GPX4. These considerations were the major reason for our choice of Erastin, rather than of specific GPX4 inhibitors - (1S, 3R)-RSL3 or ML-162 – for our study. To respond to the Reviewer’s suggestion, we employed RSL3 (HY-100218A, MCE) and ML-162 (MSL2561, Sigma), as Gpx4 suppressors and tested whether Prok2 changes could affect the ferroptotic responses. The results showed that both RSL3 and ML-162 effectively induced cytotoxicity (LDH levels) in primary cortical neurons (**a-b**). We tested whether Prok2 triggered neuroprotective responses in models of RSL3 and ML-162-induced cellular injury.

(a) 0.5 μM RSL3 efficiently induces increased LDH level in primary neurons. * $P < 0.05$ and ** $P < 0.01$ versus 0 μM group ($n = 3$ experiments). (b) 0.25 μM ML-163 efficiently promotes cytotoxicity assayed by LDH. ** $P < 0.01$ and *** $P < 0.001$ versus 0 μM group ($n = 3$ experiments). (c) Prok2 overexpression alleviates LDH level at 0.5 μM RSL3 and 0.25 μM ML-163. (d) Prok2 causes decreased lipid peroxidation at 0.5 μM RSL3 or 0.25 μM ML-162. Note: both RSL3 and ML-162 are used to treat cells for 24h. * $P < 0.05$ and ** $P < 0.01$ ($n = 3$ experiments). Error bars represent S.D. These results were not included in the revised manuscript.

Prok2 overexpression alleviated LDH release from cells prompted to ferroptosis by 0.5 μM RSL3 or 0.25 μM ML-162 (c). Declined lipid peroxidation (MDA levels) was found after Prok2 overexpression (d).

Therefore, we concluded that cells transfected with Lv-Prok2 responded to GPX4 inhibitors-induced lipid peroxidation (MDA levels) and cytotoxicity (LDH levels). Given the recently identified alternative to GPX4 enzymatic anti-ferroptotic redox mechanisms (such as FSP1²⁴, iNOS/NO•²⁵), it is possible that Prok2 regulation of ferroptosis is also associated with its GSH-mediated effects on these alternative pathways. Therefore, Prok2 changes could affect the ferroptotic responses induced by GPX4 inhibitors.

4. Moreover given the broad importance of this regulatory link the authors could address if this is limited to neurons or could be generally applicable to different cell types.

Response: In our study, we explored the effect of Prok2 in neurons. At the initial design, we addressed the location of Prok2 in neurons, astrocytes and microglia cells. Based on the results of Brain RNA data base (https://web.stanford.edu/group/barres_lab/brain_rnaseq.html), Prok2 RNA is mainly present in astrocytes and neurons, but low in microglia cells (Supplementary Fig. 1e). In dual-immunofluorescent staining with Prok2 and cell markers, we found CCI-induced increase in Prok2 mainly expressed in neurons, and to a much lesser extent in astrocytes and microglia cells (Fig.1 g). In addition, Prokr2, the receptor for Prok2, was present in large amounts in neurons. Therefore, it was appropriate to manipulate the neuronal Prok2 expression and explore the ferroptosis related biological changes. However, we cannot exclude Prok2 as an important regulator of ferroptosis in astrocytes and microglial cells. We searched some available publications to discuss.

Dr. Marimélia Porcionatto's team¹⁰ found glutamate- and amyloid-beta-induced toxicity resulted in upregulation of Prok2 in both neurons and astrocytes *in vitro*. They were also surprised to find that Prok2 was not expressed *in vivo* by astrocytes (GFAP+ cells) 7 days after the injury. But they demonstrated that reactive microglia induced by brain injury could express Prok2, not detected in the intact brain. Their results were different from our study that we found little Prok2 expression in microglial cells with or without brain injury stimulus.

However, an earlier study in 2012 also elucidated the results that no expression of PROK2 by cultured microglia⁹.

As for astrocytes, Dr. Arthi Kanthasamy's team demonstrated that Prok2 induced an alternative A2 astrocyte reactive phenotype and lenti-Prok2 overexpression induced the anti-oxidant response proteins arginase-1 and Nrf2⁸.

In addition, the regulation of ferroptosis process might occur differently between different cell types. A Previous study reported that BDNF-mediated Nrf2 activation in

astrocytes might be transported via microparticles/exosomes and increase neuronal resistance to ferroptosis²⁶. Another study about microglial cells showed that NO• donors and/or suppression of NO• production by iNOS inhibitors represent a novel redox mechanism of regulation of ferroptosis in pro-inflammatory conditions, which affect neural death subsequently²⁵.

Therefore, based on above controversial results, Prok2 regulation of ferroptosis may be also operational in astrocytes and microglia and this will be the subject of future research.

5. Might be unrealistic for the revision – but a key experiment defining this link would have been to demonstrate if AVV delivery of Prok2 effect is lost in an ACSL4 knockout background. Perhaps given this lack of “hard-evidence” the authors might wish to tone down some of their statements.

Response: This is an excellent suggestion from the Reviewer. We found that Prok2 neuroprotective effects and preservation of the mitochondrial function were dependent on downregulation of *Acsl4* *in vitro*. However, we did not verify this in CCI model. Therefore, we designed additional experiments to examine whether Prok2 effects were realized at lowered expression of *Acsl4* *in vivo* as follows (Supplementary Fig. 14).

Supplementary Fig. 14 Prok2 overexpression alleviates CCI-induced ferroptosis in an Acs14 dependent manner. (a) AAV-shAcs14 jointly with HA is injected into mouse brain prior to CCI. Immunostaining for HA is used to test the transfection efficiency. Hoechst staining of the nuclei is employed. (b-d) Western blot assays indicate that Acs14 significantly declines after AAV-shAcs14 treatment. GPX4 expression is not changed in Acs14 knockdown group. ** $P < 0.01$ Error bars represent S.D. ($n = 3$ mice per group). (e-j) Overexpression of Prok2 prevents CCI-induced decrease in GPX activity, total GSH levels and the GSH/GSSG ratio and increase in MDA levels and the number of TUNEL and NeuN positive cells. These effects of Prok2 overexpression are lost in Acs14 knock down group. ** $P < 0.01$ and *** $P < 0.001$. Error bars represent S.D. ($n = 3$ mice per group).

We downregulated Acsl4 expression by intracranial injection of AAV-shAcsl4 conjoined with HA. Immunofluorescence assay was used to detect the transfection efficiency (Supplementary Fig. 14a). Knockdown efficiency was also assessed by immunoblotting (Supplementary Fig. 14b, c). The low Acsl4 content caused by AAV-shAcsl4 remained at this low level after CCI. In AAV-shAcsl4 group, Gpx4 was not changed after CCI but it increased in Prok2 overexpression group (Supplementary Fig. 14b-d). Overexpression of Prok2 prevented CCI-induced decreases in GPX activity, total GSH, GSH:GSSG ratio and CCI-induced increases in MDA levels and the number of TUNEL and NeuN positive cells in an ACSL4 depended manner (Supplementary Fig. 14e-j). (Page 13)

Minor –

The manuscript could benefit from professional editing; some sentences are repetitive or grammatically wrong which make the reading tedious.

Response: The revised manuscript has been thoroughly proofread and repetitive sentences and grammatical mistakes fixed.

Once again, we thank the reviewers for the positive and constructive comments on our work. We believe that we have fully addressed all the concerns and critiques.

Sincerely

On behalf of authors

Jing Ji, Ph.D, M.D.
Department of Neurosurgery
the First Affiliated Hospital of Nanjing Medical University
300 Guangzhou Road
Nanjing, Jiangsu
China

References

1. Ji J, *et al.* Lipidomics identifies cardiolipin oxidation as a mitochondrial target for redox therapy of brain injury. *Nature neuroscience* **15**, 1407-1413 (2012).
2. Shen L, *et al.* Ferroptosis in Acute Central Nervous System Injuries: The Future Direction? *Frontiers in cell and developmental biology* **8**, 594 (2020).
3. Li Y, *et al.* Inhibition of Ferroptosis Alleviates Early Brain Injury After Subarachnoid Hemorrhage In Vitro and In Vivo via Reduction of Lipid Peroxidation. *Cellular and molecular neurobiology*, (2020).
4. Wang YQ, *et al.* The Protective Role of Mitochondrial Ferritin on Erastin-Induced Ferroptosis. *Frontiers in aging neuroscience* **8**, 308 (2016).
5. Zhang Y, *et al.* Neuroprotective effect of deferoxamine on erastin-induced ferroptosis in primary cortical neurons. *Neural regeneration research* **15**, 1539-1545 (2020).
6. Wu J, *et al.* Intercellular interaction dictates cancer cell ferroptosis via NF2-YAP signalling. *Nature* **572**, 402-406 (2019).
7. Doll S, *et al.* ACSL4 dictates ferroptosis sensitivity by shaping cellular lipid composition. *Nature chemical biology* **13**, 91-98 (2017).
8. Neal M, *et al.* Prokineticin-2 promotes chemotaxis and alternative A2 reactivity of astrocytes. *Glia* **66**, 2137-2157 (2018).
9. Cheng MY, *et al.* Prokineticin 2 is an endangering mediator of cerebral ischemic injury. *Proceedings of the National Academy of Sciences of the United States of America* **109**, 5475-5480 (2012).
10. Mundim MV, *et al.* A new function for Prokineticin 2: Recruitment of SVZ-derived neuroblasts to the injured cortex in a mouse model of traumatic brain injury. *Molecular and cellular neurosciences* **94**, 1-10 (2019).
11. Liu XQ, Wei RR, Wang CC, Chen LY, Liu C, Liu K. Relationship between PK2 and number of Kupffer cells during the progression of liver fibrosis in patients with HBV. *Turkish journal of medical sciences* **48**, 52-61 (2018).
12. Gordon R, *et al.* Prokineticin-2 upregulation during neuronal injury mediates a compensatory protective response against dopaminergic neuronal degeneration. *Nature communications* **7**, 12932 (2016).

13. Kajarabille N, Latunde-Dada GO. Programmed Cell-Death by Ferroptosis: Antioxidants as Mitigators. *International journal of molecular sciences* **20**, (2019).
14. Kojima K, *et al.* Gene therapy improves motor and mental function of aromatic l-amino acid decarboxylase deficiency. *Brain : a journal of neurology* **142**, 322-333 (2019).
15. Kenny EM, *et al.* Ferroptosis Contributes to Neuronal Death and Functional Outcome After Traumatic Brain Injury. *Critical care medicine* **47**, 410-418 (2019).
16. Xie BS, *et al.* Inhibition of ferroptosis attenuates tissue damage and improves long-term outcomes after traumatic brain injury in mice. *CNS neuroscience & therapeutics* **25**, 465-475 (2019).
17. Tyurin VA, *et al.* Oxidative stress following traumatic brain injury in rats: quantitation of biomarkers and detection of free radical intermediates. *Journal of neurochemistry* **75**, 2178-2189 (2000).
18. Bayir H, *et al.* Assessment of antioxidant reserves and oxidative stress in cerebrospinal fluid after severe traumatic brain injury in infants and children. *Pediatric research* **51**, 571-578 (2002).
19. Song JW, *et al.* Omics-Driven Systems Interrogation of Metabolic Dysregulation in COVID-19 Pathogenesis. *Cell metabolism* **32**, 188-202 e185 (2020).
20. Stockwell BR, *et al.* Ferroptosis: A Regulated Cell Death Nexus Linking Metabolism, Redox Biology, and Disease. *Cell* **171**, 273-285 (2017).
21. Burdo J, Dargusch R, Schubert D. Distribution of the cystine/glutamate antiporter system xc- in the brain, kidney, and duodenum. *The journal of histochemistry and cytochemistry : official journal of the Histochemistry Society* **54**, 549-557 (2006).
22. Lewerenz J, *et al.* The cystine/glutamate antiporter system x(c)(-) in health and disease: from molecular mechanisms to novel therapeutic opportunities. *Antioxidants & redox signaling* **18**, 522-555 (2013).
23. Nagase H, Katagiri Y, Oh-Hashi K, Geller HM, Hirata Y. Reduced Sulfation Enhanced Oxytosis and Ferroptosis in Mouse Hippocampal HT22 Cells. *Biomolecules* **10**, (2020).

24. Bersuker K, *et al.* The CoQ oxidoreductase FSP1 acts parallel to GPX4 to inhibit ferroptosis. *Nature* **575**, 688-692 (2019).
25. Kapralov AA, *et al.* Redox lipid reprogramming commands susceptibility of macrophages and microglia to ferroptotic death. *Nature chemical biology* **16**, 278-290 (2020).
26. Ishii T, Warabi E, Mann GE. Circadian control of BDNF-mediated Nrf2 activation in astrocytes protects dopaminergic neurons from ferroptosis. *Free radical biology & medicine* **133**, 169-178 (2019).

REVIEWERS' COMMENTS

Reviewer #1 (Remarks to the Author):

None

Reviewer #2 (Remarks to the Author):

This study by Bao et al. is a resubmission of a previously rejected manuscript. I have to say that the authors did an amazing job with the revision. I truly commend them for the enormous amount of work needed to address the entirety of the issues raised by the reviewers and for the quality of the editorial work that has been done on the article. I have no further questions or concerns; the authors have satisfactorily addressed all comments.

Reviewer #3 (Remarks to the Author):

The authors have been remarkably diligent in this review. The manuscript has substantially improved and I have no further comments at this stage.